



# Joint spectral retrievals of ozone with Suomi NPP CrIS augmented by S5P/TROPOMI

Edward Malina[1], Kevin W. Bowman[1], Valentin Kantchev[2], Le Kuai[1], Thomas P. Kurosu[1], Kazuyuki Miyazaki[1], Vijay Natraj[1], Gregory B. Osterman[1], and Matthew D. Thill[1]

[1]Jet Propulsion Laboratory, California Institute of Technology, Pasadena, California, USA
[2]Swift Software Group, Glendale, California, USA

**Correspondence:** Edward Malina (edward.malina.13@ucl.ac.uk)

**Abstract.** The vertical distribution of ozone plays an important role in atmospheric chemistry, climate change, air pollution, and human health. Over the twenty-first century, spaceborne remote sensing methods and instrumentation have evolved to better characterise this distribution. We quantify the ability of ozone retrievals to characterise this distribution through a combination of thermal infrared (TIR) and Ultra Violet (UV) spectral radiances, harnessing co-located TIR measurements from the Cross Track Infrared Sounder (CrIS), onboard the Suomi National Polar-orbiting Partnership (NPP), and UV measurements from the TROPospheric Monitoring Instrument (TROPOMI), which is on the Sentinel 5-Precursor (S5P) satellite. The combination of TIR and UV measurements improves the ability of satellites to characterise global ozone profiles, over the use of each band individually. The CrIS retrievals enhanced by TROPOMI radiances in the Huggins band (325-335 nm) show good agreement with independent datasets both in the troposphere and in the stratosphere in spite of calibration issues in the TROPOMI UV. Improved performance is characterised in the stratosphere from CrIS-TROPOMI. Comparable performance between CrIS-TROPOMI and CrIS-only is found in the troposphere with degrees of freedom for signal of about 2 globally, but higher in the tropics partitioned equally between the lower and upper troposphere. These results demonstrate that CrIS/TROPOMI retrievals have the potential to substantially improve our understanding of ozone. If spectral accuracy is improved in future TROPOMI calibration, the degrees of freedom of signal in the stratosphere could double when using bands 1 and 2 of TROPOMI (270-330 nm), while tropospheric degrees of freedom of signal could increase by 25%.

## 1 Introduction

Characterisation of the ozone vertical distribution is essential to separate its impacts on climate change, global atmospheric chemistry and human health (Szopa et al., 2021; WHO, 2003). Ozone present in the stratosphere ($\sim$90% of the total) is responsible for filtering out harmful UV radiation in the UV-A and B bands (<300 nm). In the stratosphere, ozone is produced by UV radiation destroying $O_2$ bonds, and liberated oxygen atoms bonding to form ozone, while it is destroyed by photochemical reactions. Ozone in the troposphere ($\sim$10% of the total) is produced by downward transport of stratospheric ozone, and from





the interaction of nitrogen oxides ($NO_x$ = NO + $NO_2$) and non-methane organic carbons (NMOCs) in the presence of sunlight (Szopa et al., 2021). In the troposphere, NOx is typically the limiting precursor in ozone production, since NMOCs are present

in high volumes (Szopa et al., 2021). As a large fraction of global NOx emissions are due to anthropogenic sources, ozone pollution is common in metropolitan areas (Jaffe and Wigder, 2012), and has a typical lifetime from several hours to several weeks depending on the local conditions (Young et al., 2013). In the troposphere, ozone acts as a greenhouse gas (GHG), contributing to atmospheric radiative forcing, as well as acting as a pollutant near the surface (Bowman et al., 2013; IPCC, 2013). Therefore, tropospheric ozone variability impacts both human health and climate change and is not well understood.

This is underscored by the classification of tropospheric ozone by the National Academy of Sciences 2017 decadal survey as a 'most important' objective (NASES, 2018).

In order to understand ozone variability, the vertical distribution of ozone must be quantified; this can be achieved through a number of remote sensing or in-situ methods. For example, electrochemical concentration cell ozonesondes, which measure ozone concentration via chemical means are attached as instruments to balloons which measure from the surface up to 30 km

altitude (Komhyr et al., 1995). While highly accurate, ozonesondes offer limited coverage, meaning that satellite instruments must be used to identify global ozone variations. There is a significant heritage in using satellite instruments to estimate ozone. TIR instruments use the $\nu_3$ spectral region (∼9.6 μm), which is sensitive to variations in pressure and temperature, yielding the most information in the free troposphere. Instruments built to exploit the TIR spectral regions include the Tropospheric Emissions Spectrometer (TES) (Bowman et al., 2006), the Infrared Atmospheric Sounding Interferometer (IASI) (Clerbaux

et al., 2010), the Atmospheric Infrared Sounder (AIRS) (Fu et al., 2018), and the Cross-track Infrared Sounder (CrIS) (Bloom, 2001). The CrIS instrument is present on multiple satellites (e.g., Suomi NPP and NOAA-20), and is also expected to be on future satellites (NOAA-21 and 22). UV instruments derive atmospheric profile information about ozone through back-scattered solar UV radiance. The UV measurement is primarily sensitive to the stratosphere due to Rayleigh scattering (Chance et al., 1997); vertical sensitivity is, therefore, limited in the troposphere. Instruments with spectral windows in the UV include the

Total Ozone Mapping Spectrometer (TOMS) (Stolarski et al., 1991), the Global Ozone Monitoring Experiment (GOME) (Liu et al., 2006), GOME-2 (Cai et al., 2012), the Ozone Monitoring Instrument (OMI) (Kroon et al., 2011), the SCanning Imaging Absorption SpectroMeter for Atmospheric CHartographY (SCIAMACHY) (Eichmann et al., 2004), the Ozone Mapper Product Suite (OMPS) Nadir Mapper (NM) and Nadir Profiler (NP) (Flynn et al., 2014; Seftor et al., 2014), and the Sentinel-5P (S5P) TROPOspheric Monitoring Instrument (TROPOMI) (Zhao et al., 2021).

These instruments, and many others (e.g., Michelson Interferometer for Passive Atmospheric Sounding (MIPAS) (Fischer et al., 2008)) have yielded a multi-decadal record of ozone variation in the atmosphere. Future planned instruments include (in the TIR) IASI-Next Generation (NG; (Crevoisier et al., 2014)) and Geostationary and Extended Orbits (GeoXO; (NOAA, 2022)). In the UV, Sentinels 4 and 5 (Ingmann et al., 2012), Tropospheric emissions: Monitoring of pollution (TEMPO) (Zoogman et al., 2017) and the Geostationary Environmental Monitoring Spectrometer (GEMS) (Nicks et al., 2018) will

augment this record.

All of the above instruments rely on using either TIR or UV wavelengths, thus leaving a gap in the characterisation of the lower troposphere. This is problematic for understanding the impact of ozone in the boundary layer. A proposed solution to this



problem is to perform multi-spectral retrievals by combining TIR and UV measurements. In particular, this can be implemented using different instruments from multiple satellites. Multi-spectral satellite retrievals have been shown to improve sensitivity
to lower tropospheric ozone (Landgraf and Hasekamp, 2007; Luo et al., 2013; Cuesta et al., 2013, 2018; Fu et al., 2013, 2018; Natraj et al., 2011; Worden et al., 2007b) and also for other trace gases (e.g. methane) (Deeter et al., 2014; Fu et al., 2016; Worden et al., 2015). In addition, there is the opportunity to combine TIR and UV information from future instruments such as IASI-NG and Sentinel 4, and incorporate information from Chemical Reanalysis to further improve knowledge of tropospheric ozone (Colombi et al., 2021). The methods used in this paper will, therefore, be applicable to the next generation of satellite
instruments.

The aim of this paper is to build on the previous generation of joint spectral retrievals pioneered by TES and OMI and more recently AIRS and OMI measurements, by characterising and quantifying the performance of ozone profile retrievals using radiance measurements from CrIS and TROPOMI. The utility of joint CrIS-TROPOMI retrievals have been shown previously, initially through simulations (Fu et al., 2018), and more recently using operational data (Mettig et al., 2021, 2022).
However, this work utilises different portions of the spectral bands. CrIS and TROPOMI orbit in formation, typically viewing the same scene within 10 minutes, thus offering an excellent opportunity for joint retrievals. For retrievals, we use the MUlti-SpEctra, MUlti-SpEcies, MUlti-SEnsors (MUSES) retrieval algorithm, which is a core part of the TRopospheric Ozone and its Precursors from Earth System Sounding (TROPESS) pipeline (https://tes.jpl.nasa.gov/tropess/get-data/products/). TROPESS, which has considerable heritage from multi-spectral, multi-instrument retrieval algorithms for the UV and the TIR, produces
long term, Earth science data records with uncertainties and observation operators (Bowman et al., 2002; Kulawik et al., 2006; Worden et al., 2007b; Natraj et al., 2011; Luo et al., 2013; Fu et al., 2013, 2018; Kulawik et al., 2021).

TROPESS products have been extensively used for scientific analyses; for example, ozone retrievals from CrIS have been assimilated into reanalysis datasets to understand tropospheric ozone during the COVID lock-downs (Miyazaki et al., 2021). Carbon monoxide products from TROPESS/CrIS have been used to understand the impact of wildfires in Australia (Byrne
et al., 2021). CrIS-TROPOMI ozone products will join the TROPESS data record for use in scientific analysis.

This paper is structured as follows: The instruments used in this study are identified in Sect. 2. Section 3 details the MUSES algorithm and the characteristics of a subset of ozone retrievals from this algorithm. Section 4 presents some sample results from the joint retrievals, and Sect. 5 shows validation and cross-comparison studies between the MUSES retrieval results, and retrievals from third party algorithms. Finally, we analyse the results in Sect. 6 and infer conclusions about the performance of
MUSES with respect to the multispectral retrievals.

## 2 Instruments

### 2.1 Suomi NPP CrIS

CrIS is on the Suomi National Polar-orbiting Partnership (NPP) satellite in a near-polar, sun-synchronous, 828 km altitude orbit with a 13:30 ascending node crossing time and has been operational since October 28, 2011 (Han et al., 2013). Suomi
NPP includes several satellite instruments relevant to ozone profile retrievals, including both the OMPS-Nadir Mapper (NM)



**Table 1** Characteristics of the Suomi NPP CrIS bands.

| Band (Name) | Spectral range (cm$^{-1}$) | Spectral resolution (cm$^{-1}$) | Spatial resolution (diameter km) |
|---|---|---|---|
| **Band 1 (Long-Wave)** | 648.75-1096.25 | FSR/NSR:0.625, | 14 |
| **Band 2 (Mid-Wave)** | 1208.75-1751.25 | FSR:0.625, NSR:1.25 | 14 |
| **Band 3 (Short-Wave)** | 2153.75-2551.25 | FSR:0.625, NSR:2.5 | 14 |

and Nadir Profiler (NP), and the Visible Infrared Imaging Radiometer Suite (VIIRS). A second copy of CrIS is also aboard NOAA's Joint Polar Satellite System (JPSS) NOAA-20, launched in 2017; the CrIS instrument is also planned to be included on a series of follow-on satellites over the coming decade. However, CrIS on NOAA-20 and 21 will not fly in formation with TROPOMI. While the focus of this paper is the CrIS instrument on the Suomi NPP satellite, the techniques are applicable to

the other instruments as well. CrIS is a nadir viewing Fourier Transform Spectrometer (FTS) that measures TIR radiances in the three spectral bands in either Normal Spectral Resolution (NSR) or Full Spectral Resolution (FSR) identified in Table 1. It provides daily global measurements with a swath width of 2300 km, sampled at 30 cross-track positions, with each position consisting of a 3×3 array of pixels with about a 14 km diameter field of view. The wide spectral range and high spatial sampling allows CrIS to retrieve a range of atmospheric composition products, including additional trace gases other than ozone, such as

ammonia and carbon monoxide, as well as high resolution temperature and water vapor profiles, useful for numerical weather prediction (Smith and Barnet, 2020).

The extensive time period of data from Suomi NPP CrIS has led to significant work towards generating a long term record of numerous trace gases from the TROPESS product (Fu et al., 2018), and other projects such as the Community Long-term Infrared Microwave Combined Atmospheric Product System (CLIMCAPS) (Smith and Barnet, 2019, 2020), as well as single

field of view (FoV) products (Xiong et al., 2022). Note that up until February 2020 Suomi NPP CrIS was providing spectra in both NSR and FSR formats, after which NSR was discontinued. This means a shorter time record is available for joint CrIS-TROPOMI using NSR, therefore this paper focuses on FSR data, which are available from December 2014. However, both CrIS NSR and FSR data is available as a part of the TROPESS project.

In May 2021, Suomi NPP CrIS reported failures in the long-wave channels of the 'side 2' electronics suite (NOAA, 2021).

In response the instrument was switched to the 'side 1' electronics in order to retain use of the long-wave (band 1) channels (Iturbide-Sanchez et al., 2021). However, the mid-wave (band 2) channels of the side 1 electronics are non-functioning. Since ozone information in the TIR is mostly in the long-wave region, the loss of the mid-wave channels will not impact ozone retrievals to a significant degree. We provide evidence of the impact through an assessment presented in Appendix A, which identifies a minor loss of sensitivity in the retrieved profiles. However, the data presented in this study is from 2020; therefore,

this reported issue will not affect any of the presented results.

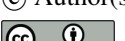



**Table 2** Characteristics of S5P/TROPOMI bands.

| Band (name) | Spectral range (nm) | Spectral resolution (nm) | Spatial resolution (km$^2$) |
|---|---|---|---|
| **Band 1 (UV1)** | 267-300 | 0.45-0.5 | 28x28.8 |
| **Band 2 (UV2)** | 300-332 | 0.45-0.5 | 5.6x3.6 |
| **Band 3 (UVIS)** | 305-400 | 0.45-0.65 | 5.6x3.6 |
| **Band 4 (VIS)** | 400-499 | 0.45-0.65 | 5.6x3.6 |
| **Band 5 (NIR1)** | 661-725 | 0.34-0.35 | 5.6x3.6 |
| **Band 6 (NIR2)** | 725-786 | 0.34-0.35 | 5.6x3.6 |
| **Band 7 (SWIR3a)** | 2300-2343 | 0.227 | 5.6x7.2 |
| **Band 8 (SWIR3b)** | 2343-2389 | 0.225 | 5.6x7.2 |

## 2.2 S5P/TROPOMI

The Sentinel-5P (S5P) satellite (https://sentinel.esa.int/web/sentinel/missions/sentinel-5p/satellite-description) was launched in October 2017 with the aim of providing global information on air quality and greenhouses gases (GHGs) (Veefkind et al., 2012). S5P is in a mid-afternoon low Earth orbit (13:30 ascending node crossing time) in a
tandem orbit with the Suomi NPP satellite. The TROPOMI instrument onboard S5P is an imaging spectrometer with a swath width of roughly 2600 km on the ground, providing data in eight separate wavebands. For each band, the swath width is split up into 'cross track' pixels, which form the individual instrument measurements of size indicated by the spatial resolution in Table 2. The spectral response for each band is characterised by the number of spectral pixels and the instrument line shape function (ILSF).
Bands 1, 2 and 3 (UV1, UV2 and UVIS) are sensitive to ozone, and profile retrievals are possible using windows within these bands, with **?** using UV1 & UV2 for profile retrievals and Zhao et al. (2021) using UVIS. UV1 & UV2 are located on the same detector, but are binned differently due to the SNR differences in the bands. UV1 & UV2 suffer from calibration issues (Ludewig et al., 2020; **?**), requiring significant effort to generate L1b spectra of sufficient quality for ozone retrievals. UVIS is not affected to the same degree (although some soft-calibration is still necessary (Zhao et al., 2021). The primary goal of
this paper is to ensure high quality retrievals from a spectral combination of CrIS and TROPOMI, rather than maximising the potential of TROPOMI alone. We therefore use TROPOMI UVIS, which is least affected by calibration errors. In particular, we use the 325-335 nm window in the Huggins band for total ozone column (TOC) retrievals (Garane et al., 2019). The approach described herein can readily accommodate a larger spectral range as TROPOMI calibration improves.

## 2.3 Validation and cross-comparison datasets

Datasets used for evaluation are split into three types. Cross-comparisons are made with independent ozone retrievals based upon a co-location of one day, with co-location criteria described in each subsection. Chemical reanalysis datasets, which provide global and consistent ozone, can be constrained by multiple satellite observations including precursor data, (e.g.,




NOx emissions). These cross-comparisons provide insight into how the retrievals are capturing chemical processes. Their data
fields use a more stringent co-location criteria owing to their higher data abundance. We further validate the retrievals with
ozonesondes. While ozonesondes are the most accurate (5%) for evaluation, they do not have the same sampling density as
the other datasets. In order to get a better statistical sampling, coincidence criteria is looser. The combination of these datasets
provides a clearer picture of the CrIS/TROPOMI retrieval performance.

### 2.3.1 Microwave Limb Sounder (MLS)

MLS is a microwave limb sounder aboard the Aura satellite launched in 2004, on a sun-synchronous orbit similar to S5P and
Suomi NPP, with an ascending node crossing time of 13:45 (Waters et al., 2006). The similar orbits of Aura, S5P and Suomi
NPP, ensure that the maximum distance between the closest MLS, TROPOMI and CrIS pixels is roughly 1000 km and 1.5
hours. MLS measures emissions using seven radiometers in a spectral range between 118 GHz and 2.5 THz. MLS has a spatial
sampling of $\sim$ 6 km across track and $\sim$ 200 km along track, and a vertical resolution ranging from 2.5 to 3.5 km starting in
the upper troposphere. The ozone retrievals between 9 and 75 km can be used for scientific analysis. MLS has been validated
extensively, (Froidevaux et al., 2008; Livesey et al., 2008), and is therefore a key dataset for cross-comparisons with TROPOMI
and CrIS in the stratosphere.

### 2.3.2 AIRS-OMI

The NASA A-Train satellites are in a near-polar sun-synchronous, $\sim$700 km altitude orbit with an ascending node crossing
time of $\sim$13:30 local time. In the A-train is the Aqua satellite, which includes the AIRS instrument. AIRS is a grating spec-
trometer that measures TIR emissions in the 650–2665 cm$^{-1}$ spectral range, similar to CrIS (Aumann et al., 2003). AIRS is
a cross-track scanning grating spectrometer that provides daily global coverage of ozone with a footprint of $\sim$13.5 km. OMI
is a nadir-viewing push broom ultraviolet–visible (UV-VIS) grating spectrometer on the AURA satellite that measures solar
back-scattered radiance. OMI measures in the 270–500 nm wavelength range (Levelt et al., 2006), covering the same ozone
absorption bands as TROPOMI. The ground pixel size of OMI at nadir is $\sim$13×24 km when using the 310–330 nm spectral
range. Since 2009, instrument issues have affected the quality of some OMI pixels limiting the latitudinal range (Levelt et al.,
2018).

Ozone profiles from joint AIRS-OMI L1B radiances are retrieved using the MUSES algorithm with spectral windows sim-
ilar to those employed for the CrIS-TROPOMI product generated here (Fu et al., 2018). The AIRS-OMI retrieval has been
extensively validated, and has been used as a key component for chemical re-analysis datasets (Miyazaki et al., 2020b, a).

### 2.3.3 TROPOMI Offline TOC product

The operational TROPOMI TOC product is split into two parts, near real time and offline (OFFL). We compare the MUSES
TROPOMI TOC retrievals against the OFFL product, which is based on the operational implementation of the GOME-type
Direct FITting (GODFIT) v4 algorithm (Garane et al., 2018). GODFIT is designed to retrieve TOC products from satellite





nadir-viewing instruments such as SCIAMACHY and GOME/GOME-2. GODFIT simulates radiances in the Huggins band
(325–335 nm) and fits to the observed radiances by varying a number of atmosphere parameters, including total ozone and
effective scene albedo (Garane et al., 2019). GODFIT and the MUSES TROPOMI algorithm (Sect.3.3) have a number of
differences including ancillary data and retrieval strategy that will lead to different ozone values. Most notably, GODFIT is a
scalar retrieval algorithm, i.e., the TOC is determined by multiplying the a priori ozone profile by a fixed value. MUSES is a
profile fitting algorithm.

### 2.3.4 Chemical reanalysis Cross-Comparisons

To improve the understanding of emission variability and the processes controlling atmospheric composition, several chemical
reanalysis products have been produced by integrating satellite measurements through chemical data assimilation (Sandu and
Chai, 2011). Chemical data assimilation is the process through which atmospheric models incorporate additional information
from observations, such as satellite observations, to generate an optimal representation of atmospheric composition. Based on
long-term chemical data assimilation, chemical reanalysis is a systematic approach to create a long-term data record consistent
with model processes and observations, which mitigates the limitations of current observing systems using models.

Two independent chemical reanalysis products are used for cross-comparison in this study. First, the Copernicus Atmo-
spheric Monitoring Service (CAMS) reanalysis consists of 3-dimensional time-varying atmospheric composition fields, in-
cluding ozone. The CAMS reanalysis was produced using data assimilation from the ECMWF Integrated Forecasting System
with 60 hybrid sigma/pressure (model) levels in the vertical, interpolated to 25 pressure levels with the top level at 0.1 hPa
(Flemming et al., 2017). The CAMS reanalysis infers only the atmospheric state, not emissions.

The second reanalysis dataset is the Tropospheric Chemistry Reanalysis Version 2 (TCR-2) obtained from the JPL Multi-
mOdel Multi-cOnstituent Chemical data assimilation (MOMO-Chem) system, which uses an ensemble Kalman filter (EnKF)
data assimilation technique (Miyazaki et al., 2020a, b) to simultaneously estimate concentrations and emissions of various
species. For the years 2005–2020 at 1.1° horizontal resolution assimilation is performed for ozone, CO, $NO_2$, $HNO_3$, and $SO_2$
using OMI, SCIAMACHY, GOME-2, TES, MLS, and MOPITT satellite measurements. The evaluation results demonstrate
the capability of the chemical reanalysis to improve understanding of the processes controlling variations in atmospheric
composition, including long-term changes in near-surface air quality and emissions (Miyazaki et al., 2021). For the 2020 year
comparisons used in this study, the tropospheric ozone in MOMO-Chem was constrained by measurements of precursors (OMI
$NO_2$ and MOPITT CO) and stratospheric concentrations (MLS ozone and $HNO_3$), whereas no tropospheric ozone data was
assimilated facilitating independent validation of the satellite tropospheric ozone products.

### 2.3.5 Ozonesondes

Ozonesondes are sensors that use a dilute solution of potassium iodide to produce a weak electrical current proportional to
the ozone concentration of the sampled air (Komhyr et al., 1995). These sensors are placed on balloons and provide in-situ
data from the surface to the stratosphere (about 35 km) with vertical resolution of 150m and accuracy of 5% (Witte et al.,
2017, 2018). The World Ozone and Ultraviolet radiation Data Centre (WOUDC) operates and freely publishes data from





these ozonesondes, producing multi-year datasets over global locations. These data are key in the validation of ozone profiles measured by satellite instruments (Thompson et al., 2017), and have previously been used to validate TROPESS ozone products (Worden et al., 2007b; Fu et al., 2018).

## 3 TROPESS/MUSES

### 3.1 Algorithm description

The MUSES algorithm is described in detail in other publications (Worden et al., 2007b; Fu et al., 2013; Luo et al., 2013; Fu et al., 2016, 2018), but is summarised here. MUSES is designed to be flexible such that multiple trace gas retrievals from various instrument types are possible. This algorithm has a flexible and generic Radiative Transfer Model (RTM) that covers the entire wavelength range from the UV to the TIR and a nonlinear retrieval algorithm based on the Optimal Estimation (OE) method (Rodgers, 2000). MUSES has a long heritage in retrieving atmospheric parameters from a number of different satellite missions and instruments, including AIRS, TES and OMI, as well as for performing joint spectral retrievals (Bowman et al., 2006; Kulawik et al., 2006; Fu et al., 2013; Luo et al., 2013; Fu et al., 2018; Worden et al., 2019; Kulawik et al., 2021).

MUSES optimally fits the simulated spectral radiance output from a RTM to observed spectral radiance measurements from satellite instruments in order to infer surface and atmospheric parameters. The OE algorithm at the heart of MUSES computes the best-estimate state vector $\hat{\mathbf{x}}$, which represents that atmospheric state and ancillary variables, by minimising the following cost function adapted for multiple instruments:

$$
J(\mathbf{x}) = \underbrace{\left[ [\mathbf{y}_{\text{Cr}} - \mathbf{F}(\mathbf{x}, \mathbf{b}_{\text{Cr}})]^T \mathbf{S}_{\epsilon-\text{Cr}}^{-1} [\mathbf{y}_{\text{Cr}} - \mathbf{F}(\mathbf{x}, \mathbf{b}_{\text{Cr}})] \right]}_{\text{CrIS}}
$$

$$
+ \underbrace{\left[ [\mathbf{y}_{\text{TO}} - \mathbf{F}(\mathbf{x}, \mathbf{b}_{\text{TO}})]^T \mathbf{S}_{\epsilon-\text{TO}}^{-1} [\mathbf{y}_{\text{TO}} - \mathbf{F}(\mathbf{x}, \mathbf{b}_{\text{TO}})] \right]}_{\text{TROPOMI}}
$$

$$
+ (\mathbf{x} - \mathbf{x}_a)^T \mathbf{S}_a^{-1} (\mathbf{x} - \mathbf{x}_a), \quad (1)
$$

Cr refers to CrIS and TO refers to TROPOMI. The difference between the observed radiance $\mathbf{y}$ and the simulated radiance $\mathbf{F}(\mathbf{x}, \mathbf{b})$ inversely weighted by the instrument error co-variance matrix $\mathbf{S}_\epsilon$ is represented by the first half of the equation. $\mathbf{F}(\bullet)$ represents the forward model, with $\mathbf{b}$ representing a vector of elements necessary for radiance simulation, but not actively retrieved. The instrument noise $\mathbf{S}_\epsilon$ is obtained from CrIS or TROPOMI L1b files. The difference between the retrieval vector $\mathbf{x}$ and the a priori state $\mathbf{x}_a$ inversely weighted by the a priori covariance matrix $\mathbf{S}_a$ is represented by the second term in Equation 1. Equation 1 is minimised through an iterative update of the state vector based upon the trust-region Levenberg–Marquardt



scheme (Bowman et al., 2006):

$$
\mathbf{x}_{i+1} = \mathbf{x}_i + \left( \underbrace{\mathbf{K}_{\mathrm{Cr}}^{T}\mathbf{S}_{\epsilon-\mathrm{Cr}}^{-1}\mathbf{K}_{\mathrm{Cr}}}_{\mathrm{CrIS}} + \underbrace{\mathbf{K}_{TO}^{T}\mathbf{S}_{\epsilon-TO}^{-1}\mathbf{K}_{TO}}_{\mathrm{TROPOMI}} + \mathbf{S}_a^{-1} + \lambda_i \mathbf{D}^2 \right)^{-1}
$$

$$
\times \left( \underbrace{\mathbf{K}_{\mathrm{Cr}i}^{T}\mathbf{S}_{\epsilon-\mathrm{Cr}}^{-1}[\mathbf{y}_{\mathrm{Cr}} - \mathbf{F}(\mathbf{x}_{\mathrm{Cr}i}, \mathbf{b}_{\mathrm{Cr}})]}_{\mathrm{CrIS}} + \underbrace{\mathbf{K}_{\mathrm{TO}i}^{T}\mathbf{S}_{\epsilon-\mathrm{TO}}^{-1}[\mathbf{y}_{\mathrm{TO}} - \mathbf{F}(\mathbf{x}_{\mathrm{TO}i}, \mathbf{b}_{\mathrm{TO}})]}_{\mathrm{TROPOMI}} \right)
$$

$$
+ \mathbf{S}^{-1}_{\mathrm{a}}(\mathbf{x}_i - \mathbf{x}_a), \quad (2)
$$

where $\lambda$ is the Levenberg-Marquardt parameter, chosen at each iteration step, multiplied by scaling matrix $\mathbf{D}$. This factor is important in controlling the convergence of the algorithm, with large $\lambda$ values prioritising the speed of the convergence, but making the steps more non-linear, while small values reduce the speed of the convergence, and is more linear in the iteration, similar to the conjugate gradient method. $\mathbf{K}$ represents the Jacobian matrix for the sensitivity of the radiances to the atmospheric state and is defined as $\mathbf{K} = \delta\mathbf{F}(\mathbf{x}, \mathbf{b})/\delta\mathbf{x}$. Eqs. 1 and 2 refer to the joint CrIS-TROPOMI retrieval, and therefore are a combination of the elements in Tables 4 and 6. For the CrIS-only and TROPOMI-only retrievals shown in Fig. 1, Eqs. 1 and 2 are modified to take into account only the relevant instruments.

## 3.2 CrIS retrieval configuration

The retrieval strategy for CrIS ozone retrievals is shown in Fig 1. The strategy consists of steps to refine atmospheric parameters prior to the retrieval of the ozone (Kulawik et al., 2006; Eldering et al., 2008; Worden et al., 2012).



**Figure 1.** MUSES ozone retrieval pipeline outline for CrIS-TROPOMI. The retrieval steps unique to CrIS and TROPOMI are indicated by the box colours. All steps are used in the CrIS-TROPOMI retrieval, with ozone retrievals and errors calculated in the CrIS and TROPOMI-only steps fed into the CrIS-TROPOMI retrieval. This feed-forward can be enabled or disabled depending on the requirements of the user. The spectral windows and interfering elements for CrIS are described in Table 3 and TROPOMI in Table 5.



The simulated radiance and Jacobians for the MUSES CrIS retrievals are calculated by the fast Optimal Spectral Sampling (OSS) RTM (Moncet et al., 2008, 2015) over a series of micro-windows defined in Table 3. To calculate the synthetic radiances, the OSS RTM is provided the viewing geometry and instrument response function from Version 2 of the CrIS L1b data (Revercomb and Strow, 2018). The RTM also requires the current atmospheric state, $\mathbf{x}$, along with the list of fitting parameters, a priori values and the a priori covariance as identified in Table 4, as well as ancillary data, $\mathbf{b}$ necessary for the simulation but not actively retrieved. The sources of the a priori data (indicated in Table 4) are described in other papers relating to MUSES (Kulawik et al., 2006; Fu et al., 2013, 2018; Worden et al., 2019; Kulawik et al., 2021), but are summarised here: the Model for OZone and Related chemical Tracers (MOZART)-3 and 4 (Brasseur et al., 1998; Park et al., 2004; Emmons et al., 2010) for the ozone profile and covariance. Water vapour, temperature profile and surface temperature data from the Goddard Earth Observing System Model, Version 5 (GEOS-5) (Suarez et al., 2008) and covariance from the National Center for Environmental Prediction (NCEP) reanalysis (Kalnay et al., 1996). The emissivity a priori are taken from the University of Wisconsin-Madison (UOW-M) Global infrared land surface emissivity database (Seemann et al., 2008). The a priori cloud properties come from an 'initial guess' refinement step using brightness temperature differences; here, as only one FoV per observation from the CrIS cross track position is used in the processing, the impact of clouds will be less pronounced. The spectroscopic parameters for the target gases and interfering gases are derived from the High Resolution TRANSmission (HITRAN) 2012 database (Rothman et al., 2013).





**Table 3** MUSES micro-windows used for CrIS pipeline ozone retrievals. Version 2.0 of the CrIS L1B data product, available on the NASA GES DISC (Revercomb and Strow, 2018), are used in this study.

| CrIS Band | Target | Window Start [cm$^{-1}$] | Window Stop [cm$^{-1}$] | Species |
|---|---|---|---|---|
| Long-Wave | O3 | 950.00 | 1031.25 | H2O, HDO, NH3, O3, CO2 |
| Long-Wave | O3 | 1043.125 | 1048.75 | H2O, HDO, NH3, O3, CO2 |
| Long-Wave | O3 | 1068.75 | 1088.75 | H2O, HDO, NH3, O3, CO2, CH4, CFC11, CFC12 |
| Long-Wave | O3 | 1094.375 | 1095.00 | H2O, CH3OH, HDO, NH3, O3, CO2, CH4, CFC11, CFC12 |
| Mid-Wave | O3 | 1211.25 | 1215.00 | H2O, HDO, O3, CO2, CH4, N2O |
| Mid-Wave | O3 | 1223.75 | 1227.50 | H2O, HDO, O3, CO2, CH4, N2O |
| Mid-Wave | O3 | 1258.75 | 1261.25 | H2O, HDO, O3, CO2, CH4, N2O |
| Mid-Wave | O3 | 1265.00 | 1267.50 | H2O, HDO, O3, CO2, CH4, N2O |
| Mid-Wave | O3 | 1268.75 | 1271.25 | H2O, HDO, O3, CO2, CH4, N2O |
| Mid-Wave | O3 | 1311.25 | 1317.50 | H2O, HDO, O3, CO2, CH4, N2O |





**Table 4** List of parameters in the MUSES/CrIS ozone retrieval state vector, including sources of a priori and covariance

| Fitting parameter | Number of parameters | A priori | A priori error |
|---|---|---|---|
| O$_3$ at each pressure level | 25 | MOZART-4 | MOZART-3 ~10-40% |
| H$_2$O at each pressure level | 16 | GEOS-5 | NCEP ~ 30% |
| Surface temperature | 1 | GEOS-5 | 0.5 K |
| Surface emissivity | 19 | UOW-M database | 0.006 |
| Cloud extinction | 10 | Initial Brightness Temperature Difference | 300% |
| Cloud top pressure | 1 | 500 mbar | 100% |

### 3.3 TROPOMI retrieval strategy

For the purposes of joint CrIS-TROPOMI retrievals, we use the Huggins band in UVIS. The retrieval pipeline for TROPOMI is identified by the green boxes in Fig 1, beginning with a cloud fraction guess, followed by the full retrieval.

The simulated radiance and Jacobians for the MUSES TROPOMI retrievals are calculated by the Vector LInearized Discrete Ordinate Radiative Transfer (VLIDORT) RTM (Spurr, 2006; Spurr et al., 2008). Synthetic radiances are generated by providing VLIDORT viewing geometry and instrument response information from version 1 of the TROPOMI L1b data. All radiances

are normalised in the retrieval process using version 1 TROPOMI L1b irradiance products (i.e., I$_{earthshine}$/I$_{sunshine}$, where I refers to irradiance). The TROPOMI retrieval state vector, along with the a priori source and covariance, is described in Table 6. The ozone profile (and non-retrieved elements such as temperature profiles) are the same as those identified for the CrIS state vector. Additional databases providing further information are as follows: the cloud fraction is determined from the 346.5-347.5 nm window; however, an initial guess is taken from the TROPOMI level 2 cloud product (Loyola et al., 2018). The a

priori for the first-order albedo term is taken from OMI climatology (Kleipool et al., 2008). Digital elevation data are provided by GMTED2010 (Danielson and Gesch, 2011). Serdyuchenko ozone cross sections, as used by the TROPOMI TOC product, are used in the calculation of the ozone optical depth (Serdyuchenko et al., 2014).

**Table 5** MUSES micro-windows used for TROPOMI pipeline ozone retrievals. Version 1.0 of the TROPOMI L1B data product, available on the NASA GES DISC, are used in this study

| TROPOMI Band | Target | Window Start [nm] | Window Stop [nm] | Species |
|---|---|---|---|---|
| UVIS | O3 | 325 | 335 | O3 |
| UVIS | Cloud Fraction | 346.5 | 347.5 | Cloud Fraction |





**Table 6** List of parameters in the MUSES/TROPOMI ozone retrieval state vector, including source of a priori and covariance

| Fitting parameter | Number of parameters | A priori | A priori error |
|---|---|---|---|
| $O_3$ at each pressure level | 25 | MOZART-4 | MOZART-3 $\sim$10-40% |
| Cloud Fraction | 1 | Derived from 346.5-347.5 nm | 0.05 |
| Ring Scaling | 1 | 1.9 | 1.0 |
| Irradiance shift | 1 | 0.0 | 0.02 nm |
| Radiance/Irradiance shift | 1 | 0.0 | 0.02 nm |
| Albedo (zero order) | 1 | OMI climatology | 0.05 |
| Albedo (first order) | 1 | 0 | 0.05 |
| Albedo (second order) | 1 | 0 | 0.01 |

### 3.4 MUSES CrIS-TROPOMI

The joint CrIS-TROPOMI retrieval integrate the CrIS and TROPOMI retrievals as identified in Fig. 1. The first step in the joint
CrIS-TROPOMI retrieval is pairing of the footprints. Our pairing method is as follows:

- Use only daytime CrIS soundings (SZA < 90°).

- Find all CrIS and TROPOMI (UVIS) pixels within a 20 minute time frame (where Suomi-NPP and S5P pass the same scene within $\sim$10 mins).

- From the current sounding subset, select all pairs that are within < 50 km distance, and 4) select the pair that has the
minimum distance.

As identified in Table 2, the TROPOMI bands have varying spatial resolution; therefore, for any future work, each band must be paired separately with the same CrIS footprint. Following the pairing, the CrIS-TROPOMI pipeline retrieval is performed as shown in Fig. 1, additional steps with respect to other target gases do occur in the pipeline, but are not highlighted here. In summary all CrIS retrieval steps are performed initially, beginning with improvements of ancillary information such as surface
temperature and cloud parameters. For handling clouds in CrIS retrievals, the MUSES algorithm picks each FoV from the 3×3 FoV structure of the CrIS Field of Regard (FoR). Relevant cloud properties (e.g., cloud top height and extinction) for the FoVs are retrieved and passed into the ozone retrieval. In contrast, other CrIS ozone algorithms (e.g., CLIMCAPS) use cloud clearing by aggregating cloud-free spectral channels across the 3×3 FoV structure (Susskind et al., 2003), meaning the MUSES CrIS retrieval has x9 higher spatial resolution. Ancillary information is provided by Operational Support Products
(OSPs) which include a priori information, covariances, ILSFs. Following the refinement of ancillary information, trace gas retrievals are undertaken, such as water vapour and methane. Information about water vapour, temperature, emissivity and cloud parameters are then passed onto the CrIS ozone retrieval (Table 4). After the CrIS ozone retrieval several other trace gases are retrieved (e.g. CO), but these do not affect the ozone retrieval. The next step is the TROPOMI ozone, which consists of two parts. Initially the cloud fraction is determined from the radiance spectra, followed by the ozone retrieval (Table 6).





The initial guess for the TROPOMI ozone profile can be either the a priori vector, or the CrIS retrieval. The next stage is
the CrIS-TROPOMI retrieval which is the final retrieval step, where the initial guess can either be the ozone a priori, or the
output from the TROPOMI retrieval. Following the retrievals, VMRs are obtained for each retrieval step (i.e., ozone for CrIS,
TROPOMI, CrIS-TROPOMI, and the other trace gases from CrIS). Retrieval errors are calculated at each step, and propagated
through the retrieval chain, such that the final CrIS-TROPOMI retrieval will incorporate errors from all of the previous steps.

The VMRs and errors finally are passed through quality control, after which the final trace gas products are generated. The
state vector for the CrIS-TROPOMI retrieval is a combination of the elements identified in Tables 4 and 6, and the spectral
windows are a combination of those indicated in Tables 3 and 5.

## 3.5 TROPOMI, CrIS and CrIS-TROPOMI Characterisation

### 3.5.1 Quality of fit

The spectral windows used in the joint CrIS-TROPOMI window shown in Fig. 2. For TROPOMI, the whole spectral range
is relevant to the ozone retrieval, from UV1 through to part of UVIS (UVIS extends to 400 nm, but ozone absorption is not
significant beyond 340 nm). However, these are not fully exploited because of calibration errors.

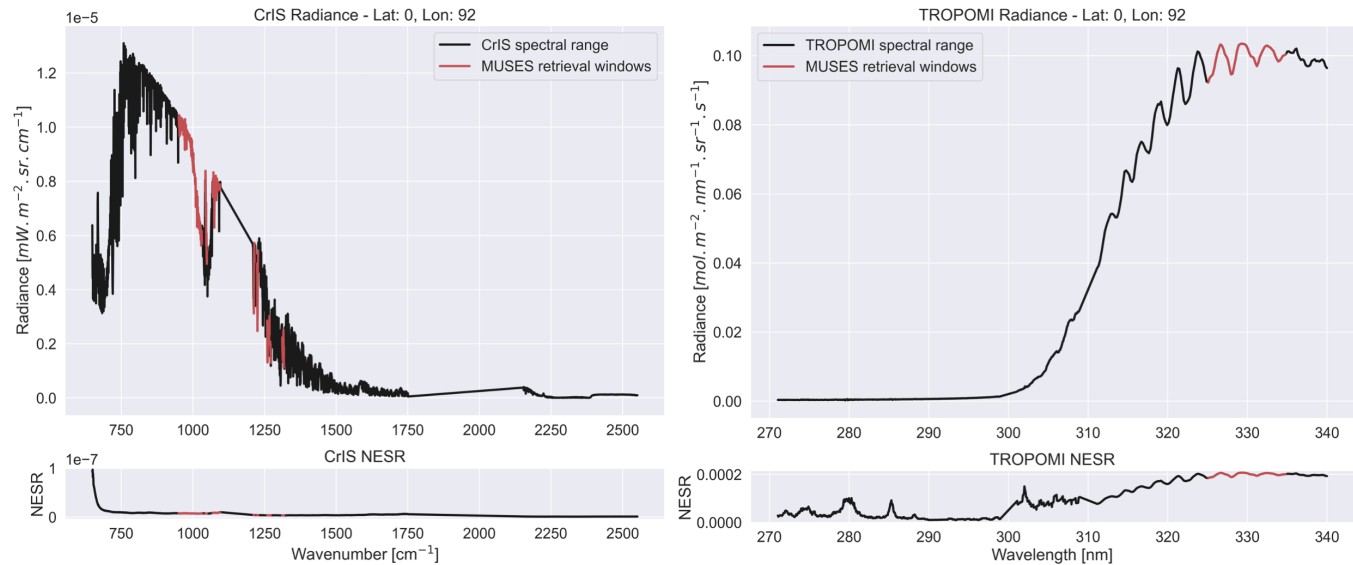

**Figure 2.** Example radiance from CrIS (left hand plot) and TROPOMI (right hand plot). The bold black line in the top plots indicates the
radiance from a specific case over the indicated lat/lon. The red areas indicate the retrieval windows used in this study. The bottom plot
indicates the Noise Equivalent Spectral Radiance (NESR) for this specific retrieval case.

The quality of the fit in these windows is expressed through multiple metrics, two of which are highlighted here. First,
we have the normalised residual radiance Root Mean Squared Error (RMSE), which is the RMSE between observed and
calculated radiance normalised by measurement error, indicating the impact of any large localised deviations. Second, we use





the normalised absolute residual radiance mean, which identifies the mean bias between the calculated and measured spectra, again normalised by the measurement error. Table 7 identifies examples of these metrics for the CrIS-TROPOMI, TROPOMI and CrIS cases, as well as the standard deviation over 10k cases on a day in August 2020.

**Table 7** Example CrIS, TROPOMI and CrIS-TROPOMI fit quality statistics based on 10k retrievals. All statistics are normalised by measurement error)

| Statistic (Normalised) | Mean | Standard Deviation |
|---|---|---|
| CrIS Residual radiance RMSE | 1.04 | 0.34 |
| CrIS Absolute residual radiance mean | -0.002 | 0.0340 |
| TROPOMI Residual radiance RMSE | 1.17 | 0.171 |
| TROPOMI Absolute residual radiance mean | 0.07 | 0.08 |
| CrIS-TROPOMI Residual radiance RMSE | 1.42 | 0.373 |
| CrIS-TROPOMI Absolute residual radiance mean | 0.055 | 0.019 |

The results shown in Table 7 suggest that TROPOMI retrievals have a larger absolute mean residual radiance, which is more
variable over the whole dataset, than either CrIS or CrIS-TROPOMI retrievals. The RMSE values indicate larger standard deviations in the CrIS-TROPOMI and CrIS cases, suggesting that CrIS are subject to larger fit errors, as opposed to TROPOMI, which is understandable given the wider retrieval windows. These quality of fit parameters form a part of the MUSES quality assurance procedure (described in 3.6).







**Figure 3.** Example normalised Jacobians on a logarithmic scale from TROPOMI (a) and CrIS (b). Jacobians are indicated for all MUSES pressure levels, normalised for each instrument. The square root of the diagonal elements of the measurement error matrix is used as the normalising element. Jacobians for the CrIS retrieval are split into separate panels, identifying the Jacobians for each spectral window (Table 3). The latitude and longitudes are indicated in the title of each panel, as well the Degrees of Freedom of Signal (DFS) values for the stratospheric (SP) and tropospheric profiles (TP).

.

The TROPOMI ozone Jacobians indicated in Fig. 3 a) show greater sensitivity at shorter wavelengths to stratospheric ozone.
Sensitivity decreases significantly with increasing wavelength and decreasing altitude. For the CrIS Jacobians shown in Fig. 3b) there is more ozone sensitivity in the lower stratosphere. In general, longer wavelengths have greater sensitivity in the lower troposphere whereas shorter wavelengths are more sensitive to upper tropospheric ozone. For the CrIS Jacobians, the majority of ozone sensitivity is found in the 9.6 micron band primarily between 100 and 10 hPa, but does extend to the lower troposphere





and the upper stratosphere. It is important to note that the Jacobians identified in Fig. 3 are for a single retrieval in the tropics,
where the tropospheric profile DFS are significant (1.2). Other examples (in high latitudes) not highlighted in this paper show
less sensitivity in the lower troposphere in the 950-1030 cm$^{-1}$ window. The ability of CrIS-TROPOMI and CrIS to resolve
information in the lower troposphere is further highlighted by the DFS presented in Fig. 5.

### 3.5.2   Information content

For moderately non-linear problems, the OE method allows for the characterisation of the relationship between the truth and
the retrieved state vector as follows Rodgers (2000)

$$\hat{\mathbf{x}} = \mathbf{x}_a + \mathbf{A}[\mathbf{x}_{true} - \mathbf{x}_a] + \mathbf{G}\epsilon + \delta_{cs}, \tag{3}$$

where $\mathbf{A}$ is the Averaging Kernel (AK), which indicates the sensitivity of the retrieved solution to the 'truth', i.e. the sensitivity
of the retrievals to changes in trace gas concentration in the atmosphere. $\delta_{cs}$ is the cross state (CS) error, calculated as $\delta_{cs} = \mathbf{A}_{CS}[\mathbf{x}_{CS} - \mathbf{x}_{CS_{a\,priori}}]$ which represents the impact of retrieving multiple species simultaneously (Fu et al., 2018). $\mathbf{x}_{true}$ is the
true state vector, $\mathbf{G}$ is the Gain matrix Eq 4:

$$\mathbf{G} = \left(\mathbf{K}_{Cr}^T \mathbf{S}_{\epsilon-Cr}^{-1} \mathbf{K}_{Cr} + \mathbf{K}_{TO}^T \mathbf{S}_{\epsilon-TO}^{-1} \mathbf{K}_{TO} + \mathbf{S}_a^{-1}\right)^{-1} \times \left[\mathbf{K}_{Cr}^T \mathbf{S}_{\epsilon-Cr}^{-1} + \mathbf{K}_{TO}^T \mathbf{S}_{\epsilon-TO}^{-1}\right], \tag{4}$$

$\epsilon$ is the spectral noise of the satellite instrument. The use of the OE method in MUSES allows for the generation of the AK and
error matrices for each retrieval, which are defined as

$$\mathbf{A} = \mathbf{G}\mathbf{K} = \frac{\partial \hat{\mathbf{x}}}{\partial \mathbf{x}_{true}}, \tag{5}$$

and the total covariance error

$$\mathbf{S} = \underbrace{(\mathbf{I} - \mathbf{A})\mathbf{S}_a(\mathbf{I} - \mathbf{A}^T)}_{\text{smoothing error}} + \underbrace{\underbrace{\mathbf{G}\mathbf{S}_\epsilon\mathbf{G}^T}_{\text{measurement error}} + \underbrace{\mathbf{A}_{cs}\mathbf{S}_{cs}\mathbf{A}_{cs}^T}_{\text{cross-state error}}}_{\text{satellite instrument observation error}}, \tag{6}$$

from the AK, we can determine the number of independent pieces of information in the retrieval, known as the degrees of
freedom for signal (DFS), defined as DFS = trace($\mathbf{A}$). DFS values larger than 1 indicate the capability to determine some
profile information. We define DFS values in discrete portions of the atmosphere. The tropospheric profile covers from the
surface pressure to the tropopause (as defined by GEOS-5). The lower tropospheric profile is from the surface pressure to 500
hPa and the upper tropospheric profile is from 500 hPa to the tropopause. The stratospheric profile is from the tropopause to
the top of the atmosphere.





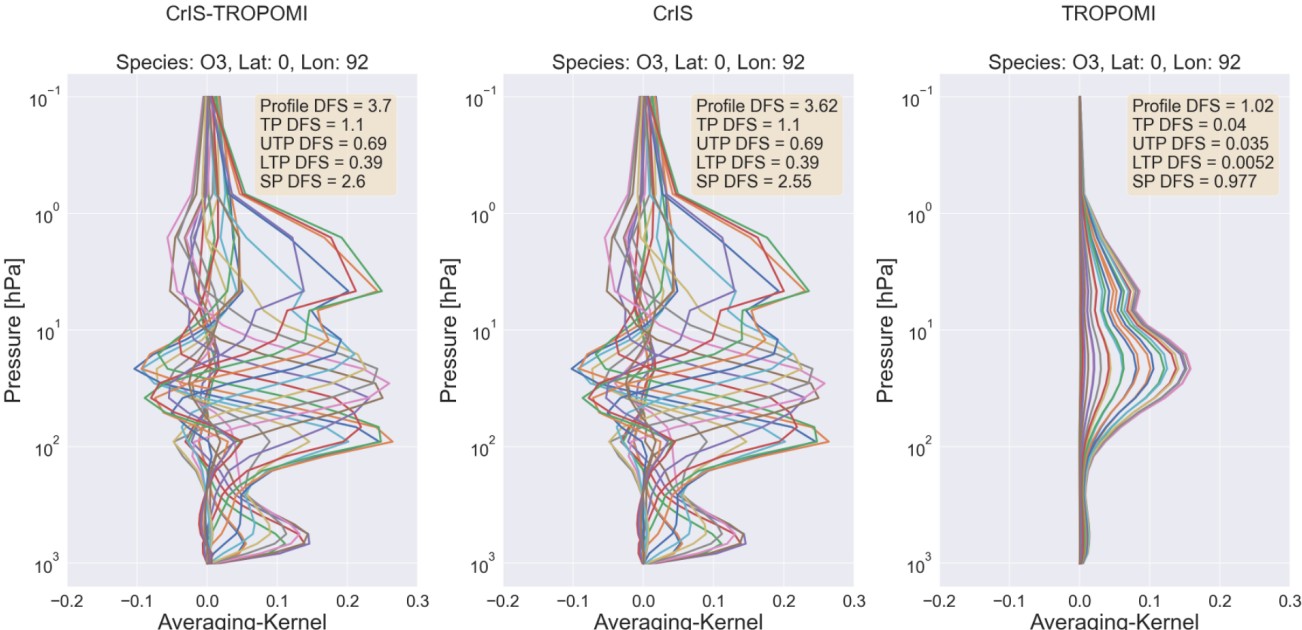

**Figure 4.** Example AKs from MUSES retrievals in August 2020. The left hand plot indicates the AKs for CrIS-TROPOMI, the middle for CrIS and the right-hand plot for TROPOMI. The DFS values for these AKs are indicated in the labels for each plot. Profile DFS shows the DFS for the whole retrieved profile, TP for the tropospheric profile, UTP for the upper tropospheric profile, LTP for the lower tropospheric profile and SP for the stratrospheric profile. The different colours represent the AK for each pressure level.

In Fig. 4 examples of AKs for each instrument on August 12 are shown. The TROPOMI AK (right panel) indicates 1 DFS for the whole profile as expected from the Huggins band, with almost all of the information confined to the stratosphere. The CrIS
AK indicates 1 DFS in the troposphere and 2.5 in the stratosphere, with the total profile generating about 3.6 DFS. Referring to Fig. 5, at similar latitudes total DFS values can be up to 4.5, indicating the shown example is a relative outlier. For comparison purposes Zhao et al. (2021) indicates 1.5-2 DFS over the whole profile for TROPOMI UVIS, and **?** calculate 6.5 for the whole profile in UV1 & UV2, with 1.5 for the troposphere. The implication is that TROPOMI and CrIS have similar performance in the troposphere for UV1 & UV2. This comparable performance can change however, given the performance of CrIS is
dependent on thermal contrast. The joint profile qualitatively retains the characteristics of the CrIS AK, while quantitatively, we note a minor increase in the profile and stratospheric DFS values, but no increase in the troposphere. In comparison, **?** show a CrIS-TROPOMI retrieval based on UV1 & UV2 of TROPOMI; the joint retrieval AK is dominated by TROPOMI in this case with the CrIS retrieval adding about 1 DFS to the TROPOMI AK for the case shown.

Comparisons of the AKs in Fig. 4 with the Jacobians in Fig. 3 provide some explanation for the indicated sensitivities. For
TROPOMI, the Jacobians indicate sensitivity for all wavelengths only at a small number of pressure levels in the stratosphere. Therefore, the AK peaks at one location in the stratosphere. CrIS Jacobians, in contrast, show sensitivity at multiple pressure levels due to the many spectral windows used for the CrIS retrieval. This is indicated by the multiple peaks in the AKs.



**Figure 5.** (a) DFS by latitude over the globe on a day in August 2020 for each of the instruments under investigation in this study, as indicated in the subplot titles. DFS are identified for the whole retrieval profile, as well as sub-columns as shown in the legend. In this figure, the upper troposphere refers to the region from 500 hPa to the tropopause, and lower troposphere is the region from the surface to 500 hPa. (b) Spread of profile DFS over the globe for a day in August 2020, with the order of subplots the same as those in (a). Two colour bars are shown, due to the large differences in DFS between CrIS-TROPOMI/CrIS and TROPOMI.

The global spread of DFS values is investigated in Fig. 5. The latitudinal spread of DFS values shown in row a) of Fig. 5 generally indicates similar DFS values for each sub-column for CrIS and CrIS-TROPOMI, as predicted by Fig 4. For both

CrIS and CrIS-TROPOMI, we note that the profile and tropospheric column DFS values tend to be at a maximum in the region surrounding the equator, while the DFS values in the stratosphere remain largely consistent over all latitudes. For TROPOMI, as suggested by Fig 4, DFS values are small in the troposphere over all latitudes, while those in the whole profile/stratosphere remain largely constant over the globe. Considering the global spread of profile DFS values for each instrument, as shown by





row b) of Fig. 5, CrIS and CrIS-TROPOMI have similar patterns, with the region surrounding the equator showing the largest
values.

Note that according to the histograms in Fig. C1 b) and c) CrIS-TROPOMI generally shows higher DFS values than CrIS
alone in both the troposphere and the stratosphere, although as indicated by Fig. 4, the differences can be subtle. For example,
focusing on the Atlantic Ocean, there are regions with clearly improved DFS values from CrIS-TROPOMI, as opposed to
CrIS. There are numerous cases for CrIS and CrIS-TROPOMI where DFS values of ∼2 are achieved in the whole troposphere
(Fig. C1). This suggests that CrIS-TROPOMI and CrIS are highly useful instruments for tropospheric ozone estimation. For
the TROPOMI retrievals, a constant DFS value of roughly 1 is apparent all over the globe, indicating that local conditions have
limited impact on the TROPOMI retrieval.

### 3.5.3 Uncertainties

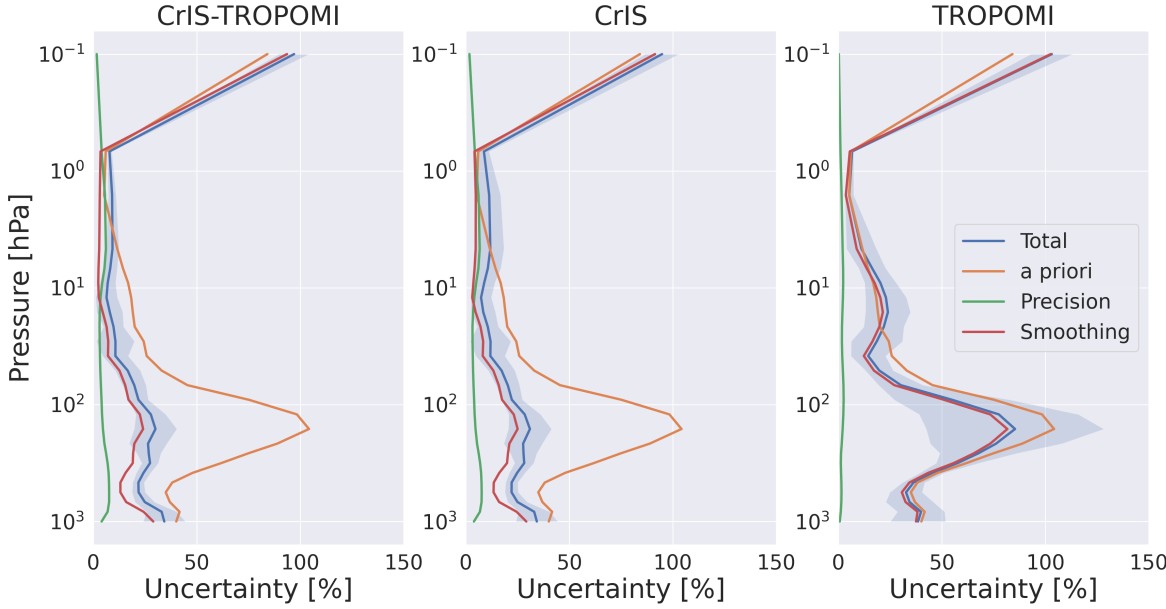

**Figure 6.** Retrieval uncertainty for CrIS-TROPOMI (left panel), CrIS (middle panel) and TROPOMI (right panel). Solid lines indicate the
mean uncertainties on a day in August 2020, while the shaded region indicates the standard deviation of the total error. The blue line indicates
the total error, the orange line shows the a priori error, the green line the measurement or precision error and the red line the smoothing error.

For CrIS, total uncertainty (Eq. 6) is a maximum at the surface, reducing in magnitude until the upper stratosphere (except for
a local reduction in magnitude in the lower troposphere). We note that the precision errors form a small component of the total
error in the troposphere and lower stratosphere; the error budget is largely dominated by the smoothing error. Comparisons
of the total uncertainty with the a priori uncertainty shows a general reduction in the uncertainty, with the most reduction at





the tropopause. Given that the majority of the DFS are contained within the stratosphere for CrIS (Fig. 5), this is the expected result. The variability of the total uncertainty is largely consistent throughout the atmosphere, except in the lower and upper
stratosphere where the magnitude is lower. The uncertainties shown for TROPOMI are notably larger than those shown for CrIS, with the total uncertainty indicating a minor reduction of the a priori uncertainty. This is the expected result, given that only 1 DFS is available from the TROPOMI retrieval alone. The standard deviation of the upper tropospheric retrieval is significantly larger than that of CrIS. As Fig. 4 indicates, there is very little information content for TROPOMI retrievals outside of the stratosphere, which is expected given the chosen spectral window. For the CrIS-TROPOMI retrieval, the uncertainty
profile is similar to that of CrIS. The key difference is that the variability of the total uncertainty is smaller than that of CrIS, suggesting that the inclusion of the TROPOMI radiances reduces the uncertainty of the CrIS retrievals.

### 3.6 Quality Assessment

Following the completion of the retrieval, the MUSES algorithm undertakes an assessment of the quality of the retrieval using a number of diagnostics (Kulawik et al., 2021). The primary quality flags are based upon the residual radiance between the
calculated and observed spectra, as well as on identifying cloud contamination, based on cloud optical depth for TIR and cloud fraction for UV. The quality flags for the CrIS-TROPOMI, TROPOMI and CrIS MUSES ozone retrievals are based on a statistical analysis of small clusters of retrievals. For MUSES CrIS, this analysis made use of the significant timeline of CrIS. For MUSES TROPOMI, only subsets of retrievals have been performed; therefore, the flags are based on MUSES OMI retrievals.
For a typical day (in August 2020) consisting of roughly 40k individual retrievals, the CrIS-TROPOMI retrievals have a pass rate of 39%; the corresponding numbers for CrIS and TROPOMI are 70% and 38%, respectively. The CrIS-TROPOMI and TROPOMI failures are by and large due to the presence of clouds in the sounding. The CrIS failures are primarily due to the residual radiance having too large a magnitude.

### 3.7 Observation Operator

In order to directly compare the MUSES retrievals with chemical reanalysis datasets and ozonesondes, we must account for the vertical sensitivity of the MUSES retrievals. The averaging kernel and a priori can be used to construct the observation operator (Jones et al., 2003; Worden et al., 2007a):

$$\mathbf{H}(\cdot) = \mathbf{x}_a + \mathbf{A}(\mathbf{M}(\cdot) - \mathbf{x}_a), \tag{7}$$

where $\mathbf{H}(\cdot)$ is the forward transport model relating the chemical reanalysis or ozonesonde profile to the retrieval, $\mathbf{M}(\cdot)$ is the
chemical reanalysis or ozonesonde profile at the location of the retrieval, and all other values are as previously defined.



# 4   Retrievals of CrIS-TROPOMI

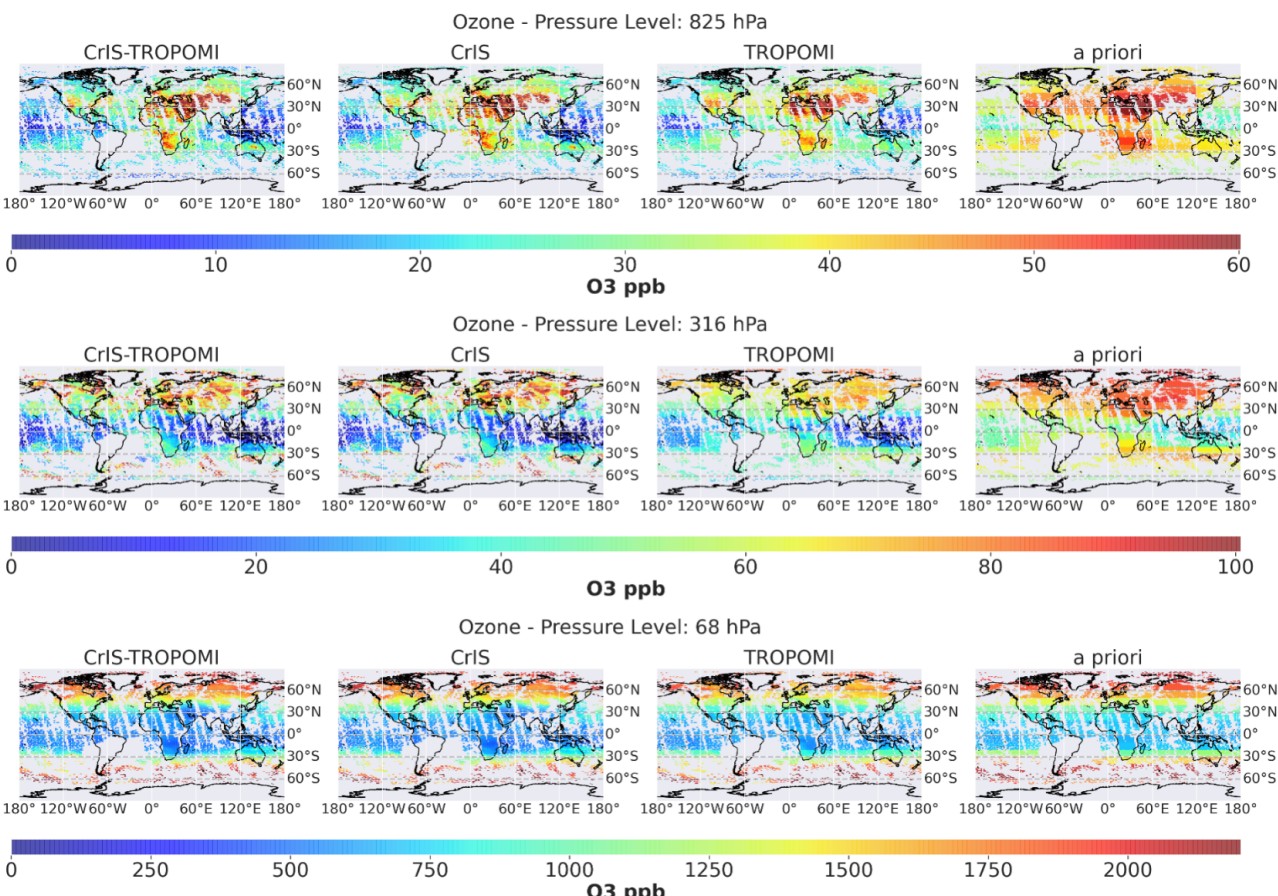

**Figure 7.** Global distributions of ozone on August 12 2020 in ppb, comprising of ~10k quality retrievals. The results shown in the left hand column are from the CrIS-TROPOMI retrievals, followed by CrIS, TROPOMI and the a priori in the right hand column. Each row indicates the ozone concentrations from an example pressure level; the first row is 825 hPa, the second row is 316 hPa, and the final row is 68 hPa. These pressure levels were chosen to represent a wide range of atmospheric conditions (lower troposphere, upper troposphere and stratosphere), as well as to coincide with peaks in the CrIS-TROPOMI AK.





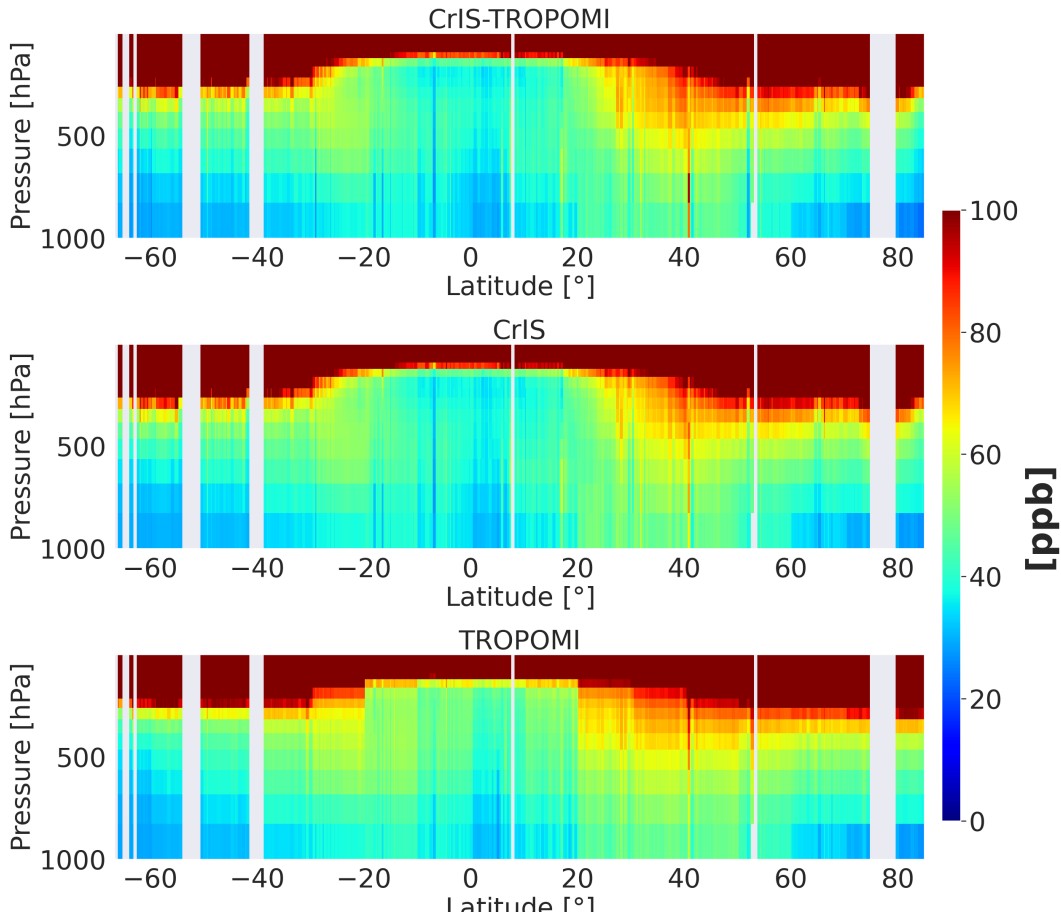

**Figure 8.** Vertical distribution of ozone binned into latitudinal bins, based on the data shown in Fig. 7. The titles of each subplot indicate the instrument.

Side-by-side comparisons of retrievals from CrIS-TROPOMI, CrIS and TROPOMI, and the a priori on August 12 are shown in Fig. 7. The top and middle rows indicate ozone retrievals at 825 hPa and 316 hPa (lower and upper troposphere), respectively. The troposphere is a complex region to characterise, with the interplay of chemistry and dynamics dictating the geographic and

vertical structure of ozone. The retrievals from CrIS-TROPOMI/CrIS/TROPOMI show a number of regions with significant ozone enhancements. For example, all cases show enhanced ozone in southern Africa, with some transport towards Australia. Southern Africa is dry in August and significant biomass burning occurs. Fires generate the precursors necessary to form ozone, thus yielding significant ozone over biomass burning (Sinha et al., 2004). Comparisons with the a priori show that the retrievals suggest much more localised extreme ozone enhancements, as opposed to broad low level enhancements. Other regions of

enhancement highlighted (especially by CrIS-TROPOMI and CrIS) include central and east Asia. In the summertime Asia experiences a monsoon, which has a significant impact on the climate of north Africa, the middle East and Central Asia though complex interactions described elsewhere (Gettelman et al., 2004; Liu et al., 2009). The Asian monsoon has been found to



enhance tropospheric ozone concentrations (Worden et al., 2009), as can be seen in the spatial maps, as well as the profile plots (Fig. 8) that indicate significant enhancements in the troposphere between 30° and 50°. The ozone distribution at 68 hPa
(stratosphere) shows the characteristics of the Brewer-Dobson circulation, with higher ozone concentrations at the poles, and lower magnitudes in the tropics. Both the 316 and 68 hPa pressure levels identify an interesting significant ozone enhancement (roughly double the surrounding regions) in the southern Indian ocean (60°S). It has been suggested that this enhancement is due to the geographical distribution of outflow from tropical convection in the upper troposphere and lower stratosphere (UTLS), which is concentrated over the southern Indian Ocean (Hitchman and Rogal, 2010). At these pressure levels, in
general, the a priori and the retrievals are similar, with the key difference that the a priori does not suggest the existence of significant enhancements in the southern Indian Ocean.

The TROPOMI results broadly capture the identified regions of enhancement at the specified pressure levels. However, they tend to provide underestimates with respect to CrIS and CrIS-TROPOMI as a consequence of the lower DFS values in the TROPOMI retrievals, which average between 0.9 and 1 in the stratosphere, and 0 and 0.1 in the troposphere. Figs 4 and 5 show
that the CrIS-TROPOMI retrievals are dominated by CrIS because of the limited well-calibrated UV bands in TROPOMI. Considering the tropospheric column results shown in the middle row, the TROPOMI results match those of the a priori, which is expected due to the limited information content (Fig. 5).

It is important to note that as a TIR instrument, CrIS is capable of retrievals at night, which TROPOMI, reliant on solar backscattered reflectances is unable to achieve. Figure 9 contrasts the night-time measurements of CrIS with those in the
daytime.



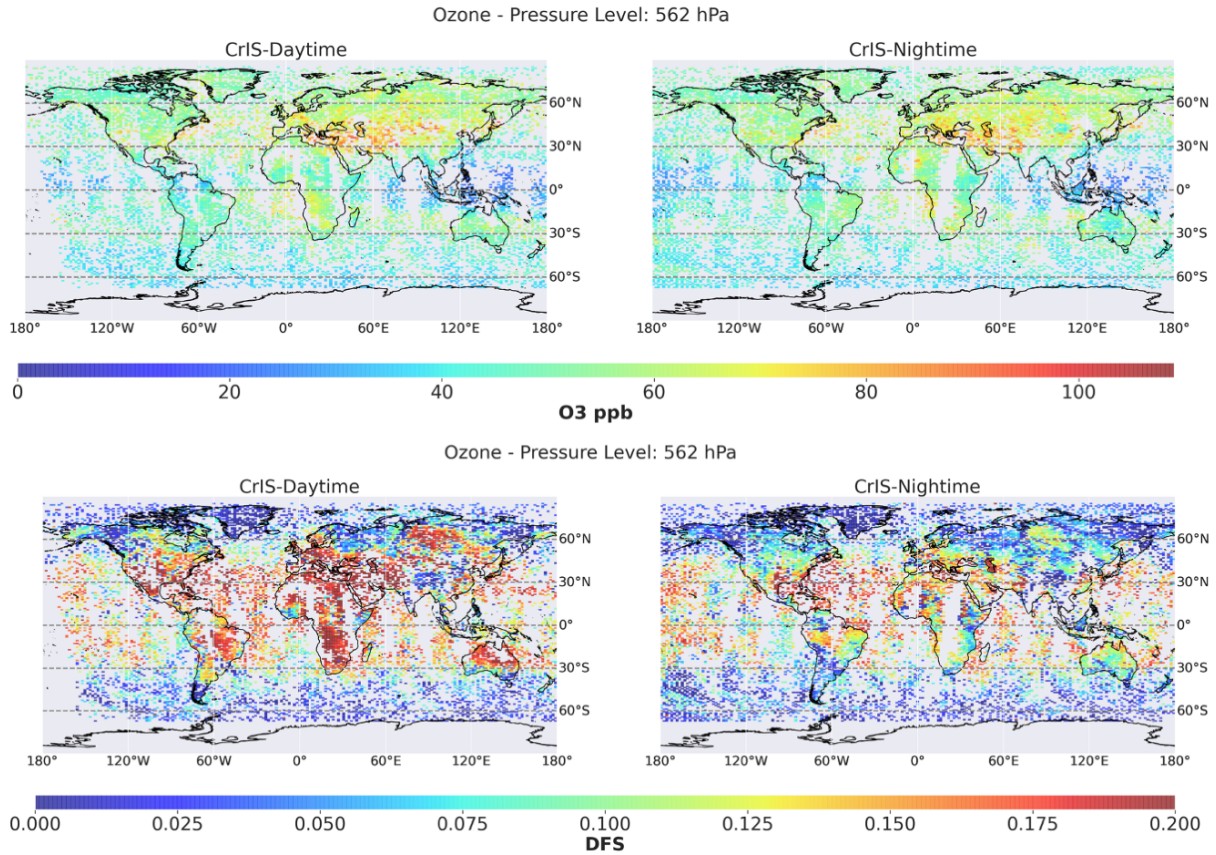

**Figure 9.** Top row, global distributions of ozone on August 12 2020 in ppb at 562 hPa. The results shown in the left-hand plot are daytime-only CrIS retrievals, and the right-hand plot are nighttime-only CrIS retrievals. Bottom row shows the DFS values for daytime and nighttime.

The day-time and night-time measurements are spatially consistent with some differences. For example, the daytime retrievals have a significant gap between 160°W and 180°W, while the nighttime retrievals are not available along the spine of Africa. Further, differences between the retrievals can be seen in some locations. For example, in Mongolia, there is at least a 50% drop in ozone concentration between the day-time and night-time cases. Some of these differences can be explained through the differences in DFS between day-time and night-time retrievals. In general we found day-time and night-time retrievals yielded similar DFS values in the sub-Arctic and Arctic regions. In the mid-latitudes and tropics, however, daytime retrievals had $\sim \times 2$ greater magnitude, implying that the results should be assessed differently.

Night-time ozone measurements are currently not a topic of significant study and of potential value (Tweedy et al., 2013). As with other similar plots shown in this paper, day-time/night-time retrievals are available at multiple pressure levels; the results shown in Fig. 9 are presented as an exemplar case.

Note that the consistent gap in retrievals over south America in Fig. 7 is due to the presence of the south Atlantic anomaly, where cosmic rays cause additional noise on the detectors, which in turn generate poor quality retrievals. This anomaly is not





apparent in the CrIS-only retrievals shown in Fig. 9. This implies that the CrIS hardware is more robust, or the TIR detectors are less sensitive to cosmic rays than the UV detectors of TROPOMI.

## 5    Validation and intercomparison

All results focus on August 12th 2020, for an in-depth global analysis, except for ozonesonde validation which uses data over the course of a year. Cross comparisons for additional days are shown in the supplementary material.

### 5.1    MLS

For cross-comparisons of CrIS-TROPOMI, CrIS and TROPOMI with MLS, the footprints are matched based on a distance of <1° and a time frame of <2 hours. This co-location criterion (along with AIRS-OMI) is the least stringent of all the comparisons shown in this study, due to the reduced coincidences from different satellite orbits and the relative low variability in stratospheric ozone. MLS profiles are interpolated to the MUSES pressure grid and converted to sub-columns. Only the stratospheric sub-columns are compared as the MLS data providers do not recommend the use of MLS data below 200 hPa for scientific interpretation.







**Figure 10.** Cross-comparisons of MLS with CrIS, TROPOMI and CrIS-TROPOMI on August 12 2020 for the stratospheric profile. The three subplots highlighted by a) show the linear relationship between the matched stratospheric columns of MLS and, moving top to bottom, CrIS-TROPOMI, CrIS, TROPOMI and the MUSES a priori. For the CrIS plot, the black dots represent daytime retrievals, and red dots show nighttime retrievals. The linear fit statistics are indicated in the plot, showing in order the linear slope, intercept, coefficient of determination and the number of matched footprints. The second section of subplots indicated by b) shows the percentage difference between the instrument retrievals and MLS in the stratospheric column on a global grid. Moving from left to right, CrIS-TROPOMI, CrIS and TROPOMI data are shown, with the plot on the far right showing the MLS ozone stratospheric column for reference. The third section of subplots shown by c) indicates the difference between the retrieved profiles from (moving top to bottom) CrIS-TROPOMI, CrIS, TROPOMI and the MUSES a priori, compared with MLS. All values are based on the co-located points shown in a) and b) binned into one longitudinal bin, shown between 0 and 200 hPa. The black lines indicates the tropopause.

The cross-comparison results shown in Fig. 10 indicate good agreement between all instrument retrievals and MLS. Indeed, the scatter plots shown in Fig 10(a) indicate that the MLS vs CrIS-TROPOMI comparisons show especially high levels of





linearity (slope of 1.03) and correlations ($R^2$ of 0.952). With respect to CrIS-only retrievals, again good agreement is shown, but to a lesser degree than with respect to CrIS-TROPOMI (slope of 0.921 and $R^2$ of 0.926). For TROPOMI alone, while strong correlation ($R^2$ of 0.924) and linearity is observed (slope of 0.898), a much larger bias is also present (intercept of 30.6 DU),

implying a poorer agreement when compared to biases seen in CrIS-TROPOMI and CrIS retrievals. All instrument retrievals show improved correlation over the MUSES a priori. Considering the plots shown in Fig 10(b), the differences between MLS and CrIS-TROPOMI show a general good level of agreement (between -5 and +5%). However, there are a number of locations with larger disagreements; for example, poleward of 60°S, 10% differences are observed. For CrIS, in general a positive bias is observed across the globe, which is supported by the results from the scatter plot. Again, larger disagreements are observed

below 60°S, in identical positions to those shown for the CrIS-TROPOMI comparison. For TROPOMI, larger negative biases are apparent across the globe; however, the largest biases occur in the same locations as for CrIS-TROPOMI and CrIS. However, for all three datasets, there is no obvious spatial bias, except for below 60°S. Referring to the MLS stratospheric column in Fig 10(b), we see notable anomalies in the southern hemisphere that are also captured and well represented by the MUSES retrievals (Fig 7), according to the spatial comparison points. The reference MLS plot indicates both ascending and descending

node retrievals, but only descending node retrievals are compared with the MUSES data. The plots shown in Fig 10(c) indicate the differences between the datasets and MLS at each pressure level by latitude. For CrIS-TROPOMI there are mixed results with significant differences between -40 and +40% around the tropopause, which is highest pressure MLS data is usable. In the tropics, good agreement is achieved (typically in the region of 0-10%) up to 10 hPa, with greater disagreement found from the mid-latitudes up to the poles. This agreement is to be expected since the major peaks of the CrIS-TROPOMI AK

are found at these pressure levels. Larger disagreement is apparent above 10 hPa, with differences up to 40% shown outside of the tropics. This is expected as the CrIS-TROPOMI sensitivity decreases above this pressure level, while MLS retains high sensitivity. Comparisons with CrIS-only retrievals show some similar characteristics to CrIS-TROPOMI. For example, good agreement up to the 10 hPa pressure level, typically 0-10% difference, with larger disagreement towards mid-latitudes and the poles. However, above 10 hPa, while CrIS-TROPOMI shows mostly an under estimation, CrIS-only shows an over estimation

with respect to MLS. Differences of between 10-40% are seen, although some examples of negative biases are also apparent, especially towards the south pole.

## 5.2  AIRS-OMI

For the comparison with MUSES AIRS-OMI the CrIS-TROPOMI/CrIS and TROPOMI retrievals, as with MLS, are matched based on a distance of <1° and a time frame of <2 hours. For this cross-comparison we investigate two sub-columns, the

tropospheric sub-column, Fig. 11) and the stratospheric sub-column (Fig. B2).






**Figure 11.** Cross-comparisons of MUSES AIRS-OMI with CrIS, TROPOMI and CrIS-TROPOMI on August 12 2020 for the tropospheric sub-column. The structure of the figure is as shown in Fig. 10.

Starting with the scatter plots shown in Fig. 11(a), reasonable agreement is found between CrIS-TROPOMI/CrIS and AIRS-OMI in the tropospheric column, while AIRS-OMI vs TROPOMI-only indicates poorer agreement. The results from the comparison against CrIS-only retrievals are similar to those of CrIS-TROPOMI, with CrIS-only indicating a very minor closer agreement (slope of 0.793 and R$^2$ of 0.72 vs a slope of 0.793 and R$^2$ of 0.719). TROPOMI, shows poorer linearity and corre-

lation as a consequence of the UVIS band. These scatter comparisons are shown spatially in the plots in Fig. 11(b). In general CrIS-TROPOMI, CrIS-only and TROPOMI-only show between ±20% differences across the globe, with a number of common regions showing large disagreements. For example, west Africa shows differences >40% while. The vertical profile differences between CrIS-TROPOMI/CrIS/TROPOMI and AIRS-OMI are shown in Fig. 11(c). Both CrIS-TROPOMI and CrIS have similar differences between -10 and 10% in the mid-latitudes through the troposphere. However, differences exceed 40% in the





tropics. The differences in the TROPOMI profile are more significant, with up to 40% at near the tropopause in the tropics and mid-latitudes.

Stratospheric assessment of AIRS-OMI is shown in Fig. B2. Fig. B2(a) indicate excellent agreement in most cases. This is particularly true for CrIS-TROPOMI, with a linear slope of 0.997 and an $R^2$ of 0.89. While both CrIS-only and TROPOMI-only indicate similar (if lower in magnitude) linear correlation values, both indicate larger biases (intercepts $>\sim \times 3$ CrIS-

TROPOMI), implying that both CrIS and TROPOMI overestimate in the stratosphere with respect to AIRS-OMI. These results highlight the utility of the CrIS-TROPOMI retrievals in the stratosphere. The global distribution of the differences in the stratospheric profile indicated by the plots shown in Fig. B2(b) show global biases for each instrument, negative for CrIS-TROPOMI and TROPOMI, and positive for CrIS. There is also some indication that larger biases occur at regions with significant differences from the regional mean, e.g., the southern Indian Ocean. The differences in the stratospheric profile, highlighted by

the subplots in Fig. B2(c), are more pronounced than those in the tropospheric profile. CrIS-TROPOMI results show between 20 and 40% difference between 10 and 1 hPa across a wide range of latitudes, dropping to <10% above this point. CrIS, in contrast, indicates a largely uniform ∼10% difference above the 60 hPa level, with a number of notable exceptions, e.g., 19°. TROPOMI shows a different pattern, showing large (at least 40%) differences in the mid-latitudes between 1 and 20 hPa. However theses differences are lower around the tropics.

## 5.3 GODFIT OFFL TROPOMI TOC product

The GODFIT product uses the same Huggins band so that differences between the MUSES product should be attributable to retrieval methods and models.





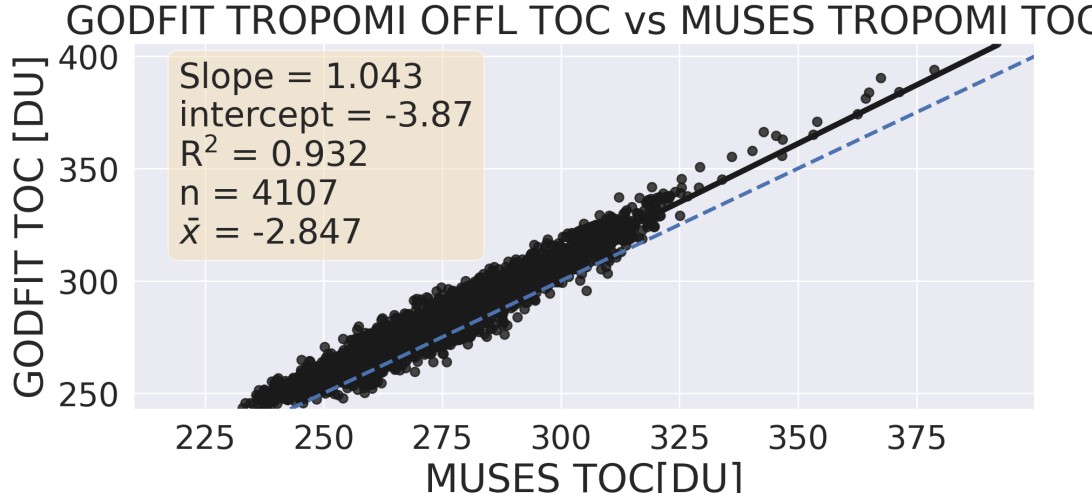

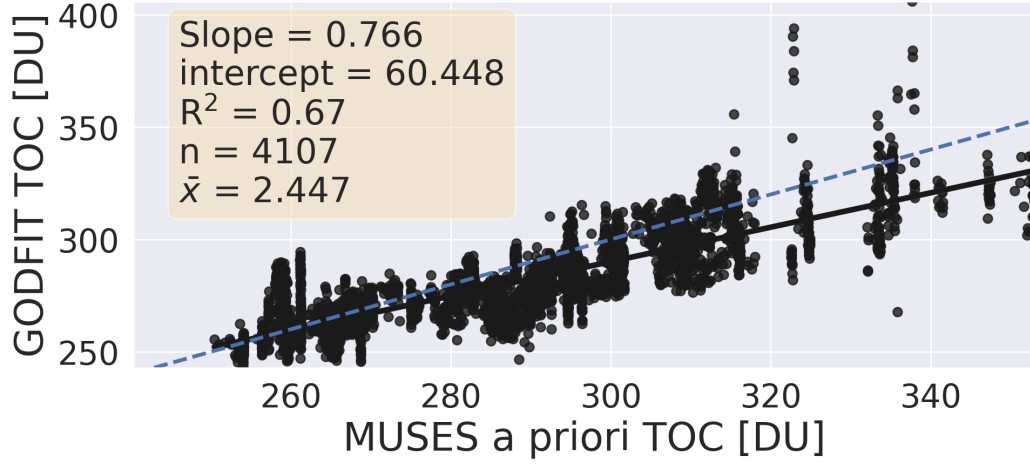

**Figure 12.** Cross-comparisons of TROPOMI OFFL GODFIT TOC product with MUSES TROPOMI TOC on August 12 2020. Comparisons of the MUSES retrieved product with the GODFIT product indicated in the top panel, comparisons with the MUSES a priori columns shown in the bottom panel.

Fig. 12 show high correlation ($R^2$=0.93) and a near unity slope (1.04) with a small positive bias of about 4 DU, which is about 1% of the TOC. These differences are attributable in part to optimal estimation approach used in the MUSES profile retrieval versus the total column fit in the GODFIT algorithm. Both show that restricting the retrieval to the Huggins band limits TROPOMI to a total ozone column.

A significant improvement, both in terms of bias and correlation is apparent when the GODFIT product is compared against the MUSES retrieval product, as opposed to the MUSES a priori.



## 5.4  Chemical reanalysis

The following subsection contains the comparison between the chemical reanalyses CAMS and MOMO-Chem against the
CrIS-TROPOMI/CrIS/TROPOMI retrievals. These reanalyses have distinct advantages over the other comparison sources used
in this paper. These include high spatial and temporal resolution ozone profiles, not limited in vertical sensitivity, while at the
same time taking advantage of the observations of numerous satellite instruments. However, these reanalysis datasets are not
validation sources, and areas of disagreement should be considered as a point for further discussion and evaluation.

The criteria for this co-location are spatial differences of <0.5°, and temporal differences of <2 hours, the temporal resolution
of the chemical reanalysis. The observation operator identified in Eq. 7 is then applied to the co-located reanalysis profiles.



### 5.4.1 CAMS Reanalysis

**Figure 13.** Cross-comparisons of CAMS-reanalysis with CrIS, TROPOMI and CrIS-TROPOMI on August 12 2020 for the tropospheric sub-column. The layout of this figure is as Fig. 11.

Fig. 13(a) show good agreement of the CAMS-reanalysis with CrIS-TROPOMI and CrIS-only results. CrIS-only has a slightly better agreement with a closer linear relationship (slope of 0.851 vs slope of 0.827) and higher correlation ($R^2$:0.785 vs

0.757). TROPOMI-only, however, shows the best agreement of all three instruments. This initially runs counter to the results shown in Figs. 10 and 11. Yet, considering Eq. 7, when using the TROPOMI AK, (which based on Fig. 4 is effectively 0 in the troposphere). The term $\mathbf{H}(\cdot)$ is effectively equivalent to the a priori with only minor TROPOMI sensitivity in the troposphere. Therefore when comparing the TROPOMI troposphere retrievals against CAMS tropospheric column modified by the observation operator what is actually being compared is the TROPOMI sub-column and a slightly modified a priori





sub-column. These two quantities are going to be very similar, hence the close results shown in Fig. 13. This was confirmed when we compared the TROPOMI-only tropospheric column results against CAMS, unmodified by the observational operator. Where the a priori and TROPOMI-only tropospheric column show almost identical comparisons.

For CrIS-TROPOM/CrIS-only retrieval datasets, CAMS overestimates with respect to MUSES, especially for larger or lower magnitude retrievals where more scatter can be observed. Considering the spatial plots shown in Fig. 13(b), we note that while
there are numerous locations where the MUSES datasets individually disagree with CAMS, CrIS-TROPOMI/CrIS-only show multiple instances of >20% difference in large ocean bodies, e.g., the Pacific. Building on this, both Fig. 7 and Fig. 13 show low ozone concentrations in south east Asia. The tropospheric profile results shown in Fig. 13(c) indicate good agreement between CAMS and CrIS-TROPOMI/CrIS-only across most latitudes, with differences <5%, apart from between 0 and 20° where >20% difference is observed. These differences match in location with south East Asia identified in Fig. 13(b). Other
sources of disagreement are focused on the tropopause in the tropics, and at high latitudes for CrIS-TROPOMI, with >20% differences. Plots b) and c) highlight the disagreement between CAMS and the MUSES retrieval in Tropical regions, the causes of these differences should be investigated in future work.

Fig. B1 shows the differences in the stratospheric sub-column. Starting with the scatter plots in Fig. B1(a), all three instruments perform well. CrIS-TROPOMI and CrIS show similar levels of linearity based on the slope and intercept values, although
CrIS has slightly improved performance, while CrIS-TROPOMI has higher correlation ($R^2$=0.936 vs 0.911). TROPOMI does not agree as well as either of the other datasets, but retains a strong correlation ($R^2$=0.93). All datasets show improvements over the a priori. Fig. B1(b) shows that CrIS-TROPOMI largely exhibits a minor negative bias, with some examples of large negative biases towards Antarctica, which is replicated by CrIS-only and TROPOMI-only results. CrIS-only, meanwhile, largely exhibits a positive bias across the globe, with few areas of significant disagreement. TROPOMI-only shows a negative bias
spread across the globe, ranging from 0 to >20%. Key areas of disagreement include the Indian and Pacific oceans. For the profile differences shown in Fig. B1(c), CrIS-TROPOMI generally indicates good agreement (typical difference <10%), with exceptions in the 10-1 hPa range in the mid-high latitudes, where disagreements of ~20% are observed. CrIS-only shows a larger positive bias, typically >10% above 10 hPa, that remains fairly constant across all latitudes, while indicating similar levels of bias as CrIS-TROPOMI about the tropopause. TROPOMI-only generally indicates good agreement through the
stratosphere; however, there are some regions of significant differences, most notably towards Antarctica, where differences greater than 40% are observed through multiple pressure levels.

### 5.4.2 JPL MOMO-Chem

As with using different satellite retrievals for satellite inter-comparisons, we use two chemical reanalysis datasets to provide a wide ranging assessment of the accuracy of CrIS-TROPOMI retrievals. The reanalysis datasets themselves show substantial
disagreements. MOMO-Chem, unlike the CAMS-reanalysis, optimises the precursor emissions of ozone, as well as ozone itself.





**Figure 14.** Cross-comparisons of JPL MOMO-Chem with CrIS-TROPOMI, CrIS and TROPOMI for the tropospheric column level on August 12 2020. The structure of the figure is the same as shown in Fig. 13.

Fig. 14(a) shows excellent agreement between CrIS-TROPOMI and CrIS-only with MOMO-Chem in terms of linearity. The linearity of the comparisons are better than those shown with the CAMS cross-comparison (Fig. 13), however the $R^2$ values which are very similar. The implication from these comparisons is that both MOMO-Chem and CAMS capture the

same enhancements as the MUSES retrievals, but there is less bias between MOMO-Chem and MUSES, as opposed to CAMS, in the troposphere. As with CAMS, TROPOMI comparisons with MOMO-Chem in the troposphere are not relevant when a satellite operator is applied. The spatial differences indicated in the panels under Fig. 14(b) show some similar patterns of agreement/disagreement for CrIS-TROPOMI/CrIS. In the northern hemisphere, there are large regions with significant negative biases, with differences up to 40%. In the southern hemisphere, a general positive bias is shown (~20%), with fewer

locations with larger disagreement. Comparisons with the MOMO-Chem panel show that the major disagreements occur where



MOMO-Chem indicates significant enhancements over oceans, e.g., the Atlantic ocean. Indeed, the AIRS-OMI (Fig. 11) and the CrIS-TROPOMI/CrIS/TROPOMI retrievals (Fig. 7) do not suggest the presence of significant enhancements in the Atlantic ocean, making this a key point of disagreement between the satellite retrievals and the reanalysis estimations. We further note that the tropospheric ozone concentrations are significantly larger than those indicated by CAMS, highlighting a point of

contention between the two reanalysis datasets. The tropospheric differences indicated by the plots shown in Fig. 14(c) show very similar results for CrIS-TROPOMI and CrIS. Good agreement (<10% difference) is found in the northern hemisphere above 40° and in the tropical southern hemisphere. Worse agreement is found in the northern hemisphere tropics to mid-latitudes (up to 40% differences), caused by the disagreement in the Atlantic. Poor agreement is also found below 40°S in the upper troposphere, with differences between 20-40%. It is interesting to note that both CAMS have MOMO-Chem show

specific latitudes of disagreement, in the tropics for CAMS, and northern hemisphere mid-latitude for MOMO-Chem. In summary for the troposphere, good agreement is found between MOMO-Chem and CrIS-TROPOMI and CrIS-only, although disagreement is found in regions of outflow. The improved comparisons of MOMO-Chem in the troposphere over the CAMS-reanalysis, highlighted by the improved linearity in MOMO-Chem, points to the importance of assimilating ozone precursors into the reanalysis and the potential value of direct tropospheric ozone assimilation.

The MOMO-Chem reanalysis is primarily aimed at assessing tropospheric ozone. While the reanalysis levels do extend into the stratosphere, they are relatively limited with respect to the CAMS-reanalysis and the MUSES retrieval grid. Therefore for this study we only focus on tropospheric ozone from MOMO-Chem.

## 5.5 Ozonesondes

Validation of CrIS-TROPOMI/CrIS/TROPOMI retrievals with the ozonesondes was performed from September 2019 to Au-
gust 2020 and split into seasons (Autumn/Winter/Spring/Summer). The ozonesonde validation includes 105 separate ozonesonde soundings, and 3541 individual satellite intercomparisons, based on co-location criteria of <100 km. The MUSES observation operator has been applied following Sec. 3.7 in order to account for vertical sensitivity.







**Figure 15.** Ozonesonde comparisons MUSES CrIS-TROPOMI, CrIS and TROPOMI ozone retrievals. The columns represent the comparisons over different seasons (1=Autumn, 2=Winter, 3=Spring, 4=Summer), the rows represent the instrument comparisons (a=CrIS-TROPOMI, b=CrIS, c=TROPOMI). The solid black lines are the mean difference, the dashed black lines are $\pm 1\sigma$, and the dashed green line is the zero line, these values for the tropospheric columns are indicated on the plots.



**Table 8** Global statistical comparisons between CrIS-TROPOMI/CrIS/TROPOMI and ozonesondes with satellite observation operator applied over a 12 month period in 2019/2020. Analysis is at the pressure levels 681, 464 and 316 hPa.

| Pressure Level | Metric | Autumn CrIS-TROPOMI/ | CrIS/ | TROPOMI | Winter CrIS-TROPOMI/ | CrIS/ | TROPOMI | Spring CrIS-TROPOMI/ | CrIS/ | TROPOMI | Summer CrIS-TROPOMI/ | CrIS/ | TROPOMI |
|---|---|---|---|---|---|---|---|---|---|---|---|---|---|
| **681 hPa** | Mean (ppb) | -5.62 | -6.63 | 2.37 | -3.13 | -3.92 | 2.72 | -1.25 | -2.64 | 2.69 | -1.43 | -2.65 | 2.54 |
| | Mean (%) | -10.7 | -13.0 | 4.39 | -6.13 | -7.96 | 5.26 | -2.26 | -5.55 | 5.11 | -2.69 | -5.10 | 4.97 |
| | RMS (ppb) | 8.70 | 9.51 | 6.16 | 8.07 | 7.53 | 4.95 | 5.91 | 5.87 | 5.38 | 6.16 | 6.65 | 6.03 |
| | RMS (%) | 16.4 | 18.8 | 11.5 | 15.1 | 15.0 | 9.21 | 12.2 | 12.0 | 9.72 | 12.3 | 13.0 | 11.3 |
| **464 hPa** | Mean (ppb) | 0.155 | -2.89 | 1.79 | -1.57 | -3.64 | 2.83 | 2.00 | 1.58 | 2.20 | 3.06 | 0.606 | 2.49 |
| | Mean (%) | -0.154 | -5.67 | 2.99 | -2.34 | -6.72 | 5.07 | 4.91 | 3.01 | 3.83 | 6.17 | 1.67 | 4.24 |
| | RMS (ppb) | 8.21 | 9.29 | 5.73 | 10.0 | 10.0 | 5.46 | 6.79 | 7.73 | 5.16 | 9.14 | 8.16 | 6.34 |
| | RMS (%) | 13.7 | 16.2 | 9.55 | 16.3 | 17.2 | 9.44 | 14.3 | 14.2 | 8.42 | 16.3 | 14.3 | 10.3 |
| **316 hPa** | Mean (ppb) | 4.96 | 4.53 | 0.598 | -0.615 | -0.09 | 3.09 | 5.09 | 9.78 | 1.68 | 5.39 | 6.50 | 1.78 |
| | Mean (%) | 7.21 | 5.97 | 0.954 | -0.874 | -0.325 | 4.26 | 9.41 | 15.9 | 2.16 | 9.56 | 11.0 | 2.52 |
| | RMS (ppb) | 10.6 | 11.68 | 4.14 | 13.6 | 22.6 | 7.81 | 11.4 | 17.3 | 6.50 | 13.0 | 13.3 | 5.91 |
| | RMS (%) | 15.2 | 15.9 | 6.10 | 19.1 | 19.5 | 10.7 | 17.8 | 23.5 | 7.74 | 19.5 | 20.5 | 7.99 |
| Number of profiles | | 25 | | | 32 | | | 20 | | | 28 | | |
| Number of satellite intercomparisons | | 745 | | | 1324 | | | 551 | | | 921 | | |

The results shown in Fig. 15 generally indicate a negative bias in the lower troposphere and a positive bias in the upper troposphere for CrIS-TROPOMI and CrIS. There are numerous outliers apparent, especially in winter with a large negative bias in the upper troposphere. For the winter cases, the upper troposphere bias is less apparent due to these negative outliers. Focusing on the results for TROPOMI, only a minor mean bias is observed through the troposphere, which can be explained due to the lack of TROPOMI sensitivity in the troposphere.

Table 8 quantifies the comparisons shown in Fig. 15. As with the comparisons with CAMS and MOMO-Chem, the low information content of TROPOMI in the troposphere means the application of Eq. 7 yields the TROPOMI precision. When calculating the difference between the TROPOMI profile (which in the troposphere is a noisy replication of the a priori) and the modified ozonesonde profile, the remainder is the TROPOMI precision. Meaning the TROPOMI/ozonesonde comparison is a useful assessment of the predicted TROPOMI precision. According to Fig. 6, TROPOMI precision is typically <5%, which roughly matches the values indicated in Table 8. In general, when compared to the AIRS-OMI/ozonesonde comparisons



identified in Fu et al. (2016), we see similar levels of deviations and differences. For example, at 316 hPa in winter, AIRS and
OMI have a mean difference between 15-22%, while AIRS-OMI has a difference of 7.7%. CrIS-TROPOMI and CrIS-only
show differences of 9-11% at this pressure level. Further the percentage RMS difference at this pressure level is generally
lower for CrIS-TROPOMI/CrIS-only than AIRS-OMI/AIRS/OMI, as well as the other presented pressure levels across all
seasons. Suggesting CrIS-TROPOMI/CrIS-only have comparable or improved performance in the troposphere. We note that
the comparisons of ozonesondes to CrIS-TROPOMI and CrIS-only have similar results across all seasons, and especially in
the RMS values, although there are exceptions to this (e.g., Spring at 316 hPa). This highlights the similar performance, but
also indicates the limited impact of TROPOMI on the CrIS-TROPOMI product in the troposphere, with AIRS-OMI showing a
marked improvement from a combination of AIRS and OMI in most cases. However, the majority of results shown in Table 8 do
indicate that CrIS-TROPOMI performs slightly better than CrIS-only, especially in Autumn. Consequently, the joint retrieval
can give improved performance in the troposphere as well as the stratosphere.

Table 9 shows comparisons of CrIS-TROPOMI/CrIS/TROPOMI against ozonesondes without the satellite operator applied.
This allows for comparisons of instruments with differing sensitivities (i.e. CrIS and TROPOMI), meaning the instrument with
the best sensitivity should show the closest agreement with the ozonesondes. The results in Table 9 show CrIS-TROPOMI
generally as the best performing instrument, except in Winter when CrIS is generally superior. As expected, TROPOMI in
almost all cases is the worst performing.

**Table 9** As Table 8, but showing comparisons with ozonesondes without satellite observation operators applied.

| Pressure Level | Metric | Autumn CrIS-TROPOMI/ CrIS/TROPOMI | | | Winter CrIS-TROPOMI/ CrIS/TROPOMI | | | Spring CrIS-TROPOMI/ CrIS/TROPOMI | | | Summer CrIS-TROPOMI/ CrIS/TROPOMI | | |
|---|---|---|---|---|---|---|---|---|---|---|---|---|---|
| **681 hPa** | Mean (ppb) | -3.41 | -4.40 | 1.24 | 0.805 | 0.180 | 6.25 | -2.26 | -3.68 | 6.35 | -2.31 | -3.56 | 3.56 |
| | Mean (%) | -5.17 | -6.91 | 4.58 | 2.92 | 1.27 | 14.6 | -4.44 | -7.54 | 12.9 | -0.42 | -2.83 | 13.1 |
| | RMS (ppb) | 9.24 | 10.7 | 10.9 | 9.45 | 9.14 | 11.7 | 7.21 | 8.06 | 9.68 | 10.9 | 11.4 | 12.9 |
| | RMS (%) | 16.4 | 18.7 | 20.3 | 20.0 | 19.3 | 25.8 | 14.7 | 16.2 | 20.4 | 25.2 | 25.3 | 36.3 |
| **464 hPa** | Mean (ppb) | 0.303 | -2.81 | 2.34 | 0.983 | -1.34 | 6.94 | -0.977 | -1.50 | 9.68 | 0.995 | -1.55 | 5.25 |
| | Mean (%) | 4.58 | -0.940 | 9.22 | 3.26 | -1.39 | 16.9 | -1.29 | -3.17 | 19.7 | 5.09 | 0.496 | 15.8 |
| | RMS (ppb) | 14.2 | 14.8 | 15.2 | 11.6 | 11.8 | 14.3 | 8.86 | 10.7 | 12.3 | 13.0 | 12.9 | 16.6 |
| | RMS (%) | 25.5 | 24.6 | 27.8 | 22.4 | 22.1 | 30.7 | 16.2 | 18.3 | 24.6 | 23.9 | 22.5 | 35.1 |
| **316 hPa** | Mean (ppb) | 0.679 | 0.214 | 4.82 | -2.07 | -1.23 | 8.90 | 7.45 | 11.9 | 24.6 | 4.70 | 5.82 | 11.0 |
| | Mean (%) | 2.83 | 1.74 | 10.8 | 1.78 | 2.89 | 26.8 | 17.8 | 24.2 | 56.2 | 8.99 | 10.5 | 24.3 |
| | RMS (ppb) | 12.4 | 13.7 | 11.4 | 22.1 | 23.6 | 25.7 | 17.8 | 21.3 | 35.2 | 16.4 | 17.5 | 20.5 |
| | RMS (%) | 19.6 | 21.1 | 23.4 | 31.4 | 34.3 | 50.5 | 33.9 | 40.8 | 70.6 | 24.1 | 25.9 | 42.0 |





## 6 Discussion

### 6.1 Investigating Windows 1 and 2 for TROPOMI in the context of CrIS

This study has focused on a small window within UVIS of TROPOMI; however, as shown in **?**, UV1 & UV2 of TROPOMI provide substantially more information than UVIS, but suffer from significant calibration challenges. In this section we investigate the potential information content of using UV1 & UV2 in combination with MUSES CrIS retrievals. For this purpose, we calculate AKs (as shown in Fig. 4 over a combination of UV1 & UV2 (270-330 nm) and UV2 alone (300-330 nm), assuming perfect calibration accuracy. The first row of Fig. 16 shows the combination of UV1 & UV2 with CrIS, as well as CrIS and TROPOMI on their own. These results indicate a significant boost in information content over the whole profile when compared to those shown in Fig. 4. TROPOMI alone accounts for 5.3 DFS in the profile, as opposed to 1 in this study. The joint profile shows the characteristics of TROPOMI in the stratosphere, but retains some of the characteristics of CrIS in the troposphere. We note that the DFS in the troposphere show an improvement over that of CrIS alone (1.9 from 1.5), suggesting that a DFS boost of more than 25% in the troposphere is gained through the combination of CrIS and TROPOMI.



**Figure 16.** Example theoretical AKs comparisons, the top row shows CrIS-TROPOMI using UV1 & UV2 of TROPOMI (a), compared to CrIS (b) and UV1 & UV2 of TROPOMI (c). The second row shows the same comparison, but only using UV2 of TROPOMI. DFS values for the total profile as well as separate sub-columns are indicated in each plot for comparison purposes.



Building on this assessment, investigations have been held into the CrIS-TROPOMI UV1 & UV2 combination (**?**). This research has found, as suggested in Fig. 16, that a significant boost to the information content is achieved by this combination. However, the CrIS retrieval in this work has a substantially different AK to that presented in this paper, as well as different a

priori and ancillary inputs, suggesting that a future MUSES joint CrIS-TROPOMI UV1 & UV2 retrieval will differ from this work.

## 6.2    TROPOMI UVIS

Zhao et al. (2021) evaluate and discuss the use of a window within UVIS of TROPOMI (314-340 nm) for ozone retrievals. It was found that while UVIS is adversely affected by calibration issues, these are not as significant as those indicated in UV1 &

UV2. Therefore, future steps will be to use the TROPOMI 314-340 nm window for the joint retrieval, once corrected for any calibration issues. The potential information content of this CrIS-TROPOMI UVIS window is shown in Fig. 17.

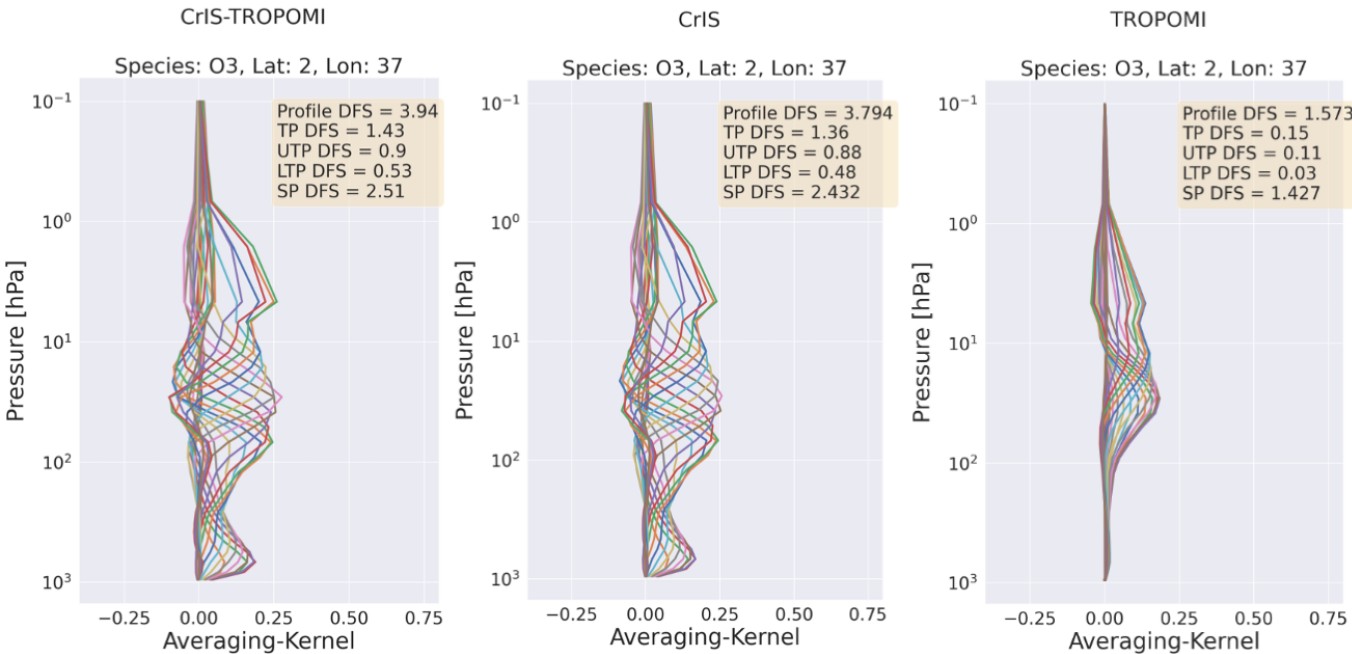

**Figure 17.** Example theoretical AKs comparisons for CrIS-TROPOMI/CrIS/TROPOMI using the 314-340 nm window in the TROPOMI UVIS band. DFS values for the total profile as well as separate sub-columns are indicated in each plot for comparison purposes.

## 7    Conclusions

MUSES TROPESS ozone profile retrievals from joint spectral measurements of Suomi NPP CrIS and S5P/TROPOMI are an important new product in understanding ozone variability. In the stratosphere we find improved performance over and above

each instrument individually especially in the tropics, based on DFS analysis (Figs 5 and C1). Cross comparisons of CrIS-



TROPOMI/CrIS/TROPOMI, with independent datasets from MLS, MUSES AIRS-OMI, JPL MOMO-Chem and CAMS, show in general CrIS-TROPOMI has the highest quality performance relative to the other instruments. Focusing on comparisons with MLS the stratospheric 'gold standard' on August 12th 2020, a linear slope of 1.029, intercept of -7.9 DU (∼3%) and correlation coefficient of 0.952 are found, highlighting the quality of the retrievals. By contrast, despite being a TIR instrument CrIS shows

close linear correlation with MLS, indicating the utility of CrIS by itself. CrIS-only shows a linear slope of 0.921, intercept of 10.3 DU (∼4%) and a correlation coefficient of 0.926. TROPOMI shows a significant bias, with a slope of 0.898 and an intercept of 30.6 (∼12%).

The analysis of CrIS-only is highlighted, with CrIS capable of tropospheric estimates of ozone, with DFS values exceeding 2 in the tropics, roughly equally partitioned between the lower and upper troposphere. As with the CrIS-TROPOMI results,

this is proven through comparisons with ozonesondes. CrIS-only shows a mean bias of between 1.4% and 10.4% depending on the season, comparable to that of AIRS-OMI (Fu et al., 2018). The performance in the troposphere for the joint retrieval is comparable to that of CrIS-only, with evidence that the joint retrieval provides benefit over CrIS-only with mean biases between 0.19% and 7.38%. CrIS-TROPOMI is in better agreement with ozonesondes at 316, 464 and 681 hPa in all seasons, except for a handful of cases (Table 8).

One of the key benefits of the joint retrieval is the use of single pixel retrievals, allowing for estimation of cloud properties, and the ability to see through clouds. Resulting in substantially larger data volumes in the final retrieval products.

Within the cross-comparison subsection, significant deviations are apparent between the MUSES retrievals, and the reanalysis datasets CAMS and MOMO-Chem. For example, CAMS finds relatively low tropospheric ozone concentrations in south east Asia (∼20 DU), while MOMO-Chem identifies a significant enhancement (∼100 DU). The causes of these differences

should be investigated further. It is logical to assume that these differences occur with other reanalysis datasets. All MUSES retrievals, including this work, provide observational operators which can allow for the retrievals to be assimilated into these reanalysis datasets, for further improvement.

The MUSES CrIS-TROPOMI ozone retrieval is current not utilising its full potential; limited to using a short window within UVIS of TROPOMI. Pending the availability of soft-calibration calculations for TROPOMI windows, or improved L1b spectra

(Ludewig et al., 2020; **?**; **?**; Zhao et al., 2021). The MUSES algorithm is capable of processing daily TROPOMI and CrIS L1b spectra in near real time, and is flexible such that when better quality spectra become available, MUSES will immediately be able to take advantage of any improvements. In preparation, we have shown the potential of CrIS-TROPOMI retrievals when using UV1 & UV2 of TROPOMI (Fig. 16), as well as a wider UVIS window (Fig. 17). The utilisation of UV1 and UV2 of TROPOMI provides a significant boost to the stratospheric DFS values (∼ ×2), and can provide a moderate improvement to

tropospheric DFS values (∼0.5 DFS) for the example shown in this work.

The MUSES CrIS-TROPOMI ozone data set is a valuable new product for characterising the variation of ozone in the atmosphere. Future developments and refinements to the MUSES algorithm, will improve the product, and prepare MUSES for future applicable satellite instruments such as Sentinel 4, and joint retrievals from sentinel 5 and IASI-NG.




Both TROPOMI and CrIS are sensitive to trace gases other than ozone, for example carbon monoxide (Fu et al., 2016) and
methane (bands 7 & 8 for TROPOMI). Therefore joint CrIS-TROPOMI retrievals offer an opportunity to explore potential
improvements to other trace gas retrievals.

*Code and data availability.* MUSES data is available from https://tes.jpl.nasa.gov/tropess/get-data/products/. MLS data are available from
https://daac.gsfc.nasa.gov/. The CAMS reanalysis data are available from https://ads.atmosphere.copernicus.eu/cdsapp!/home. MOMO-
Chem data are available from https://tes.jpl.nasa.gov/tes/chemical-reanalysis/. TROPOMI OFFL TOC products are available from
https://s5phub.copernicus.eu/dhus//home. TROPOMI L1b data, cloud products and CrIS L1b data are available from https://daac.gsfc.nasa.gov/.
TROPOMI ILSF is available from http://www.tropomi.eu/data-products/isrf-dataset.

**Appendix A: Impact of loss of CrIS Midwave Channels**

As identified in sect. 2.1, from May 2021 CrIS lost the use of Mid-Wave due to instrument failure. Prior to this failure the
MUSES CrIS retrieval algorithm made use of a number of windows in this band (Table 3). In this appendix we identify the
possible impact of the loss of Mid-Wave on CrIS-TROPOMI/CrIS ozone retrievals by calculating the AKs of CrIS retrievals
without the Mid-Wave windows. An example of these retrievals is identified in Fig. A1.





**Figure A1.** AKs from CrIS, the left hand plot shows an example with all of the original windows, the right hand plot indicates an example missing the Mid-Wave windows. The profile DFS values are identified in the title of each plot.

Figure A1 identifies only a minor loss of information content due to the loss of CrIS Mid-Wave, in this case a drop of 0.05 DFS. This suggests that the CrIS retrievals presented in this paper will only suffer a minor impact due to the loss of the Mid-Wave.



**Figure B1.** Cross-comparisons of CAMS-reanalysis with CrIS, TROPOMI and CrIS-TROPOMI on August 12 2020 for the stratospheric profile. The structure of this figure is as Fig. 10.







**Figure B2.** Cross-comparisons of MUSES AIRS-OMI with CrIS, TROPOMI and CrIS-TROPOMI on August 12 2020 for the stratospheric profile. The structure of the figure is as shown in Fig. 13.



## Appendix C: Further DFS Analysis



**Figure C1.** Visualisation of DFS values for August 12th 2020. Panel a) shows the distribution of DFS values over atmospheric layers for CrIS-TROPOMI/CrIS/TROPOMI. B) shows the spread of DFS values in the troposphere for the instruments, while c) shows the spread in the stratosphere.

*Author contributions.* EM implemented the CrIS-TROPOMI retrieval in MUSES, generated and analysed the data, and wrote the paper. VK developed the major components of MUSES, LK provided the data for AIRS-OMI comparisons and the analysis of the loss of CrIS Mid-Wave, KM provided the JPL MOMO-Chem data, GBO provided the ozonesonde comparison analysis, MDT implemented the ILSF of TROPOMI in MUSES. KWB, TPK and VN guided, provided expertise and managed the research. All authors reviewed the paper.






*Competing interests.* The authors declare no competing interests.

*Acknowledgements.* A portion of this work was carried out at the Jet Propulsion Laboratory, California Institute of Technology, under a contract with the National Aeronautics and Space Administration (80NM0018D0004).



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
