# Peer review of "Joint spectral retrievals of ozone with Suomi NPP CrIS augmented by S5P/TROPOMI"

_EGUsphere, 2022_

## Referee Comment (RC1)

**Referee Report to "Joint spectral retrievals of ozone with Suomi NPP CrIS augmented by S5P/TROPOMI" by Edward Malina et al.**

The manuscript presents an algorithm to retrieve vertical profiles of ozone by combining IR measurements from the CrIS instrument with UV measurements from TROPOMI. The idea has already been explored and results has been published by several authors. The authors highlight that their algorithm differs from the previous ones by a usage of a reduced spectral information, which makes the algorithm less demanding with respect to the calibration of the data reducing, however, substantially the information content in the stratosphere. As a result, the obtained improvement over the pure CrIS retrieval is often vanishing small. The authors must be honest and clearly state this in the abstract. Although the data from other combined CrIS-TROPOMI retrievals are available, the authors make no attempts to make comparisons. Instead many of quite useless comparisons are presented, e.g. with the total ozone product from GODFIT OFFL TROPOMI or ozone field from chemical reanalysis models. The paper is very lengthy and quite boring because of many similar plots (Figs. 10–14), which do not provide much information. In general, however, the obtained results might be useful for certain applications and the manuscript can be published after a major revision.

**Major comments**

- Please be objective in the abstract and conclusions. You completely ignore the fact that for some conditions the results from CrIS-TROPOMI are worse compared to those from CrIS retrieval. Considering only minor differences between the results from CrIS-TROPOMI and CrIS, as presented in the manuscript, the statement "These results demonstrate that CrIS/TROPOMI retrievals have the potential to substantially improve our understanding of ozone." is a clear over-rating.

- The message of the comparisons with a lot of different data products for one single day performed in Sect. 5 (Figs. 10–14) is unclear. As the declared objective of the paper is to present advantages of the CrIS-TROPOMI retrieval, 2-3 comparisons of this kind would be sufficient. Much more interesting would be an analysis of the time evolution, which is completely missing in this part of the paper. Please provide some time evolution plots or at least the plots for different seasons as it is done for ozonesonde comparisons.

- The goal of the comparisons with chemical reanalysis models is totally unclear. The authors state these comparisons cannot be treated as validation, so what exactly is the purpose of this comparisons? The purpose of the comparisons with the total ozone data from GODFIT OFFL TROPOMI is also unclear. Why do you compare the total ozone but not compare the profiles from other CrIS-TROPOMI retrievals? I think the manuscript would largely benefit if you make it shorter by removing Sects. 5.3 and 5.4.

- The comparison with ozonesondes is shown only up to 100 hPa although the ozonesondes deliver reliable ozone profiles up to about 20 hPa. Please present the comparison for the entire altitude range covered by ozonesondes.

- The information presented in Sect. 6.1 has already been published by Mettig et al. (2022) and does not need to be repeated. The fact that the information content increases when using UV bands is generally known. Furthermore, the reliability of an investigation with any assumed settings without having a working retrieval is questionable as adding new spectral ranges requires often an optimization of the settings to keep the retrieval stable, i.e. it is unknown if the retrieval of the real data using the assumed settings is possible. A pre-condition of having a perfect calibration accuracy, as assumed by authors, is never satisfied in the reality. For this reasons, I recommend to skip Sect. 6.

- Line 659: "Focusing on comparisons with MLS the stratospheric 'gold standard' on August 12th 2020, a linear slope of 1.029, intercept of -7.9 DU ($\sim$3%) and correlation coefficient of 0.952 are found, highlighting the quality of the retrievals." - A comparison for one day is definitely not enough to make any robust conclusion. Please extend the comparison with MLS to a longer period similar to the comparison with ozonesondes.

**Minor issues**

- Line 233: Please define the scaling matrix $\mathbf{D}$

- Line 234: " with large $\lambda$ values prioritising the speed of the convergence, but making the steps more non-linear, while small values reduce the speed of the convergence, and is more linear in the iteration, similar to the conjugate gradient method." - In my opinion it is other way around. Small values of $\lambda$ result in the Gauss-Newton method, which converges faster but is more non-linear, while large values of $\lambda$ result in a gradient descent update.

- Figure 1: Please comment why single CrIS and TROPOMI retrievals are necessary before the joint retrieval.

- Line 255: "...as only one FoV per observation from the CrIS cross track position is used in the processing, the impact of clouds will be less pronounced."- please clarify why the impact of clouds is less in this case

- Table 6: Please explain how the cloud fraction is used in the retrieval. Please give some details how albedo is retrieved and what the orders mean, is it wavelength dependence?

- Line 277: "...pixels within a 20 minute time frame (where Suomi-NPP and S5P pass the same scene within 10 mins)" - Please clarify why passing the same scene within 10 minutes results in 20 minutes time frame.

- Line 279: "From the current sounding subset, select all pairs that are within $< 50$ km distance, and 4) select the pair that has the minimum distance." - what is the reason first to select all pairs within 50 km distance and then select the pair with a minimum distance? I expect that selecting a pair with minimum distance skipping the intermediate selection of all pairs within 50 km distance should have the same result. For the pair with minimum distance it can then be checked if the distance is within 50 km.

- Line 283: "... additional steps with respect to other target gases do occur in the pipeline, but are not highlighted here." - This text does not provide any information. Please skip it if the over gases are not related to the ozone retrieval or provide more details otherwise.

- Line 286: "Relevant cloud properties..." - Please clarify how the clouds are handled within the ozone retrieval. Is scattering within clouds considered?

- Figure 2: Please explain how NESR is defined.

- Line 316: "...suggesting that CrIS are subject to larger fit errors,..." - This is not a correct conclusion as mean RMSE for CrIS is the smallest.

- Figure 3 and all figures below having the pressure as the vertical axis: 1) upper limit of the pressure axis must be indicated 2) providing a second y-axis in km would help the interpretation of the results and facilitate the comparison with the results of previous publications;

- Line 321: "In general, longer wavelengths have greater sensitivity in the lower troposphere whereas shorter wavelengths are more sensitive to upper tropospheric ozone." - Please precise which wavelengths and pressure levels you are talking about.

- Equation 3: In the formulation chosen by authors the term "$\delta_{cs}$" must not appear as it is implicitly contained in the $A[x_{true} - x_a]$ term. The citation to (Fu et al., 2018) is inappropriate as they just use the formula form (Worden et al., 2007) without any explanation and misinterpret the notations of (Worden et al., 2007), where the main term was written as $A_{xx}[x_{true} - x_a]$, i.e. included only a sub-matrix of $A$ related to the main parameter. In accordance with Sect. 3.1 of the manuscript under review both $A$ and $x$ contain all retrieved parameters. Thus no additional cross-term must appear.

- Equation 6: As above, the last term is a natural part of the first term and must not appear here.

- Line 363: "For example, focusing on the Atlantic Ocean, there are regions with clearly improved DFS values from CrIS-TROPOMI, as opposed to CrIS." - please explain how this sentence follows from Fig. C1.

- Line 364: "DFS values of 2 are achieved" - "values between 1.5 and 2" would be more correct.

- Line 365: "This suggests that CrIS-TROPOMI and CrIS are highly useful instruments for tropospheric ozone estimation." - the notation "instrument" is incorrect if applied to CrIS-TROPOMI.

- Figure 6 caption: "the measurement or precision error" - do you mean "measurement noise error"?

- Line 382: "with the most reduction at the tropopause." - at which pressure level is the tropopause?

- Line 383: "Given that the majority of the DFS are contained within the stratosphere for CrIS (Fig. 5), this is the expected result." - why the reduction of the uncertainty is expected at the tropopause and not in the stratosphere, where the majority of the DFS are contained?

- Line 383: "except in the lower and upper stratosphere" - please specify pressure levels.

- Line 383: "... the variability of the total uncertainty is smaller than that of CrIS, suggesting that the inclusion of the TROPOMI radiances reduces the uncertainty of the CrIS retrievals." - reducing the variability of the uncertainty does not necessary mean reducing the uncertainty itself.

- Line 424: "30° and 50°" - I guess you mean northern latitudes, please precise.

- Line 443: "For example, in Mongolia" I am not sure everybody can easily find Mongolia on your maps, please provide lat/lon.

- Sect. 5: when discussing stratospheric and tropospheric columns vertical ranges must be specified.

- Line 510: " These results highlight the utility of the CrIS-TROPOMI retrievals in the stratosphere" - It is seen from Fig. B2a that the bias of CrIS-TROPOMI results is larger than for CrIS, i.e. the black solid line is shifted more upwards from the blue dashed line for CrIS-TROPOMI case. Is the intercept calculated correctly? If yes, is it representative for a bias? Also from Fig B2b the bias for CrIS-TROPOMI seems larger (green-blue color) as compared to that for CrIS (green-yellow colors). Please re-calculate the bias using the conventional definition as a mean difference and check your conclusions.

- Line 586: "...e.g., the Atlantic ocean." - please indicate the latitude region you are talking about.

- Figure 15: Please add a comparison for a priori.

- Figure 15: The range of x-axis is unnecessary wide. Please reduce to $\pm 50\%$.

- Line 622: "as well as the other presented pressure levels across all seasons." - do you mean that RMS for CrIS-TROPOMI/CrIS-only is lower at all pressure levels in all seasons? Please reword this part of the sentence to make it more clear.

- Line 629: " ... as well as the stratosphere." - stratospheric results are not considered in Sect. 5.5. Thus, the statement about the stratosphere is not appropriate here.

- Table 9: Please provide statistics for the tropospheric column.

- Line 659: "In the stratosphere we find improved performance..." - the performance is improved only slightly, this must be said. An open question remains if the scatter plot really representative for the bias. This needs to be checked by calculating the mean difference over all data. If needed, conclusions have to be adjusted.

- Line 660: "Cross comparisons of CrIS-TROPOMI/CrIS/TROPOMI, with independent datasets from MLS, MUSES AIRS-OMI, JPL MOMO-Chem and CAMS, show in general CrIS-TROPOMI has the highest quality performance relative to the other instruments" - comparisons with JPL MOMO-Chem and CAMS cannot say anything about performance of one retrieval with respect to the other as it is unclear how to rate the agreement or disagreement with the model data.

- Line 664: "By contrast, despite being a TIR instrument CrIS shows close linear correlation with MLS, indicating the utility of CrIS by itself" - The vertical region should be mentioned, which this statement is applicable to.

- Line 686: "MUSES will immediately be able to take advantage of any improvements." - The statement is questionable as including additional spectral ranges often requires an adjustment of the retrieval parameters and subsequent tests and validation. The statement has to be removed.

- DFS for CrIS retrieval is significantly larger in Fig. 16 in comparison to Fig. 4 (4.12 vs. 3.62), although the scenario is expected to be the same. Please explain why it is the case?

**Technical corrections**

- Lines 126 and 128: Corrupted citations

- Line 279: remove "<", remove "4)"

- Lines 352 and 356: Corrupted citations

- Line 379: "is a maximum" - do you mean "is at maximum" or "is maximum"?

- Line 381: "Comparisons of the total uncertainty with the a priori uncertainty shows.." - the noun is in plural while the verb is in singular

- Line 402: "failures are by and large due to" $\longrightarrow$ "failures are by large due to"

- Line 403: "having too large a magnitude" $\longrightarrow$ "having too large magnitude"

- Line 443: "For example, in Mongolia" $\longrightarrow$ "For example, over Mongolia"

- Line 443: "retrievals had $\sim \times 2$ greater magnitude" - please replace "had" by "have" and write $\sim \times 2$ in words.

- Line 541: "Yet, considering Eq. 7, when using the TROPOMI AK, (which based on Fig. 4 is effectively 0 in the troposphere)." - incomplete sentence

- Line 543: "troposphere retrievals" $\longrightarrow$ "tropospheric retrievals"

- Line 543: "This was confirmed when we compared the TROPOMI-only tropospheric column results against CAMS, unmodified by the observational operator. Where the a priori and TROPOMI-only tropospheric column show almost identical comparisons." - Either it should be one sentence or the second sentence should be reworded to avoid a suboptimal beginning with "where".

- Line 582: "... spatial differences ..." - "spatial distribution of differences" would be more appropriate

- Line 594: "both CAMS have MOMO-Chem" - should it be "both CAMS and MOMO-Chem"?

- Line 622: "lower for CrIS-TROPOMI/CrIS-only than AIRS-OMI/AIRS/OMI" $\longrightarrow$ "lower for CrIS-TROPOMI/CrIS-only than for AIRS-OMI/AIRS/OMI"

- Line 637: Corrupted citation

- Line 647: Corrupted citation

- Line 685: Corrupted citations

**References** Fu, D., Kulawik, S. S., Miyazaki, K., Bowman, K. W., Worden, J. R., Eldering, A., Livesey, N. J., Teixeira, J., Irion, F. W., Herman, R. L., Osterman, G. B., Liu, X., Levelt, P. F., Thompson, A. M., and Luo, M.: Retrievals of tropospheric ozone

profiles from the synergism of AIRS and OMI: Methodology and validation, Atmospheric Measurement Techniques, 11, 5587-5605, https://doi.org/10.5194/amt-11-5587-2018, 2018.

Mettig, N., Weber, M., Rozanov, A., Burrows, J. P., Veefkind, P., Thompson, A. M., Stauffer, R. M., Leblanc, T., Ancellet, G., Newchurch, M. J., Kuang, S., Kivi, R., Tully, M. B., Van Malderen, R., Piters, A., Kois, B., Stübi, R., and Skrivankova, P.: Combined UV and IR ozone profile retrieval from TROPOMI and CrIS measurements, Atmos. Meas. Tech., 15, 2955-2978, https://doi.org/10.5194/amt-15-2955-2022, 2022.

Worden, H. M., Logan, J., Worden, J. R., Beer, R., Bowman, K., Clough, S. A., Eldering, A., Fisher, B., Gunson, M. R., Herman, R. L., Kulawik, S. S., Lampel, M. C., Luo, M., Megretskaia, I. A., Osterman, G. B., and Shephard, M. W.: Comparisons of Tropospheric Emission Spectrometer (TES) ozone profiles to ozonesodes: methods and initial results, J. Geophys. Res., 112, https://doi.org/10.1029/2006JD007258, 2007.

---

## Author Comment (AC1)

Dear reviewer,

Thank you for reviewing our paper, we appreciate that with its length, it was a significant time commitment.

In order to respond to your comments we have kept your original comments in black, our responses are in blue, and changes to the paper are identified with underlined blue.

Reviewer 1:

Referee Report to "Joint spectral retrievals of ozone with Suomi NPP CrIS augmented by S5P/TROPOMI" by Edward Malina et al. The manuscript presents an algorithm to retrieve vertical profiles of ozone by combining IR measurements from the CrIS instrument with UV measurements from TROPOMI. The idea has already been explored and results has been published by several authors. The authors highlight that their algorithm differs from the previous ones by a usage of a reduced spectral information, which makes the algorithm less demanding with respect to the calibration of the data reducing, however, substantially the information content in the stratosphere. As a result, the obtained improvement over the pure CrIS retrieval is often vanishing small. The authors must be honest and clearly state this in the abstract. Although the data from other combined CrIS-TROPOMI retrievals are available, the authors make no attempts to make comparisons. Instead many of quite useless comparisons are presented, e.g. with the total ozone product from GODFIT OFFL TROPOMI or ozone field from chemical reanalysis models. The paper is very lengthy and quite boring because of many similar plots (Figs. 10–14), which do not provide much information. In general, however, the obtained results might be useful for certain applications and the manuscript can be published after a major revision.

Major comments

• Please be objective in the abstract and conclusions. You completely ignore the fact that for some conditions the results from CrIS-TROPOMI are worse compared to those from CrIS retrieval. Considering only minor differences between the results from CrIS-TROPOMI and CrIS, as presented in the manuscript, the statement "These results demonstrate that CrIS/TROPOMI retrievals have the potential to substantially improve our understanding of ozone." is a clear over-rating.

We have removed over-rated statements, replacing them with more moderated statements, for example.

"These results demonstrate that CrIS and CrIS/TROPOMI retrievals have the potential to improve global satellite ozone retrievals, especially with further future developments"

• The message of the comparisons with a lot of different data products for one single day performed in Sect. 5 (Figs. 10–14) is unclear. As the declared objective of the paper is to

present advantages of the CrIS-TROPOMI retrieval, 2-3 comparisons of this kind would be sufficient. Much more interesting would be an analysis of the time evolution, which is completely missing in this part of the paper. Please provide some time evolution plots or at least the plots for different seasons as it is done for ozonesonde comparisons.

Additional comparisons were provided in the supplementary material, however further analysis is now provided. We provide a month long comparison of MLS vs CrIS-TROPOMI/CrIS/TROPOMI, which shows consistent results.

We have also provided a summary of the results from the cross-comparisons and validations to help highlight the take home messages. Which reads as follows:

"The results shown in this paper and the supplementary material shows that CrIS-only agrees well with all datasets compared against it, both in the troposphere and the stratosphere. The addition of the short TROPOMI window to form CrIS-TROPOMI, improves comparisons against MLS in the stratosphere, with all results shown in the main paper and the supplementary material having the lowest mean difference bias of CrIS-TROPOMI/CrIS/TROPOMI. Differences between CrIS-TROPOMI and CrIS-only are less clear in other stratospheric comparison cases, however, differences with CAMS and MOMO-Chem raise interesting scientific questions for further analysis. Comparisons with satellite data and reanalysis in the troposphere do not show clear differences between CrIS and CrIS-TROPOMI, however, CrIS-TROPOMI shows better performance than CrIS-only against ozonesondes, which is indicative of improved performance in the troposphere through joining CrIS and TROPOMI. "

• The goal of the comparisons with chemical reanalysis models is totally unclear. The authors state these comparisons cannot be treated as validation, so what exactly is the purpose of this comparisons? The purpose of the comparisons with the total ozone data from GODFIT OFFL TROPOMI is also unclear. Why do you compare the total ozone but not compare the profiles from other CrIS-TROPOMI retrievals? I think the manuscript would largely benefit if you make it shorter by removing Sects. 5.3 and 5.4.

The purpose of the comparison is to show how CrIS-TROPOMI compares with a chemical reanalysis that in turns has been informed by data. Given the comparisons of CrIS/TROPOMI with independent data, e.g., sondes, this comparison shows qualitatively the information that CrIS/TROPOMI could provide.

Thank you for the points, however we are confused by the reviewer's statement that the comparisons with chemical reanalysis models is unclear.

With respect to GODFIT retrieval, our TROPOMI ozone retrieval is similar to that retrieval, in that the same spectral window is employed. It is our aim therefore to prove that our TROPOMI-only retrieval is comparable to what is currently operationally in use and add confidence to its use in the joint retrievals. This is why do not compare GODFIT to our other retrievals. When we presented our work at conferences, we were specifically recommended to include this analysis, and we therefore think it remains important in this work.

We have added the following text to section 2.3.3 to further clarify.

The GODFIT and the MUSES TROPOMI retrieval algorithms aim to quantify similar quantities (identified in detail in Sect.3.3) through the use of similar spectral windows. It is therefore important to benchmark the MUSES TROPOMI algorithm against the GODFIT algorithm.

• The comparison with ozonesondes is shown only up to 100 hPa although the ozonesondes deliver reliable ozone profiles up to about 20 hPa. Please present the comparison for the entire altitude range covered by ozonesondes.

We have now added this comparison.

• The information presented in Sect. 6.1 has already been published by Mettig et al. (2022) and does not need to be repeated. The fact that the information content increases when using UV bands is generally known. Furthermore, the reliability of an investigation with any assumed settings without having a working retrieval is questionable as adding new spectral ranges requires often an optimization of the settings to keep the retrieval stable, i.e. it is unknown if the retrieval of the real data using the assumed settings is possible. A pre-condition of having a perfect calibration accuracy, as assumed by authors, is never satisfied in the reality. For this reasons, I recommend to skip Sect. 6.

Thank you for this assessment, we accept your point about the perfect calibration not yielding the results as indicated in the paper, however we disagree with the section being unnecessary. We indicate several times that the performance of our CrIS retrieval is significantly different to that shown by Mettig et al. (2022), most notably in the stratosphere. Therefore our theoretical joint TROPOMI bands 1 & 2 – CrIS retrievals are potentially substantially different from those shown by Mettig et al, and are important to show.

• Line 659: "Focusing on comparisons with MLS the stratospheric 'gold standard' on August 12th 2020, a linear slope of 1.029, intercept of -7.9 DU (∼3%) and correlation coefficient of 0.952 are found, highlighting the quality of the retrievals." - A comparison for one day is definitely not enough to make any robust conclusion. Please extend the comparison with MLS to a longer period similar to the comparison with ozonesondes.

In the supplementary material to this paper additional MUSES CrIS-TROPOMI/CrIS/TROPOMI vs MLS comparisons for a number of days are shown, along with further comparisons against the other data sources identified in this paper. We accept that these comparisons were not obvious from the paper, we have now added some additional text to make it obvious these comparisons exist, in the introduction to the 'Validation and cross-comparison' section.

"Cross comparisons for additional days, focusing on August 2020, but with additional months are shown in the supplementary material."

Further, we have now included a comparison of mean differences between MLS and CrIS-TROPOMI/CrIS/TROPOMI over the month of August 2020, please see the new Figure 10. The results showing an improvement in the CrIS-TROPOMI retrievals are consistent over the month, and add weight to our statements.

Minor issues

• Line 233: Please define the scaling matrix D

We have added the following definition:

"the calculation of these values are described in Sects. 5.5 and 6.3 of More (1978)".

• Line 234: " with large λ values prioritising the speed of the convergence, but making the steps more non-linear, while small values reduce the speed of the convergence, and is more linear in the iteration, similar to the conjugate gradient method." - In my opinion it is other way around. Small values of λ result in the Gauss-Newton method, which converges faster but is more non-linear, while large values of λ result in a gradient descent update.

Yes, thank you for correcting us, this has been swapped.

• Figure 1: Please comment why single CrIS and TROPOMI retrievals are necessary before the joint retrieval.

These steps are not necessary for the CrIS-TROPOMI retrieval, but can be used if an updated initial guess is desired by the user. The following text has been inserted.

"Note that Fig 1 shows CrIS and TROPOMI ozone retrievals feeding into the CrIS-TROPOMI retrieval, these steps are not necessary for the CrIS-TROPOMI retrieval, but may be used to provide an updated ozone initial guess depending on user needs."

• Line 255: "...as only one FoV per observation from the CrIS cross track position is used in the processing, the impact of clouds will be less pronounced."- please clarify why the impact of clouds is less in this case

We have deleted this passage, which now just reads as:

"The a priori cloud properties come from an 'initial guess' refinement step using brightness temperature differences".

More details about the cloud clearing methods of the MUSES CrIS-TROPOMI algorithm are provided in section 3.4 with the appropriate context as to why the impact of clouds is less severe than other CrIS retrieval methods.

• Table 6: Please explain how the cloud fraction is used in the retrieval. Please give some details how albedo is retrieved and what the orders mean, is it wavelength dependence?

We added the following sentence with respect to cloud fraction:

"If the cloud fraction is determined to be >0.3, then the retrieval is flagged as poor quality and not considered in further analysis."

The section referring to the albedo values has been modified as follows:

"The a priori for the zero-order albedo term is taken from OMI climatology (Kleipool et al., 2008), with the first and second order albedo terms added to allow for non-linear variation of albedo across the spectral band. Thus the effective albedo forms the quadratic equation,

$$A(\lambda) = A_0 + A_1(1 - \lambda/\lambda_0) + A_2(1 - \lambda/\lambda_0)^2,$$

where A is the effective surface albedo at wavelength $\lambda$, $A_0$, $A_1$ and $A_2$ are the zero, first and second order parameters fit by MUSES, and $\lambda_0$ is the first wavelength."

• Line 277: "...pixels within a 20 minute time frame (where Suomi-NPP and S5P pass the same scene within 10 mins)" - Please clarify why passing the same scene within 10 minutes results in 20 minutes time frame.

We have added the following clarification.

", allowing for some variation in scene overpass times".

• Line 279: "From the current sounding subset, select all pairs that are within < 50 km distance, and 4) select the pair that has the minimum distance." - what is the reason first to select all pairs within 50 km distance and then select the pair with a minimum distance? I expect that selecting a pair with minimum distance skipping the intermediate selection of all pairs within 50 km distance should have the same result. For the pair with minimum distance it can then be checked if the distance is within 50 km.

The sentence has been changed to:

"check all pairs that are within < 50 km…"

• Line 283: "... additional steps with respect to other target gases do occur in the pipeline, but are not highlighted here." - This text does not provide any information. Please skip it if the over gases are not related to the ozone retrieval or provide more details otherwise.

This sentence has been deleted.

• Line 286: "Relevant cloud properties..." - Please clarify how the clouds are handled within the ozone retrieval. Is scattering within clouds considered?

This sentence has been rewritten as follows:

*"Relevant cloud properties (e.g., cloud top height and extinction) for the FoVs are retrieved and passed into the ozone retrieval for quality control, with too optically thick clouds being flagged as poor quality".*

• Figure 2: Please explain how NESR is defined.

We have added the following text:

*"as defined in Zavyalov et al. (2013)".*

• Line 316: "...suggesting that CrIS are subject to larger fit errors,..." - This is not a correct conclusion as mean RMSE for CrIS is the smallest.

This sentence has been replaced with the following:

*"The RMSE values indicate larger standard deviations in the CrIS-TROPOMI and CrIS cases, while CrIS has the lowest mean RMSE. This suggests more variability in the CrIS fits which is understandable given the wider retrieval windows, while TROPOMI has a more constant bias."*

• Figure 3 and all figures below having the pressure as the vertical axis: 1) upper limit of the pressure axis must be indicated 2) providing a second y-axis in km would help the interpretation of the results and facilitate the comparison with the results of previous publications;

Thank you for this point, we have made these changes.

• Line 321: "In general, longer wavelengths have greater sensitivity in the lower troposphere whereas shorter wavelengths are more sensitive to upper tropospheric ozone." - Please precise which wavelengths and pressure levels you are talking about.

This sentence has been replaced with the following:

*"In general, (TIR) have greater sensitivity in the lower troposphere (surface to ~500 hPa) whereas the UV is more sensitive to upper tropospheric (~500 hPa to tropopause) ozone."*

• Equation 3: In the formulation chosen by authors the term "δcs" must not appear as it is implicitly contained in the A[xtrue – xa] term. The citation to (Fu et al., 2018) is inappropriate as they just use the formula form (Worden et al., 2007) without any explanation and misinterpret the notations of (Worden et al., 2007), where the main term was written as Axx[xtrue – xa], i.e. included only a sub-matrix of A related to the main parameter. In accordance with Sect. 3.1 of the manuscript under review both A and x contain all retrieved parameters. Thus no additional cross-term must appear.

Thank you for pointing this out, we have removed all references to cross state error in the equation.

• Equation 6: As above, the last term is a natural part of the first term and must not appear here.

Removed, as per comment above.

• Line 363: "For example, focusing on the Atlantic Ocean, there are regions with clearly improved DFS values from CrIS-TROPOMI, as opposed to CrIS." - please explain how this sentence follows from Fig. C1.

Fig. C1 refers to the following sentence, which we have rewritten to remove potential misunderstanding.

"Further, there are numerous cases for CrIS and CrIS-TROPOMI where DFS values of ~2 are achieved in the whole troposphere (Fig. C1)."

• Line 364: "DFS values of 2 are achieved" - "values between 1.5 and 2" would be more correct.

Thank you for this point, however, this was not the point we are trying to make. We are identifying that these instruments/combinations can achieve 2 DFS in the troposphere, which is highly significant.

• Line 365: "This suggests that CrIS-TROPOMI and CrIS are highly useful instruments for tropospheric ozone estimation." - the notation "instrument" is incorrect if applied to CrIS-TROPOMI.

We removed the term "instrument".

• Figure 6 caption: "the measurement or precision error" - do you mean "measurement noise error"?

Yes, sometimes these are used interchangeably, however we have changed the caption to "measurement noise error".

• Line 382: "with the most reduction at the tropopause." - at which pressure level is the tropopause?

We clarified this statement as follows:

"with the most reduction above the tropopause (~100 hPa)".

• Line 383: "Given that the majority of the DFS are contained within the stratosphere for CrIS (Fig. 5), this is the expected result." - why the reduction of the uncertainty is expected at the tropopause and not in the stratosphere, where the majority of the DFS are contained?

The previous sentence was supposed to read as:

"Comparisons of the total uncertainty with the a priori uncertainty shows a general reduction in the uncertainty, with the most reduction above the tropopause region"

With 'above' replacing 'at'.

• Line 383: "except in the lower and upper stratosphere" - please specify pressure levels.

We have changed this to:

"…above 10 hPa…"

• Line 383: "... the variability of the total uncertainty is smaller than that of CrIS, suggesting that the inclusion of the TROPOMI radiances reduces the uncertainty of the CrIS retrievals." - reducing the variability of the uncertainty does not necessary mean reducing the uncertainty itself.

We have re-written this sentence as follows:

"The key difference is that the variability of the total uncertainty is smaller than that of CrIS, and the total/smoothing error is slightly smaller. Suggesting that the inclusion of the TROPOMI radiances reduces the uncertainty of the CrIS retrievals slightly."

• Line 424: "30∘ and 50∘" - I guess you mean northern latitudes, please precise.

Yes, thank you, corrected.

• Line 443: "For example, in Mongolia" I am not sure everybody can easily find Mongolia on your maps, please provide lat/lon.

Rough coordinates were added.

• Sect. 5: when discussing stratospheric and tropospheric columns vertical ranges must be specified.

At the beginning of section 3.5. we have added a paragraph explaining our pressure level sub-column definitions.

"The following analysis compares ozone retrievals in sub-columns and individual pressure levels. The sub-columns are defined as the troposphere (surface to the tropopause), lower troposphere (surface to 500 hPa), upper troposphere (500 hPa to the tropopause) and the stratosphere (tropopause to 1 hPa)."

• Line 510: " These results highlight the utility of the CrIS-TROPOMI retrievals in the stratosphere" - It is seen from Fig. B2a that the bias of CrIS-TROPOMI results is larger than

for CrIS, i.e. the black solid line is shifted more upwards from the blue dashed line for CrIS-TROPOMI case. Is the intercept calculated correctly? If yes, is it representative for a bias? Also from Fig B2b the bias for CrIS-TROPOMI seems larger (green-blue color) as compared to that for CrIS (green-yellow colors). Please re-calculate the bias using the conventional definition as a mean difference and check your conclusions.

Thank you for pointing this out, we have re-written this section to take this point into account.

"CrIS-TROPOMI indicates a high degree of linearity (slope = 0.997), while CrIS-only and TROPOMI-only show similar linearity to a lesser magnitude. Comparable $R^2$ values are shown for all three cases, while the main difference is the mean difference (bias) with AIRS-OMI, where CrIS-only shows the lowest magnitude difference (-0.03), significantly lower than the other cases."

The mean difference is indicated on all plots, and is now referenced in all discussions relating to these plots.

• Line 586: "...e.g., the Atlantic ocean." - please indicate the latitude region you are talking about.

Added the latitude range 0-30°N, 0-30°W.

• Figure 15: Please add a comparison for a priori.

A priori comparison added to Figure 15, and the percentage difference added to the statistics.

• Figure 15: The range of x-axis is unnecessary wide. Please reduce to ±50%.

Corrected.

• Line 622: "as well as the other presented pressure levels across all seasons." - do you mean that RMS for CrIS-TROPOMI/CrIS-only is lower at all pressure levels in all seasons? Please reword this part of the sentence to make it more clear.

The sentence has been rephrased to:

"The percentage RMS difference for CrIS-TROPOMI/CrIS-only is generally lower than for AIRS-OMI/AIRS/OMI."

• Line 629: " ... as well as the stratosphere." - stratospheric results are not considered in Sect. 5.5. Thus, the statement about the stratosphere is not appropriate here.

Phase has been removed.

• Table 9: Please provide statistics for the tropospheric column.

We have added this in Table 8, but not in Table 9, we do not see the value in this since Table 9 is provided to show the impact of not applying the instrument operator to the ozonesondes, the science in of itself is not that useful, and we therefore do not see the value in adding additional information.

• Line 659: "In the stratosphere we find improved performance..." - the performance is improved only slightly, this must be said. An open question remains if the scatter plot really representative for the bias. This needs to be checked by calculating the mean difference over all data. If needed, conclusions have to be adjusted.

We have changed this sentence to following:

"In the stratosphere we find modest improved performance"

• Line 660: "Cross comparisons of CrIS-TROPOMI/CrIS/TROPOMI, with independent datasets from MLS, MUSES AIRS-OMI, JPL MOMO-Chem and CAMS, show in general CrIS-TROPOMI has the highest quality performance relative to the other instruments" - comparisons with JPL MOMO-Chem and CAMS cannot say anything about performance of one retrieval with respect to the other as it is unclear how to rate the agreement or disagreement with the model data.

We have changed the sentence as follows:

"Cross comparisons of CrIS-TROPOMI/CrIS/TROPOMI, with independent datasets from MLS, MUSES AIRS-OMI, show in general CrIS-TROPOMI has the highest quality performance relative to the other instruments"

• Line 664: "By contrast, despite being a TIR instrument CrIS shows close linear correlation with MLS, indicating the utility of CrIS by itself" - The vertical region should be mentioned, which this statement is applicable to.

We now indicate in the previous sentence that the stratospheric column is compared, with stratospheric column being defined earlier in the paper.

• Line 686: "MUSES will immediately be able to take advantage of any improvements." - The statement is questionable as including additional spectral ranges often requires an adjustment of the retrieval parameters and subsequent tests and validation. The statement has to be removed.

We replaced "immediately" with "quickly", we feel justified in this statement due to significant other retrieval mechanics already in place, so we will not be started from 0.

• DFS for CrIS retrieval is significantly larger in Fig. 16 in comparison to Fig. 4 (4.12 vs. 3.62), although the scenario is expected to be the same. Please explain why it is the case?

Thank you for pointing this out, this difference is because the AKs in Figure 16 are calculated from a forward model run only, which accounts for the difference in DFS values. We have added the following sentence.

"We also only run the forward model to generate these AKs, which causes a minor disparity between the CrIS DFS values in this analysis, and those shown in Fig 4.".

Technical corrections

• Lines 126 and 128: Corrupted citations

Corrected.

• Line 279: remove ">", remove "4)"

Done.

• Lines 352 and 356: Corrupted citations • Line 379: "is a maximum" - do you mean "is at maximum" or "is maximum"?

Thank you, corrected to "is maximum".

• Line 381: "Comparisons of the total uncertainty with the a priori uncertainty shows.." - the noun is in plural while the verb is in singular

Corrected.

• Line 402: "failures are by and large due to" −→ "failures are by large due to"

Corrected.

• Line 403: "having too large a magnitude" −→ "having too large magnitude"

Corrected.

• Line 443: "For example, in Mongolia" −→ "For example, over Mongolia"

Corrected.

• Line 443: "retrievals had ∼ ×2 greater magnitude" - please replace "had" by "have" and write ∼ ×2 in words.

Corrected to:

"have roughly two times".

• Line 541: "Yet, considering Eq. 7, when using the TROPOMI AK, (which based on Fig. 4 is effectively 0 in the troposphere)." - incomplete sentence

Replaced full stop with comma.

• Line 543: "troposphere retrievals" −→ "tropospheric retrievals"

Corrected.

• Line 543: "This was confirmed when we compared the TROPOMI-only tropospheric column results against CAMS, unmodified by the observational operator. Where the a priori and TROPOMI-only tropospheric column show almost identical comparisons." - Either it should be one sentence or the second sentence should be reworded to avoid a suboptimal beginning with "where".

This sentence has been modified as follows:

"This was confirmed when we compared the TROPOMI-only tropospheric column results against CAMS unmodified by the observational operator, given the a priori and TROPOMI-only tropospheric column show almost identical comparisons."

• Line 582: "... spatial differences ..." - "spatial distribution of differences" would be more appropriate

Agreed, changed.

• Line 594: "both CAMS have MOMO-Chem" - should it be "both CAMS and MOMOChem"?

Yes, thank you, changed.

• Line 622: "lower for CrIS-TROPOMI/CrIS-only than AIRS-OMI/AIRS/OMI" −→ "lower for CrIS-TROPOMI/CrIS-only than for AIRS-OMI/AIRS/OMI"

Changed.

• Line 637: Corrupted citation

Corrected.

• Line 647: Corrupted citation

Corrected.

• Line 685: Corrupted citations

Corrected.

References

Fu, D., Kulawik, S. S., Miyazaki, K., Bowman, K. W., Worden, J. R., Eldering, A., Livesey, N. J., Teixeira, J., Irion, F. W., Herman, R. L., Osterman, G. B., Liu, X., Levelt, P. F., Thompson, A. M., and Luo, M.: Retrievals of tropospheric ozone 6 profiles from the synergism of AIRS and OMI: Methodology and validation, Atmospheric Measurement Techniques, 11, 5587-5605, https://doi.org/10.5194/amt-11-5587-2018, 2018.

Mettig, N., Weber, M., Rozanov, A., Burrows, J. P., Veefkind, P., Thompson, A. M., Stauffer, R. M., Leblanc, T., Ancellet, G., Newchurch, M. J., Kuang, S., Kivi, R., Tully, M. B., Van Malderen, R., Piters, A., Kois, B., St¨ubi, R., and Skrivankova, P.: Combined UV and IR ozone profile retrieval from TROPOMI and CrIS measurements, Atmos. Meas. Tech., 15, 2955-2978, https://doi.org/10.5194/amt-15-2955-2022, 2022.

Worden, H. M., Logan, J., Worden, J. R., Beer, R., Bowman, K., Clough, S. A., Eldering, A., Fisher, B., Gunson, M. R., Herman, R. L., Kulawik, S. S., Lampel, M. C., Luo, M., Megretskaia, I. A., Osterman, G. B., and Shephard, M. W.: Comparisons of Tropospheric Emission Spectrometer (TES) ozone profiles to ozonesodes: methods and initial results, J. Geophys. Res., 112, https://doi.org/10.1029/2006JD007258, 2007.

---

## Author Comment (AC2)

Dear reviewer,

Thank you for reviewing our paper, we appreciate that with its length, it was a significant time commitment.

In order to respond to your comments we have kept your original comments in black, our responses are in blue, and changes to the paper are identified with underlined blue.

The paper by Malina et al. presents an ozone profile retrieval approach using CrIS thermal infrared measurements which is slightly improved by a sequential combination with TROPOMI/Sentinel 5P ultraviolet measurements. The method used in the approach and the results are sound. This is well in the scope of AMT. However, the ozone retrievals of the combined CrIS/TROPOMi approach are very similar to those from the CrIS only approach, both in terms of sensitivity and the derived ozone quantities. This mean that the contribution of the combination with TROPOMI is very limited. Since this very important aspect is not very clear in the text of the manuscript, I strongly recommend major revisions of the paper before considering publication. Moreover, the presentation of the method is not very clear and needs thorough revision.

We hope that the corrections applied to your comments, and the comments of reviewer 1 fulfil your criticism.

The principal major revisions I strongly recommend are the following:
- Title: I recommend indicating explicitly that it is a sequential approach from CrIS only to CrIS/TROPOMI and replacing "augmented by" by "and". The augmentation from TROPOMI is very limited in order to point it out so clearly in the title.

We thank the reviewer for this point, however we respectfully disagree. A significant aspect of the paper is building the pipeline for CrIS-TROPOMI retrievals, where we make the point that when calibration on bands 1 & 2 are improved, we will be able to take advantage with minor adjustments to our algorithm. We therefore view this paper as a first step for further papers and developments in the future. Further, we point out benefits to the joint retrieval, through comparisons with MLS, and the ozonesondes.

- Abstract (major remark 1): the enhancement of CrIS-TROPOMI with respect to CrIS only approach should be clearly quantified with precise numbers. The values of degrees of freedom and precision (bias, correlation, rmse) with respect to reference measurements (ozonesondes) for both the stratosphere and troposphere should be clearly provided in the abstract for both CrIS-TROPOMI and CrIS only methods.

We have changed the abstract to read as follows:

Improved performance is characterised in the stratosphere from CrIS-TROPOMI, firstly through a modest increase in the degrees of freedom of signal (often between 0.1-0.2). Secondly, through comparisons with the Microwave Limb Sounder, where a global month long comparison shows a mean difference ~x10 lower than either CrIS or TROPOMI individually, and $R^2$ values 3% higher. In the troposphere, CrIS-TROPOMI and CrIS show

similar degrees of freedom for signal, with about 2 globally, but higher in the tropics partitioned equally between the lower and upper troposphere. CrIS-TROPOMI validation with ozonesondes show improved performance over CrIS-only, with a difference in the tropospheric column bias between 30 and 200% depending on the season. Cross-comparisons with satellite instruments and reanalysis datasets show similar performances in terms of correlations and biases.

- Abstract (major remark 2): it should be clearly stated that the approach is a sequential method first using CrIS only measurements, then TROPOMI data and finally CrIS and TROPOMI jointly. This information is very hard to find in the paper (it only appears in page 15) and it is essential to understand the clear similarity between CrIS only and CrIS-TROPOMI approaches. It should clearly state in the abstract and at the beginning of the presentation of the approach (current section 3).

We have modified the abstract to take this comment into account, the modified sentence is highlighted below.

"The sequential combination of TIR and UV measurements, which follows retrievals from each instrument separately, improves the ability of satellites to characterise global ozone profiles, over the use of each band individually."

- Abstract (major remark 3): The sentence "The results demonstrate that CrIS/TROPOMI retrieval have the potential to substantially improve our understanding of ozone" is too vague and unclear. The paper mainly focuses on the observation of ozone and its precision, and not on the geophysical understanding of ozone-related processes and its evolution. I recommend removing such statement and focus on the metrology aspect of the new approach.

We have updated this sentence to read as follows.

"These results demonstrate that CrIS/TROPOMI retrievals have the potential to improve global satellite ozone retrievals."

- Section 3: A change in the order and explanations in this section is clearly needed to understand the new aspects of the CrIS/TROPOMI approach. I strongly recommend providing at the beginning a clear description of the main flow of information about the ozone profile, clearly stating which is the first approach applied (CrIS only), whose results are then used as a priori of subsequent approaches, etc. This is provided in page 15 (section 3.4) and therefore it is difficult to follow. After this overall explanation, I recommend explaining the details of each box in Figure 1 (current section 3.2, then 3.3 and finally current explanations of section 3.1).

We have now swapped the CrIS-TROPOMI explanation to the beginning of this section, which outlines the pipeline steps, followed by the CrIS and TROPOMI ozone steps. The description of the algorithm follows these sections.

We do not think it is completely necessary to describe each box of figure 1, as some (e.g. non-ozone VMRs) are provided for context and information, and only have limited impact on the ozone retrievals. However, describing them in detail would require substantial more text.

- The differences between CrIS/TROPOMI and CrIS only should be clearly pointed out. Currently this is very difficult to find. For example, no statements of such kind are given for Fig. 7, although it is the global comparison of ozone retrievals.

We are unclear in what the reviewer means exactly here, we point out in many locations in the paper that CrIS-TROPOMI has improved performance in the stratosphere through comparisons with MLS, and the comparisons with ozonesondes show improved performance in the troposphere. We accept that there is little obvious difference in Figure. 7, however, this would be difficult to see for any global comparison, since CrIS-only does a good job of capturing both the troposphere and the stratosphere. We argue that any obvious deviation between CrIS-TROPOMI and CrIS would indicate a problem with the retrievals. There are cases where significant differences occur, but these tend to be filtered out in the quality control.

- The comparison with ozonesondes is expected to be presented before that with respect to other datasets (models, other retrievals) as they are reference measurements.

While we understand the point of the reviewer here, we do not agree. We put the stratospheric comparison first because this is where the most change is seen in the CrIS-TROPOMI retrievals. MLS is the main standard for comparisons in the stratosphere and not ozonesondes, hence why we put the satellite comparisons first in this section.

- It is very important to show correlation coefficients and global scores of the comparison between CrIS/TROPOMI and single band retrievals with respect ozonesondes (Tables 8 and 9)

We have added correlation coefficients to table 8, however table 9 is used purely to show and contrast the impact of not applying the satellite operators to the ozonesonde measurements. We are therefore not convinced as to the benefit of adding correlation coefficients for this table. Our analysis here is as previous papers from our group have shown, e.g. Fu et al. (2018).

Other important remarks:

- Panels in Figures: I recommend assigning different letters for each panel of the figures. Currently many panels are indicated by the same letter, which is then difficult to point out without ambiguity

Thank you for this point, we are however not sure where the difficulty lies. Each panel is clearly labeled with the relevant instrument/instrument combination, we feel this is more obvious than the many labels required by individual alphabetical labelling.

- Figures 4, 5 and 6: differences between CrIS only and CrIS/TROPOMI are very tiny. I recommend providing clear statements with quantified differences.

Thank you for this point. For Figure 4 we feel this has already been made clear, through the statistics highlighted on the figure.

For Figure 5 we have added the following text.

"However, we note that CrIS-TROPOMI shows numerous cases where DFS values have a larger magnitude than CrIS, especially over mid-latitudes."

For Figure 6, there is an explanation in the related text which describes the total error variability of CrIS-TROPOMI is smaller than that of CrIS.

- Quality assessment: Lines 397-400: what about convergence to small spectral residuals of the CrIS/TROPOMI retrieval as compared to CrIS only? How it is possible to have a pass rate of 39% for TROPOMI only and (slightly) larger (40%) for CrIS/TROPOMI which is subsequential step?

Although CrIS-TROPOMI is a subsequent step of TROPOMI, the retrievals are non-linear in nature, and there may be cases where an edge case failure for TROPOMI yields a pass for CrIS-TROPOMI, due to the differences in the retrievals.

- Figure 7: too many tiny panels. I recommend using 2 pressure levels and much bigger panels, as well as indicating in the text the clear similarity between CrIS only and CrIS/TROPOMI retrieved ozone amounts.

We have removed one of the pressure levels in order to make the figure bigger. We have also highlighted in the corresponding text the similarity between the retrievals. For example:

"Note that as shown in the AKs, there are no clear differences between CrIS-TROPOMI/CrIS-only retrievals in the troposphere"

"In general there are few obvious differences between any of the retrieval cases in the stratosphere, with minor differences in magnitude apparent."

Minor revisions:
- Line 273: panel c) of Fig. 1?

We're not sure what the reviewer means here, there is no panel c) of figure 1 and we can't find the text the reviewer is referring to.

- 5 b) TROPOMI: color scale is saturated. It should be changed.

The issue here is that the DFS values for the TROPOMI-only retrievals are almost identical across the whole globe, so we would receive the same result, no matter what colour scale we apply.

- Figure 15: problem with panel indicators

We're not sure what the problem with the indicators is, please elaborate.

Added References

Fu, D., Kulawik, S. S., Miyazaki, K., Bowman, K. W., Worden, J. R., Eldering, A., Livesey, N. J., Teixeira, J., Irion, F. W., Herman, R. L., Osterman, G. B., Liu, X., Levelt, P. F., Thompson, A. M., and Luo, M.: Retrievals of tropospheric ozone 6 profiles from the synergism of AIRS and OMI: Methodology and validation, Atmospheric Measurement Techniques, 11, 5587-5605, https://doi.org/10.5194/amt-11-5587-2018, 2018.

Moré, J. J. (1978). *The Levenberg-Marquardt algorithm: Implementation and theory*. https://doi.org/10.1007/bfb0067700.

Zavyalov, V., Esplin, M., Scott, D., Esplin, B., Bingham, G., Hoffman, E., Lietzke, C., Predina, J., Frain, R., Suwinski, L., Han, Y., Major, C., Graham, B., & Phillips, L. (2013). Noise performance of the CrIS instrument. *Journal of Geophysical Research: Atmospheres*, *118*(23), 13,108-13,120. https://doi.org/10.1002/2013JD020457

---

## Referee Report (RR1)

**Referee report to the revised version of the "Joint spectral retrievals of ozone with Suomi NPP CrIS augmented by S5P/TROPOMI" manuscript by Edward Malina et al.**

In the revised manuscript, all errors and incorrect formulations have been eliminated but not all of my previous comments have been addressed in text or answered properly. The manuscript is still very lengthy and lacks a clear focus. It looks rather like the authors made a lot of work within different studies and just put the results together without carrying much about the necessity and logical flow. In my opinion, the manuscript still needs a revision to make it shorter and more concise. The authors should also clarify if the focus of their study is on the profiles or rather at stratospheric and tropospheric columns. The latter is suggested by the presentation strategy of the validation results but is not clearly stated in the text.

**Detailed comments**

- The authors ignored my request to show a reasonable time evolution of the ozone data (at least for a year). Instead they only extend the MLS comparison to one month and show one additional day in April in the Supplement. This additional day is presented in an absolutely unreadable way for all comparisons in one plot (Fig. S7). No attempt to analyze vertical profiles for a day in a different month was made. Even a sparse illustration of the time evolution for August presented for MLS in Fig. 10 was not repeated for AIRS-OMI comparisons.

- The comparison of the total ozone column from TROPOMI with the GODFIT OFFL TROPOMI TOC product is motivated by authors by the fact that they use the same Huggins band. This motivation is in my opinion not really convincing. If a comparison of the total ozone is really needed why the comparison data should necessary be derived from the Huggins band? Another question is how this comparison is related to objectives of the paper. The total ozone is not a product analyzed in the manuscript and its validation is not really relevant for the results. If the authors think, the total column comparisons might bring some additional information, they should discuss what exactly is the information and how the obtained results are related to the main topic of the manuscript. I also ask myself why then the total ozone columns from CrIS are not included in comparison, or may be even those from AIRS and OMI if these comparisons are really related to the topic.

- Comparisons with CAMS and MOMO-Chem cannot be treated as validation results. In the presented form the comparisons add nothing to the results already presented for AIRS-OMI. If the authors wish to investigate strengths and weaknesses of the chemical models, this must be done with much more details and stronger focus on the problematic locations after the validation part is completed. For this activity, the usage of all three retrievals is absolutely unnecessary. The comparison should be done

using the retrieval which is rated best based on the validation results. Comparisons with more than one intrument are also useful in this case.

- With respect to Sect. 6.1, I agree that the MUSES retrieval using the joint TROPOMI bands 1 and 2 in combination with CrIS measurements will be most probably substantially different from that discussed by Mettig et al., however, the real retrievals will also be substantially different from the synthetic simulations shown in Sect. 6.1 as the noise characteristics and regularization will need to be adjusted and it is impossible to say for now, how strong this potential adjustment will change the results. The results of Sect. 6.1 would be useful if it was the first publication on this topic, but for now, we already know that adding UV bands increases DOFs and we also know that synthetic retrievals often do not reflect the results which will be achieved in the real retrieval after making it stable. Thus, any qualitative assessment performed with synthetic data and not exactly the same retrieval settings as used for real retrievals should be taken with care. For this reason, I still do not agree Sect. 6.1 is useful.

- Line 206: "... facilitating independent validation of the satellite tropospheric ozone products": a model cannot be seen as a reasonable validation source for measured data. The only exception is a usage of a model to assess an overall plausibility of data if no other validation sources are available. Here, it is, however, not the case.

- Figure 2: showing spectral fits from the retrieval would be useful.

- Line 341: "In general, (TIR) have greater sensitivity in the lower troposphere (surface to 500 hPa) whereas the UV is more sensitive to upper tropospheric ( 500 hPa to tropopause) ozone" - this statement is very difficult to follow as there is no 500 hPa mark in the plots. Generally the statement is a bit confusing as it is unclear what e.g. "(TIR) have greater sensitivity in the lower troposphere (surface to 500 hPa)" refers to. The TIR Jacobians are clearly larger for 8 - 100 hPa level than below 500 hPa. With the values of around -6 the Jacobians for UV are much higher than those for TIR (around -15) even in the lower layers. Please be more precise and mark 500 hPa level in the plot.

- Line 381: "CrIS Jacobians, in contrast, show sensitivity at multiple pressure levels due to the many spectral windows used for the CrIS retrieval." - the statement does not seem to be correct. Jacobians have their maxima just below 10 hPa for all windows. First two windows seem to have a dominating contribution while the contribution from the last four windows is rather minor. No clear differentiation of the sensitivity with pressure for different spectral windows is seen from the plot.

- Line 391: "CrIS-TROPOMI generally shows higher DFS values than CrIS alone in both the troposphere and the stratosphere" - For the troposphere, the differences are hardly visible, the statement is poorly justified.

- Line 394: "Further, there are numerous cases for CrIS and CrIS-TROPOMI where DFS values of 2 are achieved in the whole troposphere (Fig. C1)."- If you mean cases in the 2-2.5 bin, they cannot be really defined as "numerous", if you mean 1.5 - 2.0 bin, most of them do not reach 2.

- Line 395: "CrIS-TROPOMI and CrIS are highly useful for tropospheric ozone estimation." - In my understanding, the paper is meant to be focused on highlighting the goals from using TROPOMI, surely CrIS is useful, but does TROPOMI add anything?

- Line 404: "... is largely consistent throughout the atmosphere ..." - what does "consistent" mean in this case?

- Line 405: "... where the magnitude is lower." - do you mean the magnitude (of uncertainty) is higher? If not, which magnitude is meant here?

- Figure 8: the value of the comparison between the daytime and nighttime results of CrIS is unclear. The nighttime data are clearly irrelevant in the framework of the presented study and the comparison should be removed for the sake of brevity and clarity.

- Figure 8: Information on the top and bottom pressure in the panel (c) needs to be provided. Generally, minor ticks on the pressure axis of all plots would increase the readability of the plots.

- Figure 9: The reason to show the nighttime CrIS measurements is unclear. It is also unclear if these measurements are included into the regression fit.

- Line 495: " A key statistic is the CrIS-TROPOMI mean difference (0.32 DU) is significantly lower than either CrIS (3.8 DU) or TROPOMI (-2.88 DU) alone, again indicating improved performance from CrIS-TROPOMI." - here, it should be noted that the improvement of the agreement in the stratospheric columns is most probably caused by a strong negative bias of the CrIS-TROPOMI retrieval above 10 hPa resulting in an error canceling.

- Line 512: "Larger disagreement is apparent above 10 hPa, with differences up to 40% shown outside of the tropics. This is expected as the CrIS-TROPOMI sensitivity decreases above this pressure level, while MLS retains high sensitivity." - In the same way, it must be expected for CrIS-only retrievals but it does not happen. The explanation seems to be unsuitable.

- Figure 10: It is nice that a temporal evolution within August is now shown, however, it is not sufficient to show it only for one month and only for the global mean differences. The plot needs to be extended to cover at least one year and to show differences for different latitude bands.

- Sect. 5.2 and some other sections: it is not a good idea to present an extensive discussion in the main text of figures which are shown in appendices. Either the plots need to be moved to the main text or their discussions (if considered less relevant) need to be moved to appendices.

- Line 533: "have similar differences between -10 and 10% in" - I do not think it is true, there is a lot of cyan (and partially also blue) color in the mid-latitudes, which marks the differences $< -10\%$

- Line 536: "... indicate excellent agreement ..."- CrIS - TROPOMI shows a significant negative bias in panel (a) of Fig. B2, the profile comparison in panel (c) looks just terrible for CrIS - TROPOMI. The rating "excellent agreement" is not acceptable here.

- Line 561: with respect to the CAMS and MOMO-Chem models authors state that "these reanalysis datasets are not validation sources, and areas of disagreement should be considered as a point for further discussion and evaluation.", however, in the following discussion, the models are used in exactly the same way as validation data and no additional discussion and evaluation is provided.

- Line 581: "... with differences $< 5\%$ ..." - I do not think it is true as there is a lot of cyan, blue and yellow colors in the CrIS-TROPOMI/CrIS-only plots.

- Line 665: "improves comparisons against MLS in the stratosphere" - this is only true for stratospheric columns, comparisons for the profiles look much worse than those for CrIS-only. This must be mentioned in the summary.

- Line 667: "differences with CAMS and MOMO-Chem raise interesting scientific questions for further analysis." - either the questions need to be formulated or the statement deleted.

- Line 688: "... future MUSES joint CrIS-TROPOMI UV1 & UV2 retrieval will differ from this work." - most probably it is true but, for now, this "future" retrieval does not yet exist and such a statement is a pure speculation, which should not appear in a scientific manuscript.

- Line 698: " Cross comparisons of CrIS-TROPOMI/CrIS/TROPOMI, with independent datasets from MLS, MUSES AIRS-OMI, show in general CrIS-TROPOMI has the highest quality performance relative to the other instruments." - This statement is not correct. With exception of Fig. 10, CrIS-only retrievals in comparison with MLS and AIRS-OMI show at least the same performance and sometimes are even a bit better than CrIS-TROPOMI.

**Technical corrections**

- Line 41: "... via chemical means are attached as ..." $\longrightarrow$ "... via chemical means, are attached as ..."

- Line 424: "... having too large a magnitude." $\longrightarrow$ "... having too large magnitude."

- Line 528: "... in the tropospheric column. similar magnitudes ..." $\longrightarrow$ "... in the tropospheric column. Similar magnitudes ..."

- Line 576: "CrIS-TROPOM/CrIS-only" $\longrightarrow$ "CrIS-TROPOMI/CrIS-only"

---

## Referee Report (RR2)

**Referee report to the revised version of the "Joint spectral retrievals of ozone with Suomi NPP CrIS augmented by S5P/TROPOMI" manuscript by Edward Malina et al.**

The revised manuscript is significantly shortened and much easier to read. All major issues are fixed. Some minor issues are still to be dealt with. The manuscript can be published after minor corrections listed below.

**Detailed comments**

- LIne 98: I am wondering why OMPS-LP instrument is not mentioned here.

- Line 60 -26: "In the UV, Sentinels 4 and 5 (Ingmann et al., 2012), Tropospheric emissions: Monitoring of pollution (TEMPO) (Zoogman et al., 2017) and the Geostationary Environmental Monitoring Spectrometer (GEMS) (Nicks et al., 2018) will augment this record." - the sentence seems to be grammatically incorrect and should be reworded.

- Line 144 -147: "Chemical reanalysis datasets, which provide global and consistent ozone, can be constrained by multiple satellite observations including precursor data, (e.g., NOx emissions). These cross-comparisons provide insight into how the retrievals are capturing chemical processes. Their data fields use a more stringent co-location criteria owing to their higher data abundance." - this text seem to refer to the removed comparisons and has to be removed as well.

- Fig. 2: It is fine to cite previous studies for a detailed description but all basic terms has to be defined within the paper. Please provide a definition for "Noise Equivalent Spectral Radiance (NESR)" in the text.

- Line 303: "... while TROPOMI has a more constant bias." - Is it really the case? Standard deviation of the bias (absolute residual radiance mean) is largest for TROPOMI.

- Lines 303 -304: "These quality of fit parameters form a part of the MUSES quality assurance procedure (described in 3.6)." - the statement is unclear.

- Line 305: The paragraph starting at this line does not seem to have any connection with the previous text and with the "quality of fit" subsection in general.

- Line 305: "show greater sensitivity in the UV to stratospheric ozone." - with respect to what?

- Line 307: "in the 9.6 and 10.43 micron bands" - the text discusses wavelengths in microns while x-axis is given in wavenumbers. Please use the same units in the figure and in the text.

- Lines 358-359: " For example, focusing on the Atlantic Ocean, there are regions with clearly improved DFS values from CrIS-TROPOMI, as opposed to CrIS." - I cannot identify the discussed feature. Are you referring to Fig. 5 of Fig B1?

- Lines 367-368: "... with the most reduction above the tropopause ( 100 hPa)." - I would rather say the most reduction occurs just below 100 hPa.

- Lines 373-375: "The key difference is that the variability of the total uncertainty is smaller than that of CrIS, and the total/smoothing error is slightly smaller. Suggesting that the inclusion of the TROPOMI radiances reduces the uncertainty of the CrIS retrievals slightly." - looking at the plot it is impossible to say if this statement is true. To my opinion the variability is the same for CrIS-TROPOMI and CrIS-only.

- Line 389: The text "chemical reanalysis datasets and" should be removed as it refers to the removed comparison

- Line 393 and line 394: "the chemical reanalysis or" - same as above

- Fig. 7: the plot has an extremely bad resolution

- Lines 406-407: "The Asian monsoon has been found to enhance tropospheric ozone concentrations (Worden et al., 2009), as can be seen in the spatial maps, as well as the profile plots (Fig. C1)" - for a clear illustration plots outside the monsoon season should be provided

- Lines 411-412: "Both the 316 and 68 hPa pressure levels identify an interesting significant ozone enhancement (roughly double the surrounding regions) in the southern Indian ocean (60°S)." - It is unclear what is meant here, I do not see anything special at 60°S in the Indian ocean.

- Line 422: "in the middle row" - which figure you are talking about?

- Line 445: Here you should provide the reason why comparisons for longer periods are not feasible (as you did in the replies).

- Line 447: You should make a statement whether the profile comparisons for MLS and AIRS-OMI shown below are done with or without AK convolution.

- Lines 450 - 452: "This co-location criterion (along with AIRS-OMI) is the least stringent of all the comparisons shown in this study, due to the reduced coincidences from different satellite orbits and the relative low variability in stratospheric ozone." - I think this statement refers to the deleted comparisons and should be removed as well.

- Line 482: "Differences of between 10-40% are seen" - I think the difference are rather below 20% below 1 hPa.

- Fig. 10: CrIS-TROPOMI retrieval shows a significant jump after June 2020, this needs to be discussed.

- Fig. 10: Suboptimal placement of the legend covering the data points

- Line 523: "For the tropospheric trends ..." - I guess you mean "columns" not "trends".

- Line 536 - 540: "The low information content of TROPOMI in the troposphere means the application of Eq. 8 yields the TROPOMI precision." and the following 3 sentences. - I do not think this is true. With zero precision Eq. 8 gives $H = x_a$, which has nothing to do with the precision and the retrieval in general.

- Lines 544 - 545: "Suggesting CrIS-TROPOMI/CrIS-only have comparable or improved performance in the troposphere." - This might be also the case if the sensitivity is significantly lower.

- Line 558: "...in this paper and the supplementary material..." - is there still a supplementary material?

- Lines 616 - 617: "The MUSES CrIS-TROPOMI ozone data set is a valuable new product for characterising the variation of ozone in the atmosphere." - This was already said in the first sentence of the Conclusions.

- Lines 617: "Future developments and refinements to the MUSES algorithm, will improve the product,..." - I suggest to replace "will" by "are expected" as you cannot know it for sure now.

**Technical corrections**

- Line 124: remove line break in the link

- Caption of Fig. 9: "The three subplots ..." $\longrightarrow$ "The four subplots ..."

- Line 502: Remove "while" at the end of the sentence.

- Line 561: "Comparisons with satellite data and reanalysis in the troposphere" $\longrightarrow$ "Comparisons with satellite data in the troposphere"

- Line 607: "Resulting in substantially larger data volumes in the final retrieval products." - Incomplete sentence.

- Line 612: "MUSES quickly be able" $\longrightarrow$ "MUSES will quickly be able".

---

## Author Response (AR2)

Dear Editor and Reviewer,

Thank you for your continuing efforts in reviewing our paper, in order to respond to your comments we have kept your original comments here in black, our responses are in blue, and specific changes are in underlined blue.

We thank the associate editor and reviewer for their patience in dealing with this paper, and we hope that this response will answer all remaining criticisms.

**Editor Comments**

Dear Ed,

Reviewer #1 (R#1) provided very detailed comments on your paper and suggested major revisions to your manuscript. I agree with him/her that the paper should be shortened and the focus of your paper sharpened. I suggest the following major modifications mainly based upon R#1 to shorten the paper.

Thank you for the review, we respond in line below.

* removal of the total ozone comparison

This comparison has been removed.

* removal of the comparisons with CAMS and MOMO-Chem as these comparisons, given the unknown uncertainties of the models, do not provide clues on the quality of the retrievals

We have removed these comparisons.

* removal of Section 6.1 (synthetic retrievals) as the settings for a retrieval from real data generally differ significantly

Upon further discussion with you, we have decided upon the following action which we hope the reviewer will agree with. We accept the argument from the reviewer that the synthetic retrieval will be different to an eventual operational retrieval, we do not agree that it will be as drastically different to the synthetic retrievals as stated by the reviewer. We feel these retrievals are significantly different from the Mettig el al paper, and are therefore of interest. In addition, the use of the UVIS and CrIS bands isn't shown elsewhere, and is therefore unique. Based on this argument, we have kept this study in the paper.

R#1 made additional valuable comments, which shall be addressed

Please find our responses to the review below.

Please provide a revised manuscript with track changes and an item-by-item reply to the reviewer's and my comments.

Kind regards,
Mark Weber

**Referee report to the revised version of the "Joint spectral retrievals of ozone with Suomi NPP CrIS augmented by S5P/TROPOMI" manuscript by Edward Malina et al.**

In the revised manuscript, all errors and incorrect formulations have been eliminated but not all of my previous comments have been addressed in text or answered properly. The manuscript is still very lengthy and lacks a clear focus. It looks rather like the authors made a lot of work within different studies and just put the results together without carrying much about the necessity and logical flow. In my opinion, the manuscript still needs a revision to make it shorter and more concise. The authors should also clarify if the focus of their study is on the profiles or rather at stratospheric and tropospheric columns. The latter is suggested by the presentation strategy of the validation results but is not clearly stated in the text.

In accordance with your and the associate editors recommendations, we have shorted the length of paper by removing the sections relating to the total column comparison, and the comparisons with the Chemistry Transport Models. The appendix has been shorted, and the supplementary materials removed. We highlight at the beginning of the validation/intercomparison section that from this point the study is focused on column intercomparisons.

Detailed comments

- The authors ignored my request to show a reasonable time evolution of the ozone data (at least for a year). Instead they only extend the MLS comparison to one month and show one additional day in April in the Supplement. This additional day is presented in an absolutely unreadable way for all comparisons in one plot (Fig. S7). No attempt to analyze vertical profiles for a day in a different month was made. Even a sparse illustration of the time evolution for August presented for MLS in Fig. 10 was not repeated for AIRS-OMI comparisons.

We understand the reviewers point, and we do accept that an ideal scenario would have 1-year of comparisons for the satellite retrieval cross-comparisons. Unfortunately, this is not practical at this time, currently we do not have 1-year of CrIS-TROPOMI/CrIS/TROPOMI or AIRS-OMI retrievals readily available. The processing required to create this amount of data, at the resolution generated in the last update requires a significant amount of time, especially for AIRS-OMI retrievals. Due to current HPC scheduling and staffing conflicts at the first author's previous institution where this work was undertaken, creating a 1-year dataset is unfortunately not practical at this time. What we have done to address the reviewer's concerns is extend the CrIS-TROPOMI/CrIS/TROPOMI vs MLS comparisons to two months, and included a couple of days of comparisons in different months to include some information on seasonal bias. We have also added CrIS-TROPOMI/CrIS/TROPOMI vs AIRS-

OMI comparisons for one month and several days in different seasons. We have removed
the results in the supplementary material, and included this data into the time evolution
plots of Figures 10 and 13.

- The comparison of the total ozone column from TROPOMI with the GODFIT OFFL
  TROPOMI TOC product is motivated by authors by the fact that they use the same
  Huggins band. This motivation is in my opinion not really convincing. If a comparison of
  the total ozone is really needed why the comparison data should necessary be derived
  from the Huggins band? Another question is how this comparison is related to objectives
  of the paper. The total ozone is not a product analyzed in the manuscript and its
  validation is not really relevant for the results. If the authors think, the total column
  comparisons might bring some additional information, they should discuss what exactly
  is the information and how the obtained results are related to the main topic of the
  manuscript. I also ask myself why then the total ozone columns from CrIS are not
  included in comparison, or may be even those from AIRS and OMI if these comparisons
  are really related to the topic.

In accordance with your recommendation and the directive of the associate editor, we have
removed the comparisons with the Total Column product.

- Comparisons with CAMS and MOMO-Chem cannot be treated as validation results. In
  the presented form the comparisons add nothing to the results already presented for
  AIRS-OMI. If the authors wish to investigate strengths and weaknesses of the chemical
  models, this must be done with much more details and stronger focus on the
  problematic locations after the validation part is completed. For this activity, the usage of
  all three retrievals is absolutely unnecessary. The comparison should be done using the
  retrieval which is rated best based on the validation results. Comparisons with more
  than one intrument are also useful in this case.

In accordance with your recommendation and the directive of the associate editor, we have
removed the comparisons with CAMS and MOMO-Chem.

- With respect to Sect. 6.1, I agree that the MUSES retrieval using the joint TROPOMI
  bands 1 and 2 in combination with CrIS measurements will be most probably
  substantially different from that discussed by Mettig et al., however, the real retrievals
  will also be substantially different from the synthetic simulations shown in Sect. 6.1 as
  the noise characteristics and regularization will need to be adjusted and it is impossible
  to say for now, how strong this potential adjustment will change the results. The results
  of Sect. 6.1 would be useful if it was the first publication on this topic, but for now, we
  already know that adding UV bands increases DOFs and we also know that synthetic
  retrievals often do not reflect the results which will be achieved in the real retrieval after
  making it stable. Thus, any qualitative assessment performed with synthetic data and not
  exactly the same retrieval settings as used for real retrievals should be taken with care.
  For this reason, I still do not agree Sect. 6.1 is useful.

We accept the argument from the reviewer that the synthetic retrieval will be different to an
eventual operational retrieval, we do not agree that it will be as drastically different to the

synthetic retrievals as stated by the reviewer. We feel these retrievals are significantly different from the Mettig el al paper, and are therefore of interest. In addition, the use of the UVIS and CrIS bands isn't shown elsewhere, and is therefore unique. Based on this argument, and upon further discussion with the associate editor, we have decided to keep this part of the paper.

- Line 206: "... facilitating independent validation of the satellite tropospheric ozone products": a model cannot be seen as a reasonable validation source for measured data. The only exception is a usage of a model to assess an overall plausibility of data if no other validation sources are available. Here, it is, however, not the case.

The comparisons with the CTMs CAMS and MOMO-Chem have been removed, therefore this sentence has been removed.

- Figure 2: showing spectral fits from the retrieval would be useful.

The red lines on Figure 2 are actually the spectral fit of the fit windows we use, this was not obvious from the figure caption, this has been changed.

- Line 341: "In general, (TIR) have greater sensitivity in the lower troposphere (surface to 500 hPa) whereas the UV is more sensitive to upper tropospheric ( 500 hPa to tropopause) ozone" - this statement is very difficult to follow as there is no 500 hPa mark in the plots. Generally the statement is a bit confusing as it is unclear what e.g. "(TIR) have greater sensitivity in the lower troposphere (surface to 500 hPa)" refers to. The TIR Jacobians are clearly larger for 8 - 100 hPa level than below 500 hPa. With the values of around -6 the Jacobians for UV are much higher than those for TIR (around -15) even in the lower layers. Please be more precise and mark 500 hPa level in the plot.

We have modified this section to read as follows:

For the CrIS Jacobians shown in Fig. 3 b) peak ozone sensitivity is found in the 9.6 and 10.43 micron bands primarily between 100 and 10 hPa, but does extend through to the lower troposphere (1000 hPa) and the upper stratosphere (1 hPa), indicating sensitivity through the whole atmosphere, as opposed to TROPOMI-only.

- Line 381: "CrIS Jacobians, in contrast, show sensitivity at multiple pressure levels due to the many spectral windows used for the CrIS retrieval." - the statement does not seem to be correct. Jacobians have their maxima just below 10 hPa for all windows. First two windows seem to have a dominating contribution while the contribution from the last four windows is rather minor. No clear differentiation of the sensitivity with pressure for different spectral windows is seen from the plot.

We have modified to read as follows:

CrIS Jacobians, in contrast, show sensitivity at multiple pressure levels through the troposphere and stratosphere.

- Line 391: "CrIS-TROPOMI generally shows higher DFS values than CrIS alone in both the troposphere and the stratosphere" - For the troposphere, the differences are hardly visible, the statement is poorly justified.

We have replaced this sentence with the following:

Note that according to the histograms in Fig. B1 b) and c) CrIS-TROPOMI can exhibit higher DFS values than CrIS alone in the stratosphere, although as indicated by Fig.4 the differences can be subtle. The differences are very minor for the troposphere, so this is not obvious from the histograms.

- Line 394: "Further, there are numerous cases for CrIS and CrIS-TROPOMI where DFS values of 2 are achieved in the whole troposphere (Fig. C1)."- If you mean cases in the 2-2.5 bin, they cannot be really defined as "numerous", if you mean 1.5 - 2.0 bin, most of them do not reach.

We have changed this sentence to the following, to reduce the strength of the statement:

Further, there are a small number of cases for CrIS and CrIS-TROPOMI where DFS values of more than 2 are achieved in the whole troposphere

- Line 395: "CrIS-TROPOMI and CrIS are highly useful for tropospheric ozone estimation." - In my understanding, the paper is meant to be focused on highlighting the goals from using TROPOMI, surely CrIS is useful, but does TROPOMI add anything?

Thank you for this point, a significant aspect of this paper is to highlight the capability of CrIS-only, as well as CrIS-TROPOMI. We note that according to the DFS figures, indeed TROPOMI does not add that much in terms of DFS, but according to the validation with the ozonesondes, CrIS-TROPOMI does show an improved performance. We would therefore argue that TROPOMI does add value.

- Line 404: "... is largely consistent throughout the atmosphere ..." - what does "consistent" mean in this case?

Upon reviewing this sentence, we decided that as written it does not add much value, and we have therefore removed it.

- Line 405: "... where the magnitude is lower." - do you mean the magnitude (of uncertainty) is higher? If not, which magnitude is meant here?

As with the above comment, we have removed this sentence as it does not add much value.

- Figure 8: the value of the comparison between the daytime and nighttime results of CrIS is unclear. The nighttime data are clearly irrelevant in the framework of the presented study and the comparison should be removed for the sake of brevity and clarity.

Part of the aim of this paper is to inform potential users of what data is available from this algorithm. We therefore believe it is important to highlight the existence of night-time only data, even if it does not play a significant role in the remainder of the paper.

- Figure 8: Information on the top and bottom pressure in the panel (c) needs to be provided. Generally, minor ticks on the pressure axis of all plots would increase the readability of the plots.

We have added additional pressure ticks into this plot.

- Figure 9: The reason to show the nighttime CrIS measurements is unclear. It is also unclear if these measurements are included into the regression fit.

We address this question in a previous comment, the measurements are not included in the regression fit, but are included in the earlier aspects of the paper such as the information content and uncertainty analyses.

- Line 495: " A key statistic is the CrIS-TROPOMI mean difference (0.32 DU) is significantly lower than either CrIS (3.8 DU) or TROPOMI (-2.88 DU) alone, again indicating improved performance from CrIS-TROPOMI." - here, it should be noted that the improvement of the agreement in the stratospheric columns is most probably caused by a strong negative bias of the CrIS-TROPOMI retrieval above 10 hPa resulting in an error canceling.

The following text has been added alongside the longer term comparison figure.

In addition, Fig. 10 further highlights the point shown in Fig. 9 that there is a strong likelihood that the improved performance of the CrIS-TROPOMI stratospheric column values being the result of bias cancellation in different parts of the stratosphere.

- Line 512: "Larger disagreement is apparent above 10 hPa, with differences up to 40% shown outside of the tropics. This is expected as the CrIS-TROPOMI sensitivity decreases above this pressure level, while MLS retains high sensitivity." - In the same way, it must be expected for CrIS-only retrievals but it does not happen. The explanation seems to be unsuitable.

We agree with this statement, we have modified this sentence to read as follows:

Larger disagreement is apparent above 10 hPa, with differences up to 40% shown outside of the tropics, that remain to be investigated.

- Figure 10: It is nice that a temporal evolution within August is now shown, however, it is not sufficient to show it only for one month and only for the global mean differences.

The plot needs to be extended to cover at least one year and to show differences for different latitude bands.

As discussed above in the reviewer's previous comment, we regret that we are limited in the amount of processing that can be done for this study. We have added another month (July) for extended comparisons, and additional days outside of the summer. Due to the relatively sparse nature of these retrievals, and of the other instruments, co-locations are limited and we do not feel there will be additional benefit to comparisons within latitude bands.

- Sect. 5.2 and some other sections: it is not a good idea to present an extensive discussion in the main text of figures which are shown in appendices. Either the plots need to be moved to the main text or their discussions (if considered less relevant) need to be moved to appendices.

We agree with this point, the AIRS-OMI figure that was previously in the Appendix has been moved into the main text and discussed in context there. All other instances (Figures A1 and B1) are only discussed briefly in the main text, and we have kept as in the previous version.

- Line 533: "have similar differences between -10 and 10% in" - I do not think it is true, there is a lot of cyan (and partially also blue) color in the mid-latitudes, which marks the differences < −10%

We have modified this sentence as follows:

Both CrIS-TROPOMI and CrIS have similar differences generally between -10 and 10% in the mid-latitudes through the troposphere, although there are cases with larger differences >40%.

- Line 536: "... indicate excellent agreement ..."- CrIS - TROPOMI shows a significant negative bias in panel (a) of Fig. B2, the profile comparison in panel (c) looks just terrible for CrIS - TROPOMI. The rating "excellent agreement" is not acceptable here.

We have reduced the strength of the language here to good agreement, as we feel this is what is shown in panel (a), the bias is quite low (1-3%). We agree in panel (c) that the differences are more pronounced, but these are discussed further down in the text, and we added the text:

contrasting the overall results shown in (a)

- Line 561: with respect to the CAMS and MOMO-Chem models authors state that "these reanalysis datasets are not validation sources, and areas of disagreement should be considered as a point for further discussion and evaluation.", however, in the following discussion, the models are used in exactly the same way as validation data and no additional discussion and evaluation is provided.

All references to CAMS and MOMO-Chem in this paper have now been removed.

- Line 581: "... with differences < 5% ..." - I do not think it is true as there is a lot of cyan, blue and yellow colors in the CrIS-TROPOMI/CrIS-only plots.

All references to CAMS and MOMO-Chem in this paper have now been removed, so this correction is no longer necessary.

- Line 665: "improves comparisons against MLS in the stratosphere" - this is only true for stratospheric columns, comparisons for the profiles look much worse than those for CrIS-only. This must be mentioned in the summary.

We have modified a sentence in the summary to read as follows:

The addition of the short TROPOMI window to form CrIS-TROPOMI, improves comparisons against MLS stratospheric columns, although challenges remain for stratospheric profile comparisons, requiring further investigations.

- Line 667: "differences with CAMS and MOMO-Chem raise interesting scientific questions for further analysis." - either the questions need to be formulated or the statement deleted.

All references to CAMS and MOMO-Chem in this paper have now been removed.

- Line 688: "... future MUSES joint CrIS-TROPOMI UV1 & UV2 retrieval will differ from this work." - most probably it is true but, for now, this "future" retrieval does not yet exist and such a statement is a pure speculation, which should not appear in a scientific manuscript.

Statement modified to:

Room for further investigation.

- Line 698: " Cross comparisons of CrIS-TROPOMI/CrIS/TROPOMI, with independent datasets from MLS, MUSES AIRS-OMI, show in general CrIS-TROPOMI has the highest quality performance relative to the other instruments." - This statement is not correct. With exception of Fig. 10, CrIS-only retrievals in comparison with MLS and AIRS-OMI show at least the same performance and sometimes are even a bit better than CrIS-TROPOMI.

We have modified this paragraph to read as follows:

Cross comparisons of CrIS-TROPOMI/CrIS/TROPOMI, with independent datasets from MLS, MUSES AIRS-OMI, show some positive results for CrIS-TROPOMI, especially focusing on stratospheric column comparisons with MLS the stratospheric 'gold standard' on August 12th 2020, a linear slope of 1.029, bias of -0.32 DU and correlation coefficient of 0.952 are

found, highlighting the quality of the retrievals. A month long comparison in August 2020 shows a constantly lower bias between MLS and CrIS-TROPOMI and either CrIS or TROPOMI alone. Despite being a TIR instrument CrIS shows high linear correlation with MLS, indicating the utility of CrIS by itself. CrIS-only shows a linear slope of 0.921, bias of 3.8 DU and a correlation coefficient of 0.926. TROPOMI shows a significant bias, with a slope of 0.898. However, room for improvement is also identified, with large biases in the upper stratosphere identified with CrIS-TROPOMI.

**Technical corrections**

- Line 41: "... via chemical means are attached as ..." −→ "... via chemical means, are attached as ..."

Corrected.

- Line 424: "... having too large a magnitude." −→ "... having too large magnitude."

Corrected.

- Line 528: "... in the tropospheric column. similar magnitudes ..." −→ "... in the tropospheric column. Similar magnitudes ..."

Corrected.

- Line 576: "CrIS-TROPOM/CrIS-only" −→ "CrIS-TROPOMI/CrIS-only"

We couldn't find this particular correction, it may have already been corrected.

---

## Author Response (AR3)

Dear associate editor and reviewer, please find our responses inline below. The original text has been kept in black, our responses are in blue, and explicit changes are in underlined blue.

The revised manuscript is significantly shortened and much easier to read. All major issues are fixed. Some minor issues are still to be dealt with. The manuscript can be published after minor corrections listed below.

Detailed comments

• LIne 98: I am wondering why OMPS-LP instrument is not mentioned here.

OMPS-LP has now been added to the sentence.

• Line 60 -26: "In the UV, Sentinels 4 and 5 (Ingmann et al., 2012), Tropospheric emissions: Monitoring of pollution (TEMPO) (Zoogman et al., 2017) and the Geostationary Environmental Monitoring Spectrometer (GEMS) (Nicks et al., 2018) will augment this record." - the sentence seems to be grammatically incorrect and should be reworded.

We have modified this paragraph to flow with the previous sentence about the TIR, and removed the augment this record.

• Line 144 -147: "Chemical reanalysis datasets, which provide global and consistent ozone, can be constrained by multiple satellite observations including precursor data, (e.g., NOx emissions). These cross-comparisons provide insight into how the retrievals are capturing chemical processes. Their data fields use a more stringent co-location criteria owing to their higher data abundance." - this text seem to refer to the removed comparisons and has to be removed as well.

Thank you for spotting, we have removed this.

• Fig. 2: It is fine to cite previous studies for a detailed description but all basic terms has to be defined within the paper. Please provide a definition for "Noise Equivalent Spectral Radiance (NESR)" in the text.

We have now included equation 4, which is the top level definition of NESR based on the CrIS noise defined by Zavyalov et al 2013.

• Line 303: "... while TROPOMI has a more constant bias." - Is it really the case? Standard deviation of the bias (absolute residual radiance mean) is largest for TROPOMI.

We based this statement of the RMSE, which we think gives less weight to large outliers, and therefore a more robust indication of the variability of the fit. However, we have also added the following sentence.

"the absolute standard deviation suggests more large outliers as opposed to CrIS".

• Lines 303 -304: "These quality of fit parameters form a part of the MUSES quality assurance procedure (described in 3.6)." - the statement is unclear.

This sentence has been changed to the following.

"These quality of fit parameters are included in the overall quality control of the retrievals"

• Line 305: The paragraph starting at this line does not seem to have any connection with the previous text and with the "quality of fit" subsection in general.

The Jacobian section must have accidently been moved during other changes, it has now been moved to join the information content section where it fits better.

• Line 305: "show greater sensitivity in the UV to stratospheric ozone." - with respect to what?

The has been changed to, "the most sensitivity"

• Line 307: "in the 9.6 and 10.43 micron bands" - the text discusses wavelengths in microns while x-axis is given in wavenumbers. Please use the same units in the figure and in the text.

Good catch, this was supposed to be in wavenumbers, have changed the text to say wavenumbers.

• Lines 358-359: " For example, focusing on the Atlantic Ocean, there are regions with clearly improved DFS values from CrIS-TROPOMI, as opposed to CrIS." - I cannot identify the discussed feature. Are you referring to Fig. 5 of Fig B1?

This is Figure 5, there are several regions we can identify, in the North Atlantic ocean, and over Egypt, these are admittedly subtle and we will identify this in the text.

• Lines 367-368: "... with the most reduction above the tropopause ( 100 hPa)." - I would rather say the most reduction occurs just below 100 hPa.

Corrected.

• Lines 373-375: "The key difference is that the variability of the total uncertainty is smaller than that of CrIS, and the total/smoothing error is slightly smaller. Suggesting that the inclusion of the TROPOMI radiances reduces the uncertainty of the CrIS

retrievals slightly." - looking at the plot it is impossible to say if this statement is true. To my opinion the variability is the same for CrIS-TROPOMI and CrIS-only.

These differences are subtle, but if you concentrate on the region above 10 hPa, it is apparent. We have made this clearer in the text by modifying the relevant paragraph as follows:

The key difference is that the variability of the total uncertainty (as indicated by the shaded area) is smaller than that of CrIS, most notably above 10 hPa and the total/smoothing error is slightly smaller above 10 hPa. Suggesting that the inclusion of the TROPOMI radiances reduces the uncertainty of the CrIS retrievals slightly in the stratosphere.

• Line 389: The text "chemical reanalysis datasets and" should be removed as it refers to the removed comparison

Removed, thank you.

• Line 393 and line 394: "the chemical reanalysis or" - same as above

Removed, thank you.

• Fig. 7: the plot has an extremely bad resolution

We have replaced this figure to account for your comments below, but this figure is already quite large (1.6Mb), we hope that the publication process will produce the figure at a better resolution, as can be seen outside of latex compilation.

• Lines 406-407: "The Asian monsoon has been found to enhance tropospheric ozone concentrations (Worden et al., 2009), as can be seen in the spatial maps, as well as the profile plots (Fig. C1)" - for a clear illustration plots outside the monsoon season should be provided

Unfortunately the additional processing performed to update this paper has not been done at sufficient spatial resolution to capture these events. We hope future papers that explore this data in more detail will provide this analysis.

• Lines 411-412: "Both the 316 and 68 hPa pressure levels identify an interesting significant ozone enhancement (roughly double the surrounding regions) in the southern Indian ocean (60°S)." - It is unclear what is meant here, I do not see anything special at 60°S in the Indian ocean.

This particular enhancement was mostly visible in the 316 hPa plots, which were removed in a previous iteration. This has now been brought back in this version of the paper where it is much clearer.

• Line 422: "in the middle row" - which figure you are talking about?

This was an accidental reference to an old figure which no longer exists in this paper, have removed.

• Line 445: Here you should provide the reason why comparisons for longer periods are not feasible (as you did in the replies).

We have added the following text.

Limited comparisons are shown due to computational cost in longer term comparisons.

• Line 447: You should make a statement whether the profile comparisons for MLS and AIRS-OMI shown below are done with or without AK convolution.

We have added the following text.

Note the observation operator described in Sec 3.7 is only applied to ozonesondes.

• Lines 450 - 452: "This co-location criterion (along with AIRS-OMI) is the least stringent of all the comparisons shown in this study, due to the reduced coincidences from different satellite orbits and the relative low variability in stratospheric ozone." - I think this statement refers to the deleted comparisons and should be removed as well.

Good catch, we removed this reference.

• Line 482: "Differences of between 10-40% are seen" - I think the difference are rather below 20% below 1 hPa.

We have qualified this, with the following text.

for CrIS-TROPOMI, and are less than 20% for CrIS-only

• Fig. 10: CrIS-TROPOMI retrieval shows a significant jump after June 2020, this needs to be discussed.

We have added the following text.

Note however the mean difference for CrIS-TROPOMI reduces in July 2020 in comparison to earlier retrievals, a trend that is not apparent in CrIS-only. However, the TROPOMI retrievals for these dates do show more negative mean differences, indicating that TROPOMI is the cause of these differences.

• Fig. 10: Suboptimal placement of the legend covering the data points

Updated legend placements

• Line 523: "For the tropospheric trends …" - I guess you mean "columns" not "trends".

We have replaced with tropospheric columns.

• Line 536 - 540: "The low information content of TROPOMI in the troposphere means the application of Eq. 8 yields the TROPOMI precision." and the following 3 sentences. - I do not think this is true. With zero precision Eq. 8 gives H = xa, which has nothing to do with the precision and the retrieval in general.

We possibly jumped a few steps ahead on the details in this paragraph, we have modified to be clearer as indicated below

The low information content of TROPOMI in the troposphere means the application of Eq. 9 to the ozonesonde profile yields the TROPOMI apriori. When calculating the difference between the TROPOMI profile (which in the troposphere is a noisy replication of the a priori) and the modified ozonesonde profile (which is just the apriori after the application of Eq. 9) the remainder is the TROPOMI precision.

• Lines 544 - 545: "Suggesting CrIS-TROPOMI/CrIS-only have comparable or improved performance in the troposphere." - This might be also the case if the sensitivity is significantly lower.

This is possible, but according to Fu et al (2016) both CrIS-TROPOMI and CrIS-only have larger DFS values in the troposphere than AIRS-only and OMI-only, and fairly similar DFS values to AIRS-OMI. Therefore this does not seem convincing to us.

• Line 558: "...in this paper and the supplementary material..." - is there still a supplementary material?

No, we have removed this reference.

• Lines 616 - 617: "The MUSES CrIS-TROPOMI ozone data set is a valuable new product for characterising the variation of ozone in the atmosphere." - This was already said in the first sentence of the Conclusions.

We removed this sentence.

• Lines 617: "Future developments and refinements to the MUSES algorithm, will improve the product,..." - I suggest to replace "will" by "are expected" as you cannot know it for sure now.

Agreed, we have changed to:

Are expected to

Technical corrections

• Line 124: remove line break in the link

Removed.

• Caption of Fig. 9: "The three subplots ..." −→ "The four subplots ..."

Corrected

• Line 502: Remove "while" at the end of the sentence.

Removed

• Line 561: "Comparisons with satellite data and reanalysis in the troposphere" −→ "Comparisons with satellite data in the troposphere"

Corrected

• Line 607: "Resulting in substantially larger data volumes in the final retrieval products." - Incomplete sentence.

We're not sure why this is considered incomplete?

• Line 612: "MUSES quickly be able" –→ "MUSES will quickly be able".

Corrected